# Evolution of T cells in the cancer-resistant naked mole-rat

Tzuhua D. Lin[1,5], Nimrod D. Rubinstein [1,5] ✉, Nicole L. Fong[1], Megan Smith[1], Wendy Craft[1], Baby Martin-McNulty[1], Rebecca Perry[2], Martha A. Delaney[3], Margaret A. Roy[1] & Rochelle Buffenstein [1,4] ✉

Naked mole-rats (NMRs) are best known for their extreme longevity and cancer resistance, suggesting that their immune system might have evolved to facilitate these phenotypes. Natural killer (NK) and T cells have evolved to detect and destroy cells infected with pathogens and to provide an early response to malignancies. While it is known that NMRs lack NK cells, likely lost during evolution, little is known about their T-cell subsets in terms of the evolution of the genes that regulate their function, their clonotypic diversity, and the thymus where they mature. Here we find, using single-cell transcriptomics, that NMRs have a large circulating population of γδT cells, which in mice and humans mostly reside in peripheral tissues and induce anti-cancer cytotoxicity. Using single-cell-T-cell-receptor sequencing, we find that a cytotoxic γδT-cell subset of NMRs harbors a dominant clonotype, and that their conventional CD8 αβT cells exhibit modest clonotypic diversity. Consistently, perinatal NMR thymuses are considerably smaller than those of mice yet follow similar involution progression. Our findings suggest that NMRs have evolved under a relaxed intracellular pathogenic selective pressure that may have allowed cancer resistance and longevity to become stronger targets of selection to which the immune system has responded by utilizing γδT cells.

Lymphocytes provide both innate and adaptive immunity through the ability of their receptors to sense infections and other stressful conditions. CD8 and CD4 T cells have largely evolved to eradicate intracellular and extracellular infections, respectively, through αβT-cell receptor (TCR) -recognition of peptides derived from these respective pathogens, presented on major histocompatibility complex (MHC) I and II, respectively[1–7]. Natural-killer (NK) cells have also evolved to eradicate cells infected with intracellular pathogens through their NK-cell-receptor recognition of deviation from normal expression of MHC-I and elevated expression of stress ligands[8–12]. Whereas NK cells fully develop in bone marrow and their receptors are germline encoded, progenitor T cells leave bone marrow as progenitors and in the thymus undergo somatic genome recombination whereby their

germline-encoded TCR loci are rearranged such that each cell becomes a clonotype encoding a single combination of variable, diversity, and joining TCR regions[1]. This process has evolved to produce the vast variety of αβT-cell clonotypes facilitating recognition of a myriad of possible peptides presented on MHC-I and MHC-II, which in turn, are encoded by highly polymorphic multigene families[13].

During genomic rearrangement some of the T cells are committed to the γδ lineage[14,15], which is not restricted to MHC-I-presented peptides, but rather the γδTCRs recognize both self and foreign stress-ligands[16–18]. γδT cells are the first to emerge in human and mouse embryonic thymuses and subsequently mainly populate peripheral tissues, such as the skin and gut, rather than remain in the circulation[17,19]. Studies in humans have highlighted the presence of

[1]Calico Life Sciences LLC, South San Francisco, California, CA, USA. [2]Department of Biological Science, University of Illinois at Chicago, Illinois, IL, USA. [3]University of Illinois at Urbana Champaign, Illinois, IL, USA. [4]Present address: Department of Biological Science, University of Illinois at Chicago, Illinois, IL, USA. [5]These authors contributed equally: Tzuhua D. Lin, Nimrod D. Rubinstein. ✉e-mail: nrubinstein@calicolabs.com; rbuffen@uic.edu

γδT-cell subsets with semi-invariant and hence public (shared across individuals) clonotypes, which serve innate-like functions, as well as γδT-cell subsets with larger, and hence more private, clonotypes, which serve more adaptive-like roles[20–23]. Thus, γδT cells are thought to expand the temporal and spatial immune responsiveness of αβT cells, bridging the gap between innate and adaptive immunity[24–26]. In recent times, γδT cells have been shown to be involved in cancer, performing both tumorotoxic functions as well as proinflammatory and immunosuppressive roles that favor tumor growth[27,28].

The naked mole-rat (NMR; *Heterocephalus glaber*) is a mouse-sized rodent that shows no age-associated exponential increase in the risk of dying[29] and is remarkably cancer-resistant[30–32], which may greatly contribute to its ability to live exceedingly longer than expected from its body size[33]. A cell-autonomous mechanism explaining the NMR resistance to solid tumors[34] has been challenged[35], although there may be other, yet to be characterized ones, such as evolutionary genomic expansion of tumor suppressor gene families[36], similar to the elephant's *TP53*[37,38]. In keeping with the well-established roles of the immune system in cancer immunosurveillance and maintenance of tissue homeostasis, the NMR immune system has recently garnered attention, with several intriguing findings: (1) NMRs lack NK cells, likely lost due to relaxed intracellular pathogenic selective pressure[39]; (2) NMR T cells mature into foreign-antigen-reactive cells in a unique cervical lymph node[40], unlike most species in which this process is confined to the thymus[41]; (3) contrary to most vertebrates in which thymic cellularity is dramatically reduced by puberty[42], this is not apparent in NMR thymuses[43], even at ages twenty-fold greater than their age of sexual maturity[29]; and, (4) NMRs with a higher colony social rank have enlarged spleens, possibly strengthening their antimicrobial immunity[44]. Despite these findings, a deeper characterization of the NMR immune-cell repertoire, and especially of its various T-cell subsets and how they may relate to its cancer resistance and tissue homeostasis, is still lacking.

We focus our study on NMR T-cell subsets utilizing both conventional single-cell RNA sequencing (scRNA-seq) and single-cell TCR sequencing, as well as extensive comparative genomics. We find that NMRs have a large splenic population of γδT cells and that their genome encodes a large number of γ and δ variable TCR regions. Among the NMR splenic γδT cells, we find a subset with an inhibited cytotoxic transcriptional profile that harbors a dominant and highly public clonotype. Similar to mouse spleens, the CD8 and CD4 αβT-cell subsets make up the majority of the NMR splenic T cells, yet we observe that NMR spleens maintain a considerably lower CD8/CD4 cell ratio and clonotypic diversity. This reinforces the notion that NMRs have evolved under a relaxed intracellular relative to extracellular pathogenic selective pressure. Consistent with that, we also observe that early-life NMR thymuses are considerably smaller than those of mice yet contrary to Emmrich et al.'s report[43], similar to mice undergo involution progression, which based on a comprehensive histological assessment, is already apparent in young adults.

## Results

### NMRs have a large population of γδT cells in the spleen

Our previously generated scRNA-seq data of spleens from adult NMRs (two years old; $n = 2$ females and $n = 2$ males) and mice (two months old; $n = 2$ females and $n = 2$ males) revealed that NMR T cells comprise two subsets, which based on their *Cd8a* expression patterns (expression of *Cd8b* was not detected) were labeled as "naive T cells" and "CD8 T cells"[39]. A subsequent deeper re-inspection of those data showed that the "naive T cells" are also marked by high expression levels of the α and β constant TCR regions, and that the "CD8 T cells", in addition to *Cd8a*, are also marked by high expression levels of the γ and δ constant TCR regions, and of *Gzma*, *Nkg7*, and *Xcl1*, indicative of cytotoxic T-cell function[45,46] (Fig. 1A, C and S1A,

S1C; Data S1). Here, we re-labeled the "naive T cells" as αβT cells and the "CD8 T cells" as γδT cells. In contrast, in the equivalent mouse spleen dataset, expression of the γ and δ constant TCR regions is not detected and *Gzma* and *Nkg7* are mainly expressed in NK cells (Fig. 1B, C and S1B, S1C; Data S1).

To obtain higher resolution of NMR T-cell subsets and to examine their changes with age, we generated new spleen T-cell-specific scRNA-seq datasets, from adult NMRs (two years old; $n = 3$ males) and mice (two months old; $n = 4$ males), and old NMRs (26−28 years old; $n = 4$ males) and mice (two years old; $n = 4$ males), using the more advanced 10× V3 chemistry (Fig. S1D, S1E; Data S2 and S3; Methods). These higher resolution datasets revealed that NMR T cells comprise five subsets: (1) a subset marked by high expression levels of the β constant TCR region and *Cd8a* (yet absent in expression of *Cd8b*), labeled as CD8 T cells; (2) a subset marked by high expression levels of the β constant TCR region and *Cd4*, labeled as CD4 T cells; (3) a subset marked by high expression levels of the γ and δ constant TCR regions, the *Gzma*, *Nkg7*, and *Xcl1* cytotoxicity-related genes, the *Il2rb* IL-2 induced proliferation gene, the *Klra1* and *Klrd1* cytotoxicity inhibitory genes, and of *Cd8a* (yet absent in expression of *Cd8b*), labeled as cytotoxic γδT cells; (4) a subset marked by high expression levels of the γ and δ constant TCR regions and absence of expression of *Cd8a*, *Cd8b*, *Gzma*, *Nkg7*, *Xcl1*, *Il2rb*, *Klra1*, and *Klrd1*, labeled as non-cytotoxic γδT cells; and (5) a subset largely lacking expression of the genes encoding the CD3 universal T-cell marker (*Cd3d*, *Cd3e*, *Cd3g*, and *Cd247*), highly expressing the *Rorc* transcription factor, which acts as a master regulator for differentiation and function of group 3 innate lymphoid cells (ILCs)[47], and markers of groups 1 and 2 ILCs, hence labeled as ILCs (Fig. 1D, F and S1D, S1F; Data S1 and S2).

In mice, during thymic development a large proportion of γδT cells commit to lineages, which upon stimulation, produce either IFN-γ or IL-17, where the former provides cytotoxic, anti-cancer, and anti-infection functions and the latter anti-infection functions but also tumor-favoring functions, such as angiogenesis and immune suppression[48]. While *Nkg7*, *Gzma*, and *Il2rb*, which mark NMR cytotoxic γδT cells, are known to mark mouse IFN-γ producing γδT cells[48,49], other mouse IFN-γ producing γδT-cell markers, such as *Cd27* and *Nk1.1* (*Klrb1c*)[48], are either not expressed in the NMR data; not uniquely highly expressed by NMR cytotoxic γδT cells; or have no NMR orthologs. This situation is similar for genes typically marking mouse IL-17 producing γδT cells, such as *Il17a*, *Cd44*, *Scart1*, and *Scart2*[48] and applies to non-TCR receptors of stress-ligands in mouse γδT cells, such as *Nkg2d* (*Klrk1*) and Toll-like receptors (TLRs)[50]. Hence, these data do not allow determining with high certainty whether the NMR splenic cytotoxic and non-cytotoxic γδT cells are homologous to the mouse tissue-resident IFN-γ and IL-17 producing γδT cells.

In the equivalent mouse dataset, splenic T cells were found to comprise eight subsets: (1) naive CD8 T cells; (2) naive CD4 T cells; (3) memory CD8 T cells; (4) memory CD4 T cells; (5) memory CD8 T cells with high expression levels of the *Gzmk* cytotoxic granule gene, previously reported as a hallmark of aged tissues[51]; (6) NKT cells; (7) regulatory T cells (Tregs); and (8) γδT cells (Fig. 1E, F and S1E, S1A; Data S1 and S3).

The major age-related changes observed in the NMR dataset are in ILCs, detected in only two of four old-age samples and in none of the adult-age samples. In addition, cytotoxic γδT cells are overrepresented among the old-age samples whereas CD8 T cells are underrepresented among the old-age samples (both multiple-hypotheses-adjusted $p \ll 0.05$, Methods; Fig. S1D). In the mouse dataset, all three memory T-cell subsets (CD8, CD4, and the *Gzmk*-high CD8) are predominantly found in the old-age samples, whereas the two naive T-cell subsets (CD8 and CD4) are, as expected[41], overrepresented among the adult-age samples (multiple-hypotheses-adjusted $p \ll 0.05$, Methods; Fig. S1E).

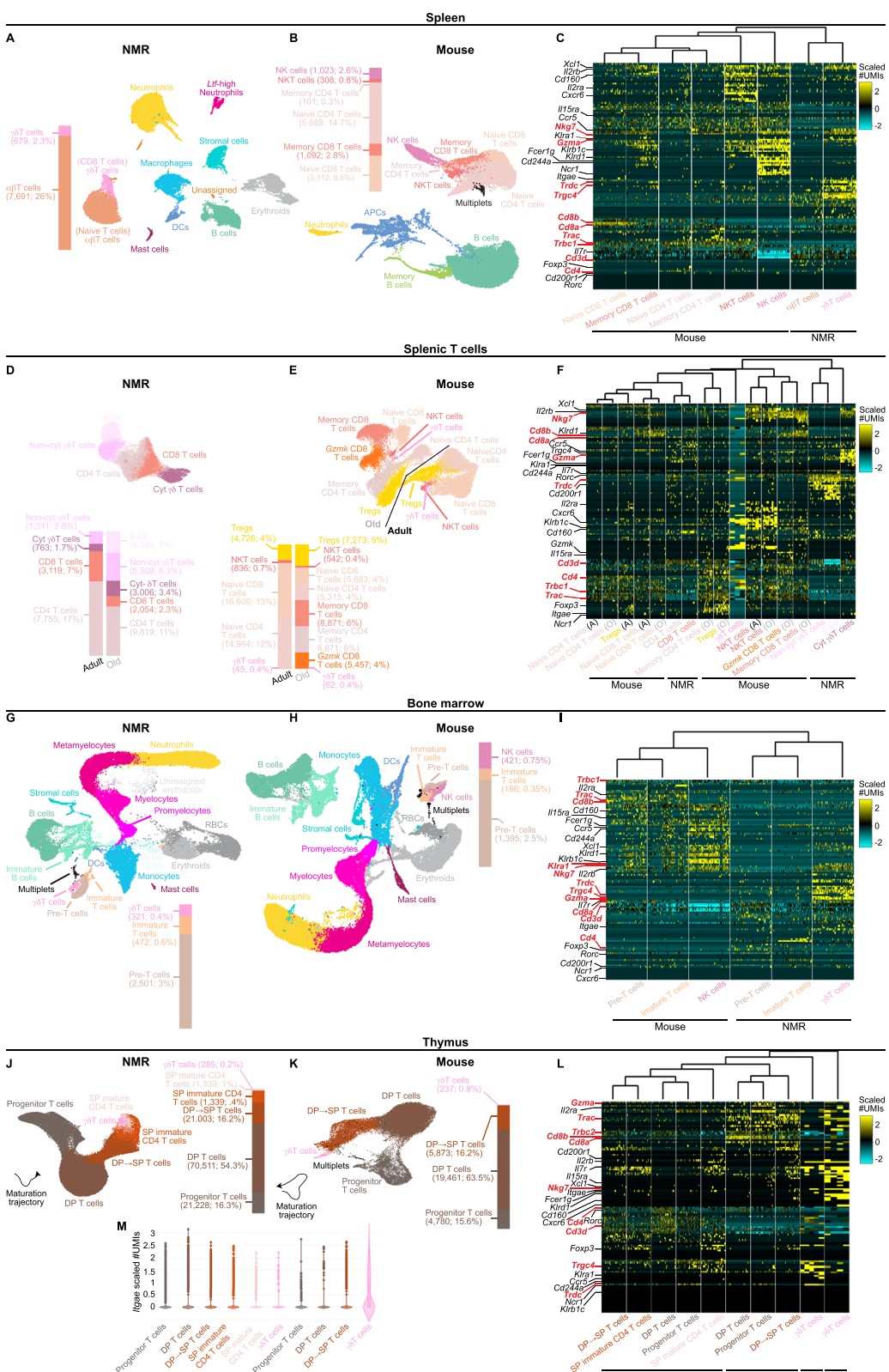

## NMR γδT cells are observed in bone marrow

To obtain a view of T cells at their tissue of birth, we additionally scRNA-seq profiled bone marrows of adult NMRs and mice (n = 2 females and n = 2 males, in each species; Fig. S1G, S1H, respectively; Data S4 and S5, respectively). Two-dimensional uniform manifold approximation and projection (UMAP) embeddings revealed that both species show a similar composition of hematopoietic cell types and differentiation

trajectories (Fig. 1G, H, respectively). Focusing specifically on T cells, three subsets were identified in NMRs: (1) pre-T cells; (2) immature T cells; and (3) cells marked by high expression levels of the γ and δ constant TCR regions, *Cd8a* (yet absence of *Cd8b* expression), and the *Gzma* and *Nkg7* cytotoxicity-related genes, labeled as γδT cells (Fig. 1G, I and S1I; Data S1). Since mature γδT cells might circulate through and/or home to bone marrow[52], we cannot reliably determine

**Fig. 1 | ScRNA-seq of circulating T-cell subsets reveals that the NMR has a large population of γδT cells.** UMAP projections and corresponding gene-by-cell expression-level heatmaps of splenocytes from: (**A**) four adult NMRs (n = 2 of each sex), (**B**) four adult mice (n = 2 of each sex), and (**C**) the corresponding heatmap of the NK and T -cell subsets of the two species; T cells from spleens of (**D**) three adult and four old NMRs (all males), (**E**) four adult and four old mice (all males), and (**F**) the corresponding heatmap of the two species; Bone-marrow cells from (**G**) four adult NMRs (n = 2 of each sex), (**H**) four adult mice (n = 2 of each sex), and (**I**) the corresponding heatmap of the T-cell subsets of the two species; T cells from (**J**) cervical and thoracic thymuses of seven adult NMRs (n = 4 females and n = 3 males), (**K**) thoracic thymuses of four adult mice (n = 2 of each sex), and (**L**) the corresponding heatmap of the two species. UMAPs provide a two-dimensional overview of all cells, color-coding them by their annotated cell-type subset. Stacked bar

charts to the sides of the UMAPs show the proportions of cells assigned to each T-cell subset out of the total immune-cell population in that tissue, as well as the absolute number of cells in each subset. Heatmaps show normalized expression levels of selected marker genes (marker genes relevant for identification of the various T-cell subsets are labeled in red) in each cell, where cells are faceted by their cell subset. Maturation trajectory arrows in (**K, L**) are knowledge based. **M** A violin plot of the distributions of the normalized *Itgae* expression levels in each of the NMR and mouse thymocyte subsets, highlighting the high expression levels unique to the mouse γδT-cell subset. Abbreviations: APC Antigen Presenting Cell, Cyt Cytotoxic, DC Dendritic Cell, DP Double Positive, ILC Innate Lymphoid Cell, NK Natural Killer, NKT Natural Killer T, Non-cyt Non-cytotoxic, RBC Red Blood Cell, SP Single Positive, Treg Regulatory T. Source data are provided as Source data file.

whether the cytotoxic and non-cytotoxic γδT-cell subsets observed in the NMR spleen already emerge in bone marrow. In mice we also observed a subset of pre and immature -T cells, as well as a subset of NK cells (Fig. 1G, I and S1I; Data S1). The absence of NK cells in NMR bone marrow, in contrast to their notable proportion in mouse bone marrow, further supports our premise of their evolutionary loss[39]. Interestingly, both species have strikingly similar proportions of the three T-cell subsets from their respective total bone-marrow cell populations. In NMRs, 2.7% ± 0.2 are pre-T cells, 0.5% ± 0.05 are immature T cells, and 0.35% ± 0.01 are γδT cells whereas in mice, 2.2% ± 0.05 are pre-T cells, 0.3% ± 0.005 are immature T cells, and 0.65% ± 0.01 are NK cells (sample means ± standard error) (Fig. 1G, H). Moreover, in our previous splenocyte scRNA-seq data[39], the proportions of the NMR γδT- and mouse NK -cell subsets are also similar (NMR: 2.3% ± 0.1; mouse: 2.6% ± 0.1) as are the proportions of the αβT-cell subsets in both species (NMR: 26% ± 1.3; mouse: 25% ± 1.5) (Fig. 1A, B, respectively). Hence, NMRs maintain a circulating γδT-cell population equivalent in proportion to the mouse circulating NK-cell population.

### NMR γδT cells are observed in the thymus

Since T cells mature in the thymus, to further track the NMR T-cell development, we generated thymic scRNA-seq data. Ref. 40. first reported that NMRs have a unique cervical lymph node where their T cells mature into foreign-antigen-reactive cells, that in their subsequent publication was referred to as an ectopic cervical thymus, present in addition to their thoracic thymus[43]. We located and scRNA-seq profiled both thymic tissues from adult NMRs (n = 4 females and n = 3 males) and found that each of the thymocyte subsets is roughly uniformly represented in all thoracic and cervical thymic samples (Figs. 1J and S1J; Data S6), suggesting that both thymic tissues are transcriptionally (and hence likely functionally) identical, although the cervical tissue is significantly larger (p < 0.05) in size, weight, and number of cells (Fig. S1M–S1O). We additionally scRNA-seq profiled adult mouse thymuses (n = 2 females and n = 2 males; Fig. S1K; Data S7) and observed similar thymocyte subsets (Fig. 1J, K, respectively). UMAPs revealed that thymocytes of both species exhibit an equivalent maturation trajectory, each with a cell subset comprising less than 1%, located at the end of the maturation trajectory, with high expression levels of the γ and δ constant TCR regions, hence labeled as γδT cells (Fig. 1J–L and S1L; Data S1). Whereas the mouse γδT cells show high and ubiquitous expression levels of *Itgae*, one of the two genes encoding the CD103 integrin that homes mouse and human γδT cells to their epithelial target tissues upon their egress of the thymus[53], this is not the case in the NMR γδT cells (Fig. 1L, M and S1L; Data S1), supporting our findings of a large proportion of NMR splenic γδT cells.

### The NMR *Cd8b* gene has been evolving under relaxed purifying selection

Among all *Cd8a*-expressing NMR T-cell subsets from all examined immune tissues, only the thymic immature double positive subset shows noticeable expression levels of *Cd8b*, similar to that of *Cd8a* in

these cells (Figs. 1L and S1L). This is in stark contrast to all mouse CD8 T-cell subsets, where expression levels and breadth of *Cd8b* matches that of *Cd8a* in all examined tissues (Figs. 1C, F, I, L and S1C, S1F, S1I, S1L). Based on this interspecies difference, we hypothesized that the molecular functions of the NMR *Cd8b* diverged from that of mice and humans, during evolution. We therefore contrasted the intensity of purifying selection operating on the *Cd8b* coding region in the genomes of NMR and a closely related hystricomorph, the Damaraland mole-rat (DMR; *Fukomys damarensis*), with that in genomes of representatives from the murine clade (Fig. S1P) (Methods; Data S8–S15). This revealed that in contrast to other T-cell co-receptor genes, namely *Cd8a*, the four CD3 encoding genes, and *Cd4* to a lesser extent, *Cd8b* of NMR and DMR has evolved under significantly relaxed purifying selective pressure relative to that of murine rodents (multiple-hypotheses-adjusted p < 0.05; Fig. S1Q), suggesting it has lost some of its molecular functions.

### The NMR TCR loci encode a small diversity of α and β variable regions and a large diversity of γ and δ variable regions

NMRs have lost their NK cells and maintain a small diversity (in terms of number of genomically-encoded genes) of NK-cell receptor and *MHC-I* genes, suggesting they have evolved under relaxed intracellular pathogenic selective pressure[39]. Since CD8 αβT cells have evolved to detect cells infected with intracellular pathogens through αβTCR-recognition of MHC-I-presented peptides[3–5], we hypothesized that there would be a smaller diversity of α and β variable TCR regions in the NMR genome relative to that in other mammalian genomes. Some diversity of NMR α and β variable TCR regions should nevertheless be maintained by selection imposed by extracellular pathogens as their MHC-II-presented peptides are recognized by αβTCRs expressed on CD4 T cells[54,55]. Although the selective pressures under which γδT cells have evolved are less well-characterized, the surprisingly large proportion of these cells in NMR spleens motivated us to additionally examine the genomic diversity of γ and δ variable TCR regions.

To address these questions, we constructed a broad mammalian phyletic pattern of the number of genomically-encoded variable and constant regions for each of the four TCR loci (Figs. 2A and S2A, S2B; Data S16; Methods). This revealed that the diversities of α and β variable TCR regions in the NMR genome are -1.75- and -1.5 -fold smaller than the mean among all rodent and all mammalian genomes, respectively (Fig. 2A; Data S16). This pattern is conserved in all hystricomorph genomes, except in the guinea pig genome, which like other rodent genomes, shows an opposite pattern (Fig. 2A; Data S16). This is consistent with the phyletic pattern of NK-cell receptor and *MHC-I* genes[39], likely reflecting the fact that the guinea pig has retained its NK cells (termed Foa-Kurloff cells[56,57]) due to selective pressure imposed by intracellular pathogens. In contrast, the diversity of α and β variable TCR regions in the mouse genome is roughly similar to the mean among all rodent genomes and is -1.1-fold larger than that among all mammalian genomes, a pattern conserved in all other murine genomes (Fig. 2A, the Mymomorpha suborder; Data S16).

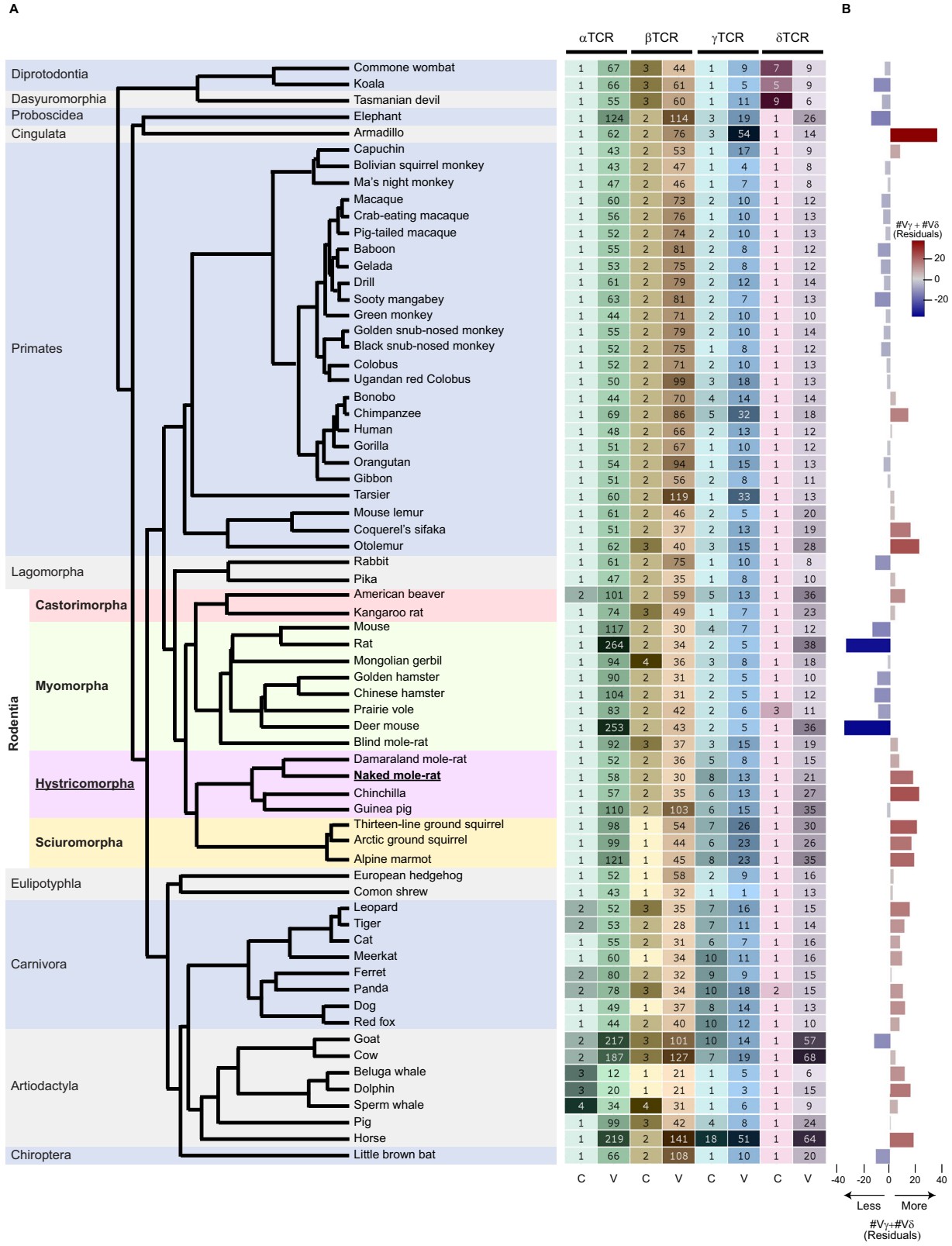

We found that the levels of diversity of the γ and δ variable TCR regions in mammalian genomes are positively associated with the levels of diversity of the α and β variable TCR regions (Fig. S2C), which may depend on the phylogenetic relationship between the represented genomes due to their shared evolutionary histories. We therefore fitted a phylogenetic least squares model (Methods) to the levels of diversity of the γ and δ variable TCR regions as a function of the levels of diversity of the α and β variable TCR regions (Fig. S2C) and used the fit residuals for expressing the levels of diversity of the γ and δ variable TCR regions. This showed that the number of NMR γ and δ variable TCR regions is ~18-fold larger than expected from its number of α and β variable TCR regions (Fig. S2B). Similar to the diversity of α and β variable TCR regions, all other hystricomorph genomes also show positive residuals, except for the guinea pig genome that shows a

**Fig. 2 | Mammalian phyletic pattern of TCR regions reveals large diversity of γ and δ, relative to α and β, variable regions in the NMR genome. A** Phyletic pattern showing the numbers of genomically-encoded constant and variable regions (annotated protein-coding and pseudo-genes, and unannotated putative pseudogenes) in each of the four TCR loci in 67 mammalian genomes (Methods), revealing the small numbers of α and β variable TCR regions yet large numbers of γ and δ variable TCR regions in the NMR genome relative to other mammalian genomes. The phylogeny is color-coded by orders, where the Rodentia order is further color-coded by suborder; (**B**) horizontal bar-chart showing the residuals of a phylogenetic least squares model fitted to the sum of γ and δ variable TCR regions

as a function of the sum of α and β variable TCR regions, accounting for the topology in the phylogeny (Methods). Negative and positive residuals are color-coded in shades of blue and red, respectively, according to their magnitude, which conveys the deviation of the sum of γ and δ variable TCR regions from what is expected based on the sum of α and β variable TCR regions. The NMR genome, as well as those of the other hystricomorphs except for the guinea pig, and several other clades show a larger diversity of γ and δ variable TCR regions than expected from the diversity of their α and β variable TCR regions. Abbreviations: C Constant, V Variable. Source data are provided as Source data file.

---

negative residual (Fig. 2B). In contrast, all murine genomes, except for the blind mole-rat (*Nannospalax galili*) genome, show negative residuals (Fig. 2B, the Mymomorpha suborder).

## NMRs harbor a dominant γδT-cell clonotype and the αβ clonotypic diversity of their CD8 T cells is smaller than that of their CD4 T cells

The phylogenetic analysis of the diversity of genomically-encoded variable TCR regions predicts that the clonotypic diversity of NMR αβT cells is smaller than of mouse αβT cells and vice versa for γδT cells. This prediction, however, is not sufficiently informative about the clonotypic diversities of the various subsets in each T-cell lineage. For example, while αβTCRs are expressed on both CD8 and CD4 T-cells, these two subsets evolved under different evolutionary selective pressures and as a result might have considerably different clonotypic diversities. The various γδT-cell subsets, both those known in mouse[48], and the NMR ones discovered here, might pose an analogous situation. Therefore, we devised a hybridization-capture approach for sequencing single-cell-barcoded full-length TCR transcripts and applied it to our NMR and mouse single-splenic-T-cell samples to profile their clonotypic diversity and how these change with age (Fig. S3A; Methods).

Our hybridization-capture approach sequenced over 99% of the cells (means across all T-cell subsets: NMR: 99.75% ± 0.05; mouse: 99.76% ± 0.08). However, in order to use cells in which both TCR chains were sequenced and mapped, we retained an average of ~39% ± 3.3 of the NMR T cells (~49% ± 2.3 of αβ subsets and ~38% ± 3.4 of the γδ subsets) and an average of ~36% ± 3.5 of the mouse T cells (~36% ± 3.5 of αβ subsets and ~50% ± 10 of the γδT subset), (NMR: Fig. S3B–S3D; Data S17; mouse: Fig. S3E–S3G; Data S18; Methods). Subsequently, we defined a T-cell clonotype by the constant, variable, diversity, and joining TCR gene IDs, and the amino-acid sequence of the complementary determining region (CDR) 3 of both of its TCR chains.

The NMR γδT-cell clonotypes show a publicity by which ~1.4% of them are observed in more than a single sample (Fig. 3A, B; Data S17), yet the cytotoxic γδT-cell clonotypes are strikingly more public (5.32%) than the non-cytotoxic γδT-cell clonotypes (0.4%). Intriguingly, the cytotoxic γδT-cell subset harbors a highly dominant and public clonotype (Cγ4, Vγ4-2, Jγ5-3, CDR3: TYWDSNYAKK; Vδ1-4, Dδ3, Jδ2, CDR3: ALWELRTGGITAQLV), represented in all samples in the range of 15.6% −65.1% of their cells (mean 34% ± 6.7), where several other nearly identical clonotypes, contribute even more to this dominance and publicity: 26.8%−81.4% of the cells across samples (mean 34% ± 7.8) (clustered together in the top left of Fig. 3B). Phylogenetic reconstruction of the mammalian Vγ TCR regions (Methods) places the NMR Vγ4-2 together with the mouse Vγ1, Vγ2, Vγ3, and Vγ4 (Fig. S3H). Mouse γδT cells utilizing Vγ1 and Vγ4 are known to arise relatively late in fetal development and even during neonatal and adult life[58]. They reportedly express a diverse set of TCRs due to pairing with several Vδ TCR regions, are heterogeneous in their capacity to produce various effector cytokines, and are found in many tissues such as peripheral lymphoid organs, blood, lung, liver, and dermis[58]. In contrast, mouse γδT cells utilizing Vγ5 and Vγ6 arise earlier in fetal development and

populate various epithelial tissues[58]. The NMR Vγ4-2 is also placed together with the conserved primate group-2 Vγ9, Vγ10, and Vγ11, which in humans are also utilized by the major circulating γδT subsets[59]. By contrast, NMR αβT-cell clonotypes are much more private and no dominant clonotypes were observed (Fig. 3A, C; Data S17).

In order to compare clonotypic diversities, both between old- and adult-age samples of the same T-cell subset, as well as between different T-cell subsets of the same age both within and between species, we used three metrics: (1) Hill number, which quantifies clonotypic richness regardless of abundance; (2) Shannon entropy, which weighs clonotypes by their abundances and therefore quantifies clonotypic evenness; and (3) Gini-Simpson index, which gives more weight to rare clonotypes[60]. Among all NMR T-cell subsets, only the cytotoxic γδT-cell clonotypes show a significantly increased evenness and rarity with age (multiple-hypotheses-adjusted $p < 0.05$; Fig. 3D, E). The cytotoxic γδT-cell clonotypes also show significantly smaller evenness and rarity relative to the non-cytotoxic γδT-cell clonotypes in old-age samples (multiple-hypotheses-adjusted $p < 0.05$; Fig. 3D, E), as expected given the dominant cytotoxic γδT-cell clonotypes (Fig. 3D). Interestingly, CD8 αβT-cell clonotypes are significantly less diverse, by all metrics, than CD4 αβT-cell clonotypes in the old-age samples (multiple-hypotheses-adjusted $p < 0.05$; Fig. 3D, E).

The mouse γδT-cell clonotypes show a slightly higher publicity than either of the NMR γδT-cell clonotypes (4%), and also harbor a dominant and public clonotype (Cγ1, Vγ6, Jγ1, CDR3: ACWDSSGFHKV; Vδ4, Dδ2, Jδ2, CDR3: GSDIGGSSWDTRQMF), represented in five of the eight samples in the range of 43.8%–100% of their cells (mean 64% ± 9.8) (Fig. 3F, G; Data S18), albeit the sample sizes in the two species are drastically different (~4500 clonotyped NMR γδT cells versus 47 clonotyped mouse γδT cells). Compared to NMR αβT-cell clonotypes, mouse αβT-cell clonotypes show a slightly larger publicity (~1.3% and 0.9% of the memory CD4 and CD8 αβT-cell clonotypes are observed in more than a single sample, respectively) (Fig. 3F, G; Data S18). Both the naive CD8 and CD4 T-cell clonotypes show an age decline in diversity (multiple-hypotheses-adjusted $p < 0.15$ for all diversity metrics), as expected[51,61,62], whereas Treg clonotypes show the opposite trend (Fig. 3H, J). The only striking T-cell-subset differences in mouse clonotypic richness and evenness are observed between the old-age *Gzmk*-high memory CD8 T cells relative to memory CD8 T cells (multiple-hypotheses-adjusted $p < 0.09$; Fig. 3H, J), as expected[51].

Comparing the clonotypic diversities between the two species, NMR non-cytotoxic γδT-cell clonotypes show a larger diversity than that of mouse γδT-cell clonotypes, significantly so for evenness and rareness among the old-age samples (multiple-hypotheses-adjusted $p < 0.05$; Fig. 3K). In stark contrast, no interspecies diversity differences were evident between NMR cytotoxic γδT-cell clonotypes and mouse γδT-cell clonotypes (Fig. 3E). Among the αβT-cell subsets, the NMR CD8 T-cell clonotypes show a smaller diversity than that of the mouse naïve CD8 T-cell clonotypes, and more strongly so among the adult-age samples (multiple-hypotheses-adjusted $p < 0.13$; Fig. 3K). By contrast, NMR CD4 T-cell clonotypes are generally more diverse than mouse naïve and memory CD4 T-cell clonotypes, only among the old-age samples (multiple-hypotheses-adjusted $p < 0.29$; Fig. 3K).

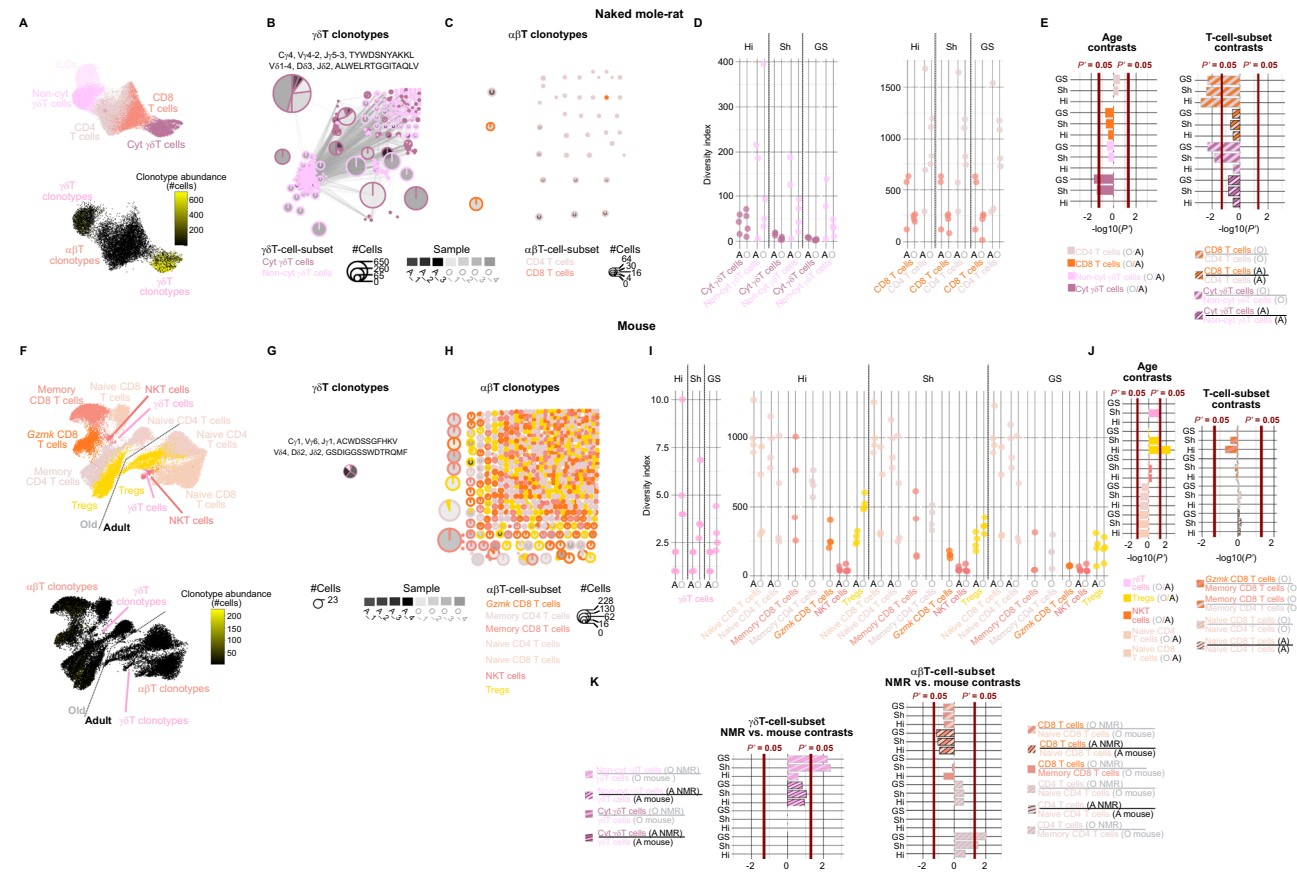

**Fig. 3 | Single-cell TCR sequencing reveals surprising features of clonotypic diversity in NMR T-cell subsets. A** UMAP projections of splenic T cells from three adult and four old NMRs. Each point represents a cell, the top UMAP is color-coded by T-cell subset and the bottom UMAP is color-coded by clonotype abundance; (**B**) NMR γδT-cell and (**C**) αβT-cell clonotype network graphs. Pies correspond to clonotype clusters, their slices correspond to samples, their lines color-coded by T-cell subsets, and their areas correspond to clonotype abundances (only showing clonotypes with three cells or more). The dominant γδT-cell clonotype is shown in Fig. 3C; (**D**) NMR T-cell clonotypic Hill, Shannon, and Gini-Simpson diversities (Methods). Each point is a sample. Larger diversities are observed in the non-cytotoxic relative to the cytotoxic γδT-cell subsets and in the CD4 relative to the CD8 αβT-cell subsets; (**E**) multiple-hypotheses-adjusted −log$_{10}(p)$ contrasting clonotypic diversities (mcpHill function; Methods), between old and adult samples per each T-cell subset (left), and between different T-cell subsets per each age group (right), showing the larger diversities in the non-cytotoxic relative to the cytotoxic

γδT-cell subsets and in the CD4 relative to the CD8 αβT-cell subsets; (**F–J**) are analogous to Fig. 3A–E for the splenic T cells from four adult and four old mice. Unlike in NMRs, mouse CD4 and CD8 αβT-cell clonotypic diversities are not significantly different; however, the *Gzmk*-high CD8 T-cell subset is enriched with rare clonotypes, likely on account of the old-age memory CD8 T-cell subset, from which it is thought to be derived; (**K**) multiple-hypotheses-adjusted −log$_{10}(p)$ contrasting the clonotypic diversities (mcpHill function; Methods), between NMR and mouse γδ (left) and αβ (right) T-cell subsets per each age group, showing the larger clonotypic diversity in the NMR non-cytotoxic γδT-cell subset relative to that of the mouse γδT-cell subset, the smaller clonotypic diversity in the NMR CD8 αβT-cell subset relative to that of the mouse CD8 αβT-cell subsets, and the larger clonotypic diversity in the old NMR CD4 αβT-cell subset relative to that of the old mouse CD4 αβT-cell subsets. Abbreviations: A Adult, CT Clonotype, Cyt Cytotoxic, NKT Natural Killer T, Non-cyt Non-cytotoxic, O Old, P' multiple-hypotheses-adjusted *P* value. Source data are provided as Source data file.

## NMR CD8 and CD4 αβT-cell clonotypic diversities, *MHC-I* and *MHC-II* gene-family sizes, and thymic involution

Although NMRs and mice maintain roughly equal proportions of T cells among their splenic immune-cell populations (NMRs 26% and mice 27.2%, Fig. 1A, B, respectively), their CD8 and CD4 T-cell subsets show disparate proportions. In adult NMRs, CD8 T cells comprise 7% whereas their CD4-T-cell population is ~2.5-fold larger (17%), becoming even more CD4-biased in old NMRs (2.3% CD8 and 11% CD4) (Figs. 1D and S1D). By contrast, naive CD8 and naive CD4 T cells comprise 13% and 12% in adult mice, respectively, and in old mice both naive and memory CD8 and CD4 T cells are at similar proportions (4% for both naive subsets and 6% for both memory subsets) (Fig. 1E). This is consistent with the smaller clonotypic diversity of CD8 versus CD4 T cells in NMR spleens (Fig. 3D, E), in contrast to the equivalent clonotypic diversities of CD8 and CD4 T cells in mouse spleens (Fig. 3H, J). It is also consistent with the smaller clonotypic diversity in NMR splenic CD8 T cells compared to those of mouse splenic CD8 T cells

(Fig. 3K). Assuming that the NMR versus mouse differential CD8 versus CD4 T-cell proportions and clonotypic diversities reflect differential intracellular versus extracellular pathogenic selective pressures under which these species have evolved, we anticipate a corresponding differential *MHC-I* versus *MHC-II* gene-family sizes in their genomes, since the two *MHC* gene families are thought to have expanded in response to these two pathogenic selective pressures, respectively[13,63]. To address this hypothesis, we generated a mammalian phyletic pattern of the *MHC-I* and *MHC-II* gene families (Methods). Consistent with our previous findings[39] and with the phyletic pattern of α and β variable TCR regions (Fig. 2A, B), the size of the *MHC-I* gene family in the NMR genome is ~11- and ~10 -fold smaller than its mean among all rodent and mammalian genomes, respectively (Fig. 4A; Data S19). By contrast, the size of the NMR *MHC-II* gene family is only ~1.6- and ~2 -fold smaller than its mean among all rodent and mammalian genomes, respectively (Fig. 4A; Data S19). Accordingly, the ratio of the NMR *MHC-I/MHC-II* gene-family sizes is ~7.8- and ~5.3-fold smaller than its mean among all

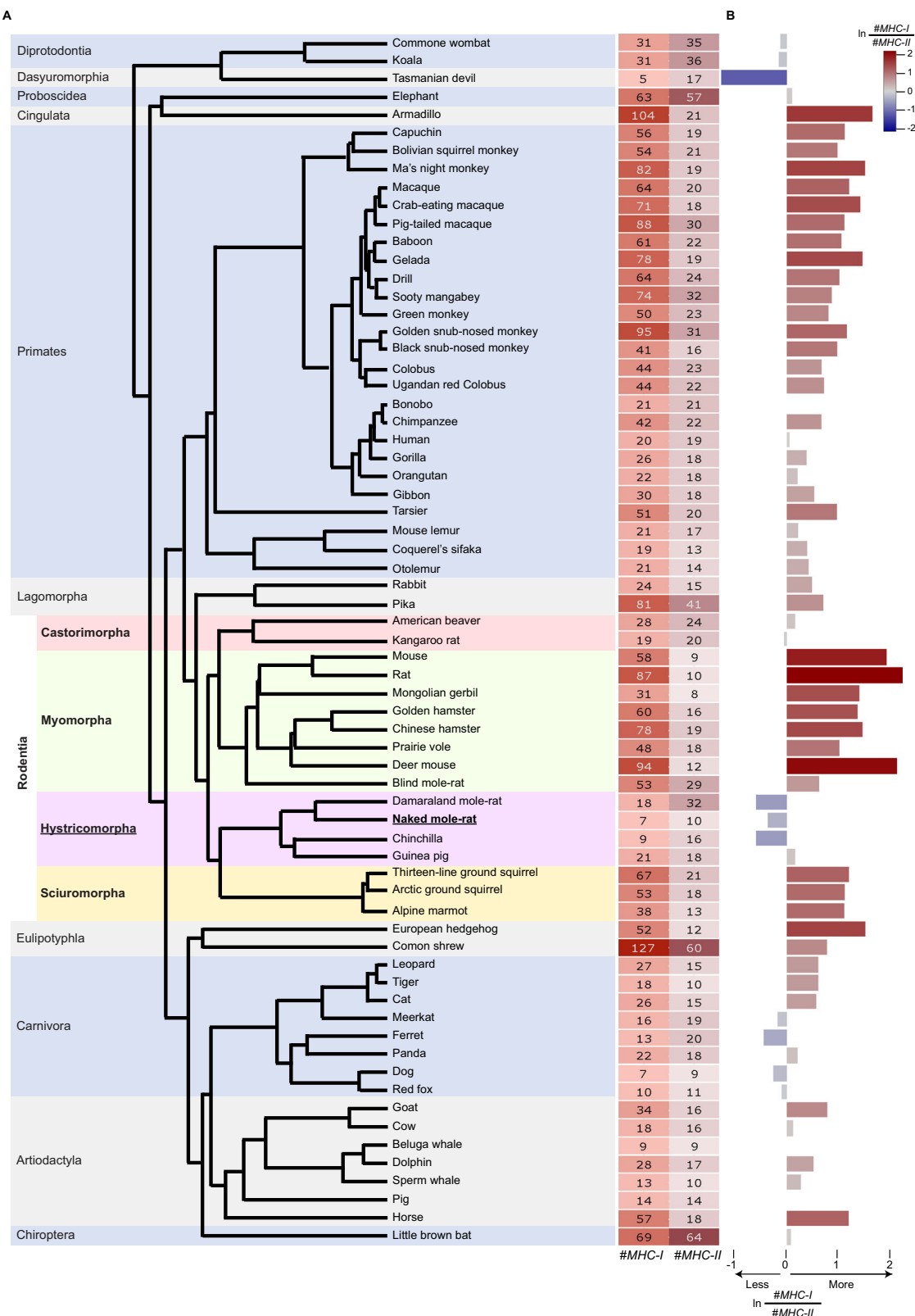

**Fig. 4 | Mammalian phyletic pattern of *MHC* gene families reveals a small *MHC-I*, relative to *MHC-II*, gene-family size in the NMR genome. A** Phyletic pattern showing the numbers of genomically-encoded *MHC-I* and *MHC-II* annotated protein-coding and pseudo -genes, and unannotated putative pseudogenes, in 67 mammalian genomes (Methods), revealing the small numbers of *MHC-I* genes in the NMR genome relative to other mammalian genomes. The phylogeny is color-coded by orders, where the Rodentia order is further color-coded by suborder; (**B**) horizontal bar-chart showing the natural log (ln) ratio of the number of *MHC-I* genes scaled to the number of *MHC-II* genes. Negative and positive ratios are color-coded in shades of blue and red, respectively, according to their magnitude, emphasizing the low ratios in the genomes of NMR and the other hystricomorphs except for the guinea pig. Source data are provided as Source data file.

rodent and mammalian genomes, respectively (Fig. 4B). Similar to α and β variable TCR regions (Fig. 2B), this pattern is conserved among the hystricomorph genomes, except for the guinea pig genome, which similar to all other rodent genomes, shows the opposite pattern (Fig. 4B).

Since CD8 αβT-cell clonotypic diversity is smaller in NMRs relative to mice, especially so among adult-age samples (Fig. 3K), we hypothesized that early in life NMRs require less thymic tissue than mice. Although NMRs have a large proportion of splenic γδT cells, the clonotypic diversity of the NMR γδT cells is dramatically smaller than that of their αβT cells (Fig. 3D), and thus might have a negligible impact on NMR thymic mass. Given that thymic involution, which manifests in a sharp decrease in mass and changes in tissue architecture, is a hallmark of the age trajectory in mice[64], we compared NMR and mouse thymic weights and histology at physiologically equivalent early life, adulthood, and middle age time points (mean NMR $n = 3.2$ females and $n = 2.4$ males per time point combining the thoracic and cervical tissues in each animal; mean mouse $n = 4.5$ females and $n = 4.7$ males per time point; Data S20). The observed age trajectory of the mouse thymic weight confirms previous reports[65–68]: the thymus increases in weight up to one month of age and sharply diminishes thereafter despite the increase in body weight. As such, by early adulthood (at six months of age) the mouse thymus is less than half of its maximal weight (Fig. 5A–C). The NMR thymus follows a relatively similar age trajectory to that of the mouse thymus with two key differences: (1) its peak weight occurs at the early-life age of two months but is roughly half the maximal weight of the mouse thymus, both in absolute weight and relative to body weight; (2) its major weight decline, which occurs between two and twelve months, is not as steep as that of the mouse thymus, which occurs at roughly equivalent physiological ages, between one and two months (Fig. 5A–C). Histologically, young adult-age mouse and NMR thymuses do not yet show microscopic signs indicative of involution (at six and sixty months, respectively; Fig. 5D). However, such signs, including less corticomedullary distinction, a reduction in cortical tissue (comprised predominantly of lymphocytes), irregular margins with increased peripheral fat and adipocyte infiltration, and formation of cysts[69,70], are visible in both species from early middle-age onwards (as of 12 and 120 months, respectively; Fig. 5D). In NMRs, these microscopic signs indicative of involution are apparent in both the thoracic and cervical thymic tissues (Fig. S5). In other words, both species show microscopic signs of involution progression at physiologically equivalent ages. By counting thymocytes, Emmrich et al. reported that the NMR thymus increases in cellularity between 30 and 150 months with no concomitant reduction in thymic mass[43]. Given our observations, obtained using the gold-standard histology approach over a wider age range, it is difficult to see how NMR thymic involution only starts after 150 months of age.

## Discussion

Our investigation of NMR T cells has uncovered several intriguing findings. NMRs have a splenic population of γδT cells at roughly the same proportion as that of mouse splenic NK cells, comprising two subsets. One of these γδT-cell subsets expresses an inhibited cytotoxic molecular profile homologous to that of mouse splenic NK cells, suggesting the two cell types are functionally similar yet are likely to differ in their activation mechanisms. Mouse γδT cells mostly reside in epithelial tissues[17,71] and the thymus is the only tissue where we observe these cells to express high levels of *Itgae*, whose encoded CD103 integrin binds to epithelial cadherin[53]. If these *Itgae*-high mouse γδT cells are indeed homed to epithelial tissues as they exit the thymus[53], the absence of equivalent *Itgae* expression patterns in NMR γδT cells in the thymus may explain their large proportion in the circulation and hence spleen. Compared to other mammalian genomes, the NMR genome, like that of other hystricomorphs, with the exception of the guinea pig, has a considerably larger diversity of γ and δ

variable TCR regions relative to that expected from its small diversity of α and β variable TCR regions. This may have evolved to generate a large clonotypic diversity of circulating γδT cells capable of recognizing a diverse spectrum of non-MHC-I ligands. In contrast, the small genomic diversity of NMR α and β variable TCR regions, the smaller proportion of the CD8 relative to the CD4 αβT-cell subset in the NMR spleen along with a corresponding bias in the clonotypic diversities of these NMR αβT-cell subsets, and the smaller size of the NMR *MHC-I* relative to its *MHC-II* gene family, corroborate the loss of NMR NK cells[39]. These observations support the hypothesis that this occurred as a result of relaxed selective pressure imposed by intracellular pathogens[39]. Although intracellular pathogens are thought to be one of the most dominant selective forces in mammalian evolution[72], the subterranean ecological niche that NMRs occupy in isolated colonies, is likely an evolutionary dead end for mammalian intracellular pathogens due to its limiting effect on spread of infections. This contrasts with bats, which occupy a much more exposed and richer ecological niche and harbor more virus species than any other mammal[73]. Thus, the selective pressure operating on NMRs might have shifted from defense against intracellular pathogens towards elimination of early malignancies, maintenance of tissue homeostasis, and perhaps defense against various extracellular pathogens, to which their immune system has responded by utilizing γδT cells. It is tempting to speculate that the dominant and public clonotype among the cytotoxic NMR γδT cells recognizes frequently encountered ligands that signal a threat of strong selective magnitude, which therefore must be rapidly eliminated.

The absence of *Cd8b* expression in NMR *Cd8a*-expressing T cells outside the thymus and the relaxed purifying selective pressure operating on the NMR *Cd8b* gene, suggest that the cytotoxic activity of NMR *Cd8a*-expressing T cells is regulated differently than of human and mouse CD8 αβT cells[74–77]. In these latter species the CD8α-CD8β heterodimer interacts with the monomorphic part of MHC-I thereby facilitating tight αβTCR-MHC-I binding and subsequent CD8-T-cell activation. Conversely, human and mouse CD8αα homodimers decrease TCR sensitivity to MHC-I[78], but can also facilitate cytotoxic-T-cell activation by interacting with non-classical MHC-I ligands[79]. Human and mouse CD8αα T cells populate the gut as γδT cells[80] and the skin as αβT cells, and even comprise tumor-infiltrating innate-like T cells with high cytotoxic potential (ILTCK) as αβT cells[81]. It remains to be established whether these human and mouse CD8αα T-cell subsets are homologs of the NMR *Cd8a*-expressing T-cell subsets.

Compared to the mouse genome the NMR genome has a larger diversity of γ and δ variable TCR regions. However, the absolute diversity of α and β variable TCR regions in the mouse genome is substantially larger than in the NMR genome, likely reflective of the stronger intracellular pathogenic selective pressure that mice have evolved under. This translates to a much larger potential diversity of mouse thymocyte clonotypes, which need to pass the strict selection imposed by thymic epithelial cells to ensure they are released into the circulation as functional and self-tolerant T cells, a process especially critical in early postnatal life when the various immune system compartments need to be populated[42,82]. Based on the much larger potential diversity of mouse thymocyte clonotypes, the strikingly disparate early-life weights and subsequent rates of decline between the age trajectories of mouse and NMR thymuses become intuitive. One of the theories explaining thymic involution suggests that once a T-cell clonotypic repertoire is established the metabolically costly process of thymopoiesis is turned off in order to divert energy to other physiological processes[42,83]. This reasoning can explain the thymus age trajectories we observed in this study: the larger perinatal thymuses of mice are required for generating their larger initial 'pulse' of T cells, with their overall larger clonotypic diversity and a subsequent steeper decline, yet by their equivalent middle age, thymuses of both species show signs of involution.

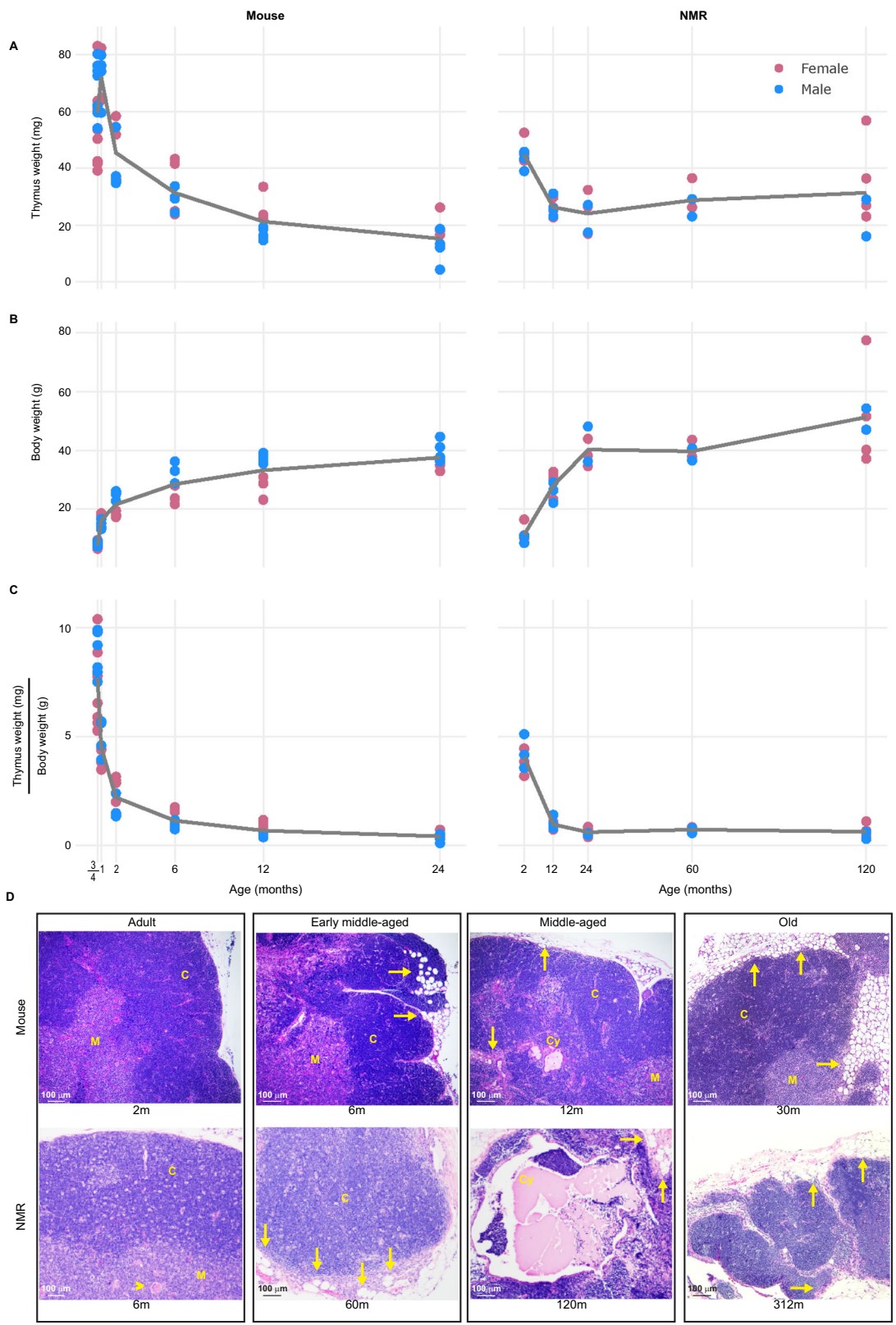

Notwithstanding our findings our study has several limitations. Although the evidence for relaxed selective pressure imposed by intracellular pathogens under which NMRs have evolved is strong, data regarding NMR susceptibility to such insults is limited[84,85] and challenging to generate because it is unclear which intracellular pathogens might be able to infect NMR cells. In addition, despite the surprisingly large fraction of splenic γδT cells in NMRs, and especially the subset with the inhibited cytotoxic transcriptional profile that resembles mouse splenic NK cells, we do not offer direct evidence regarding their effector functions. Hence whether and how these splenic γδT cells contribute to any of the unique NMR features, namely cancer resistance, remains to be investigated. A possible follow-up study to that end would involve single-cell and TCR profiling combined with spatial transcriptomics in NMR tumor microenvironments to test for the

**Fig. 5 | Age trajectories of mouse and NMR thymuses reveal that NMRs develop significantly smaller thymuses early in life but follow a similar trajectory to mouse thymuses, accumulating similar microscopic signs of involution.** Age trajectories of (**A**) thymic weights (mg), (**B**) corresponding body weights (g), and (**C**) thymic weights (mg) scaled to body weights (g) from physiologically-equivalently-aged mice (mean $n$ per time point = 9.16; left) and NMRs (mean $n$ per time point = 5.6; right), where in each NMR the thymus weight is the sum of its thoracic and cervical thymic tissue weights. The trend lines in each of the figures are values obtained from fitting the polynomial regression: $\sim age + sex$ to the y-axis values, emphasizing the much smaller NMR thymic mass compared to the mouse thymic mass in early life; (**D**) images of hematoxylin and eosin (H&E) stained sections from the thoracic thymuses of physiologically-equivalently-aged young adult (left), middle-aged (center), and old (right) mouse and NMR. Microscopic signs indicative of involution are not apparent in young adults of both species, yet from middle-age onwards thymuses of both species show these signs, including less corticomedullary distinction, a reduction in cortical tissue, irregular margins with increased peripheral fat and adipocyte (fat) infiltration (arrows), and cysts. Abbreviations: C Cortex, Cy Cyst, M Medulla. Source data are provided as Source data file.

presence of expanded NMR γδT-cell clonotypes which engage in cancer-cell killing. This obviously depends on the ability to transform NMR cells and have them form tumors in an NMR host yet might be feasible as the former has already been demonstrated[35]. It should be noted that a large proportion of circulating γδT cells and large genomic diversities of γ and δ variable TCR regions is not unique to NMRs as these are also found in ruminants and equines[86–88]. Whether γδT cells in these species relate to cancer resistance, maintenance of homeostasis, and lifespan is also unknown and compounded by the possibility that these traits may have been affected by the domestication of these farm animals[89,90]. In humans, however, larger proportions of γδT cells in a diverse set of tumor types were found to be significantly associated with a favorable cancer prognosis[90]. Finally, phyletic patterns, considerably utilized in this work, are sensitive to the integrity levels of the genome assemblies. Although we limited our data to genomes assembled at the scaffold and chromosome levels and benchmarked our approach using the well assembled and annotated human and mouse genomes, the gene-family sizes we obtained are likely only an approximation of their true magnitudes.

The hybridization-capture approach we introduced in this work allows studying clonotypic dynamics of any T-cell lineage, in any species, unlike the 10× V(D)J kit, which is limited to human and mouse αβT cells. Moreover, our approach is relevant to any target genes of interest and can be applied to previously generated and currently stored 3′-barcoded scRNA-seq samples, which should be abundant as only ~25% of them are required for short-read sequencing. Although we obtained paired αβ and γδ TCR sequences in only a ~1/3 of the profiled αβ and γδ T cells, likely due to suboptimal capture-probe hybridization efficiencies and insufficient sequencing depth, using many more cells by defining γδ and αβ T-cell clonotypes by their γ and β chains, respectively, does not change our results and conclusions.

In summary, our work provides a detailed view of the T-cell subsets of the NMR immune system in terms of their composition, genome evolution, development, and clonotypic diversity. This adds another steppingstone in the path towards deciphering the blueprint of the remarkable longevity and cancer resistance of the NMR and might provide invaluable contributions for developing immune-based therapeutics.

## Methods

### Ethics statement
All animal use and experiments were approved by the Buck Institute institutional animal care and use committee (IACUC) protocol number A10173.

### Animals
In this study, we used comparative histomorphology, comparative genomics, and single-cell transcriptomics to study the immune system of the NMR in comparison with that of the mouse. The complete list of animals can be found in Data S21. In brief, the animals used for scRNA-seq in this study comprised:

1. Sixteen C57BL/6 mice, purchased from Jackson Laboratories (Bar Harbor, ME), JAX stock #000664, ($n$ = 4 females and $n$ = 8 males at 2 months of age, referred to as "adult" throughout the text and figures; $n$ = 4 males at 24 months of age, referred to as "old" throughout the text and figures; all virgins), housed within the Laboratory Animal Resources (LAR) vivarium, which is part of the AAALAC-accredited animal care and use program at the Calico Life Sciences LLC, at a room temperature range of 14.5−26 °C with a humidity range of 30−70%, maintained on a 12-house dark-light cycle, receiving food and water *ad libitum*, and used in experiments only after two weeks of vivarium housing.

2. Nineteen NMRs ($n$ = 6 females and $n$ = 9 males 2 years of age, referred to as "adult" throughout the text and figures; $n$ = 4 males at 26−28 years of age, referred to as "old" throughout the text and figures; all non-breeding virgins), from 20 different captive colonies housed within Calico Life Sciences colonies at the Buck Institute, Novato, CA, at a room temperature range of 28−31 °C with a humidity range of 40−50%, maintained on a 12-house dark-light cycle, receiving food *ad libitum* yet no supplemented water since the water content of their fresh fruit and vegetable diet is sufficient for maintaining appropriate hydration, in accordance with standard colony management.

Figure 1A−C and S1A-S1C are based on data from our previous study[39]. The animals used for the thymus age trajectory study (Fig. 5A−C) comprised 55 C57BL/6 mice, purchased from Jackson Laboratories (Bar Harbor, ME), JAX stock #000664, ($n$ = 27 females and $n$ = 28 males, both sexes at an age range of 1−24 months) and 28 NMRs ($n$ = 16 females and $n$ = 12 males, both sexes at an age range of 1−120 months), housed in the same conditions as detailed above. Thymuses from all mice at 2, 6, and 12 months of age, as well as from thirteen mice at 30 months of age ($n$ = 8 females and $n$ = 5 males), and thymuses of all NMRs at 24, 60, and 120 months of age, as well as from four NMRs at 6 months of age ($n$ = 2 females and $n$ = 2 males) and two NMRs at 312 months of age (both males), were used for histological examination (Figs. 5D and S5) The age range we chose for comparing adult versus old animals was targeted to allow use of healthy individuals that were physiologically age-matched between the species (approximately 5−10% and 70−80% of observed maximum life spans in adult and old animals, respectively). The age range for the thymus age trajectory study was determined based on previous studies in mice[91,92].

### Organ collection and processing
Organ collections for sequencing purposes were performed on all animals between 8 AM and 10 AM. Animals were euthanized using isoflurane followed by thoracotomy and cardiac exsanguination. Details regarding the dissociation of each of the organs to single-cell suspensions, for the purpose of constructing 10× GEM single-cell libraries are detailed below. For the thymus age trajectory study, we characterized the thymuses of the NMRs and mice by their weight and histology (for NMRs both thoracic and cervical tissues were included for each animal). Thymuses were immersion fixed in neutral buffered formalin, and were routinely processed into paraffin-embedded blocks, section to 5 μm and stained with hematoxylin and eosin.

Histological assessment of these thymic sections was performed by a certified veterinary pathologist.

## Spleen dissociation

Spleens were removed by dissection, transferred to a sterile petri dish containing PBS with 5% FBS, and minced using surgical blades. Spleen fragments were then ground through 100 µm and 40 µm cell strainers, pelleted, and resuspended in the ACK lysis buffer as described in the thymus tissue collection section. Cell density and viability were determined using Countness II Cell Counter. Subsequently, spleno-cytes were resuspended at 1 million cells per ml of 0.04% BSA/PBS. In order to sort out the splenocyte T-cell subsets for constructing 10× GEM single-cell libraries, splenocytes were resuspended at 1 million cells per ml of FACS buffer (PBS with 1% FBS and 5mM EDTA). Subse-quently, mouse splenocytes were pre-incubated with 1 µg mouse Fc blocker (BD 553141) per 1 million cells/100 µl FACS buffer for 5 min and incubated with an anti-mouse CD3e (clone 17A2, eFlour450, 2 µg/ml) and an anti-mouse CD11b (M1/70, Alexa 488, 2 µg/ml) on ice for 40 min. NMR splenocytes were pre-incubated with the isotype control antibodies (100 µg/ml mIgG1 & 50 µg/ml rat IgG1) per 1 million cells/ 100 µl FACS buffer for 10 min. Following that, cells were washed and resuspended at 2 million cells per 1 ml FACS buffer and incubated on ice for 40 min with an anti-NMR CD3e (Abbvie-clone-5, Alexa 647, 1:10, hence -10 µg/ml) and the anti-mouse CD11b (M1/70, Alexa 488, R-PE, 5 µg/ml), which cross reacts with the NMR CD11b[93]. The anti-NMR CD3e antibody was validated in-house using a series of dilutions of the fluorescence-conjugated antibody and measuring the resulting per-centage of CD11b-/CD3e+ sorted cells. For both mouse and NMR, after incubation, cells were washed and resuspended in the FACS buffer containing Sytox Blue Live/Dead dye for cell sorting using FACS BD Aria. Viable, single cells were gated, and the CD11b-/CD3e+ cells were collected for the generation of 10× GEM single-T-cell libraries.

## Bone-marrow dissociation

Femurs and tibias of the euthanized animals were dissected using sterile surgical techniques and transferred to a petri dish containing the basal medium (Minimal Essential Medium supplemented with 10% FBS and 1X antibiotics antimycotic solution (Gibco)). The dorsal and distal ends of femurs and tibias were cut with scissors to reveal the bone marrow cavity, and the bone marrow cells were flushed out with the basal medium using needles/syringes. Cells were then passed through 100 µm and 40 µm cell strainers, washed, pelleted, and resuspended in 5 mL of ACK lysis buffer (Lonza BP10-548E, Basel, Switzerland) for 5 min at room temperature. After determining their density and viability, cells were resuspended at 1 million cells per ml of 0.04% BSA/PBS prior to generating the 10× GEM single-cell libraries.

## Thymus dissociation

The thoracic and cervical thymic tissues were dissected using sterile surgical techniques and subsequently transferred to a petri dish con-taining PBS. Fat tissue surrounding the thymic parenchyma was care-fully removed with microsciscors and tissue forceps under a surgical microscope. Excess liquid/PBS was carefully removed with Kimwipes before weight measurement. For histological analysis, the tissues were fixed in 10% formalin and following standard tissue processing pro-cedures. For the purpose of creating single-cell suspensions, thymic tissues were ground through 100 µm and 40 µm cell strainers (Falcon 352360 and 352340, Corning, NY) with a syringe plunger. Cells were subsequently pelleted (300 g, 3 min, 4 °C), resuspended in 5 mL of ACK lysis buffer (Lonza BP10-548E, Basel, Switzerland) for 5 min at room temperature, and washed and resuspended with PBS with 0.04% BSA. Cell density and viability were determined using the Countness II Cell Counter (Thermofisher AMQAF1000). The cells were pelleted and resuspended at 1 million cells per ml of 0.04% BSA/PBS prior gen-erating the 10× GEM single-cell libraries.

## ScRNA-seq data and analysis

Single cells were captured in droplet emulsion using the Chromium Controller (10× Genomics, Pleasanton, CA), and GEM single-cell libraries were constructed according to the 10× Genomics protocol using the Chromium Single-Cell 30 Gel Bead and Library 3′ V3 kit (10× Genomics, Pleasanton, CA). In brief, cell suspensions were diluted in PBS with 0.04% BSA to a final concentration of $1 \times 10^6$ cells/mL (1000 cells per µL). Cells were loaded in each channel with a target output of 10,000 cells per sample. All reactions were performed in a C1000 Touch Thermal Cycler (Bio-Rad Laboratories, Hercules, CA) with a 96 Deep Well Reaction Module. Twelve cycles were used for cDNA amplification and sample index PCR. Amplified cDNA and final libraries were evaluated using a Bioanalyzer 2100 (Agilent Technologies, Santa Clara, CA) with a high sensitivity chip.

ScRNA-seq fastq files were demultiplexed to their respective barcodes using the 10× Genomics Cell Ranger "mkfastq" utility. Unique molecular identifier (UMI) counts were generated for each barcode using the Cell Ranger "count" utility. The mm10 reference genome and the mouse Gencode vM25 primary assembly annotation were used for mapping the mouse reads[94]. The HetGla2.0 reference genome along with the combined RefSeq[95] GCF_000247695.1 and Ensembl[96] 101 release Heterocephalus_glaber_female 1.0 annotations (see Phyletic patterns section below for further details on how genome annotations were combined), were used for mapping the NMR reads. Similar to our previous work[39], for each sample, barcodes that were not likely to represent captured cells were filtered out by detecting the first local minimum above 2 in a distribution of log10(#UMIs). Similarly, for each sample, genes that were too sparsely captured across barcodes were filtered out by detecting the first local minimum above 3 in a dis-tribution of log10(#barcodes). Finally, barcodes capturing more than a single cell (multiplets) were sought as local modes in the distributions of log10(#genes) and log10(#UMIs), whose x-axis maxima are more than 1.5-fold larger than the x-axis location of the global maximum of the respective distribution and include less than 5% of the total number of barcodes. In other words, barcodes that, based on the number of genes or UMIs they captured, appeared inflated with respect to all other barcodes, were regarded as multiplets and thus filtered out.

To identify transcriptionally-defined cell clusters, we followed the same steps as performed in our previous work[39]. Samples from each species were first concatenated, UMIs were then scaled to the read depth of their respective barcodes, then multiplied by a scaling factor of $1 \times 10^4$, added a constant of 1, and finally natural log transformed. Following that, genes with high expression dispersion were obtained using the "FindVariableGenes" function implemented in the R[97] Seurat v2.3.3 package[98]. Subsequently, principal components analysis (PCA) was performed on these variable-gene-by-cell–scaled UMI matrices using the R[97] rsvd package[99] in order to reduce the gene dimension, retaining the 50 PCs explaining the largest amount of variation. We then used Seurat's methodology[98] to build a shared nearest neighbor (SNN) graph of these cell-embedding data, first generating a $k$-nearest neighbor ($k$NN) graph using k = min(750, #cells-1) and a Jaccard dis-tance cutoff of 1/15. The SNN graph was then used as input to the Louvain algorithm, implemented in the ModularityOptimizer software[100]. We searched the 0.05–1.225 range of the resolution para-meter implemented in this software, for the value maximizing the mean unifiability isolability clustering metric. This process was initially done on all cells in our data and subsequently repeated for each cluster individually, in an iterative manner in which convergence was defined as not being able to break down a cluster into sub-clusters.

In order to assign cell-subset identities to the transcriptionally defined clusters, we used gene markers which we derived for each cluster both computationally, using the R[97] singleCellHaystack package[101] applied to the variable-gene-by-cell–scaled UMI matrix, as well as manually, using genes which are known to mark mouse immune-cell subsets (cell barcode to cell-type subset maps for NMR

and mouse, respectively: spleen: Data S2 and S3; bone marrow: Data S4 and S5; thymus: Data S6 and S7).

In the thymus dataset, in addition to T cells, thymic APCs were captured and sequenced yet for the purpose of focusing on T cells were omitted from Fig. 1J–L and S1J–S1L. Maturation trajectories of NMR and mouse thymocytes (Fig. 1J, K, respectively) were drawn based on biological knowledge since neither the R[97] Slingshot package[102] nor the Python[103] Scanpy[104] diffMap[101,105] and PAGA[105] methods produced trajectories that fit this well-established biological knowledge as it is laid out in the UMAP embedding space.

## Estimating effects on cell-type proportions

In order to estimate the effect of age on cell-type proportions in the spleen T-cell datasets, we fitted the mblogit multinomial logit random effects model, implemented in the R[97] mclogit package[106], to the cell-type labels, expressing age as a fixed effect and sample (animal) as a random effect (i.e., cell-type ~ age + 1|sample). We created an artificial baseline cell-type category which is the mean across all ages and samples, hence estimating the age effect on cell-type count relative to the mean cell-type count per each age and sample.

## Phyletic patterns

In order to obtain counts of the α, β, γ, and δ constant and variable TCR regions, and of the *MHC-I* and *MHC-II* gene families, across a large and representative sample of mammalian genomes, we applied the following procedure:

## Merging Ensembl and RefSeq genome assemblies and annotation.

We first downloaded the genome assemblies and annotations of 67 mammalian species from the Ensembl database[96], release 101, except for mouse and human, for which we used the vM25 and v35 primary assembly Gencode annotations, respectively[94,95] (Data S22). We noticed that some well-established immune-cell marker genes are absent from the NMR Ensembl annotation, such as the constant regions of the β and γ TCR loci yet are present in the RefSeq[95] GCF_000247695.1 annotation. Moreover, the Ensembl and RefSeq genome assemblies do not have a perfect one-to-one mapping between their respective scaffolds, and therefore it is possible that coordinates of genes annotated by only one of these databases might not exist in the genome assembly of the other database, and vice versa. In addition, in the construction of the phyletic patterns we include hits of putative pseudogenes, which are not annotated by either database (see more details below) yet may reside in scaffolds only annotated by one of the databases. Due to these challenges, we merged the Ensembl and Refseq genome assemblies and annotations for each of the 67 mammalian genomes. To achieve this while minimizing introduction of assembly and annotation redundancies, for each of the 67 mammalian species, we downloaded the <species_name>.<genome_assembly_name>.<ensembl_release>.dna.top.level.fa.gz genome sequence file, the <species_name>.<genome_assembly_name>.<ensembl_release>.gtf.gz genome annotation file, and the <species_name>.<genome_assembly_name>.<ensembl_release>.entrez.tsv.gz and <species_name>.<genome_assembly_name>.<ensembl_release>.refseq.tsv.gz files mapping between Ensembl and RefSeq genes, and their corresponding RefSeq <RefSeq_assembly_accession>_<genome_assembly_name>_genomic.fna genome sequence file, the <RefSeq_assembly_accession>_<genome_assembly_name>_genomic.gtf genome annotation file, as well as the RefSeq <RefSeq_assembly_accession>_<genome_assembly_name>_assembly_report.txt file that maps between the genome scaffold names of the two databases. Following that, we eliminated from the RefSeq genome annotation file all genes present in the Ensembl-to-RefSeq mapping files, and from the RefSeq genome sequence file all scaffolds with a one-to-one mapping in the Ensembl assembly. If subsequent to these filtering steps genes and/or scaffolds remained in the RefSeq genome annotation and

sequence files they were augmented to the corresponding Ensembl set of genome annotation and sequence files. Coordinates of augmented genes located on scaffolds with an Ensembl mapping were converted to the coordinate system of the latter database. Finally, for each species, we created a set of amino-acid and RNA (cDNA and non-coding RNA) sequences files that encompass the merged annotations (Data S22).

## Detecting annotated protein-coding and pseudo -genes, and putative unannotated pseudogenes.

Immune gene families, such as those encoding the α and β variable TCR regions and their MHC-I and MHC-II ligands, have been subject to strong pathogenic selective pressures and are therefore characterized by having evolved under high rates of gene birth and death[13,107,108]. In order to construct a phyletic pattern that represents the evolutionary histories of these gene families we sought to detect both their annotated protein-coding and pseudo -genes (hence extant and recently extinct genes) and unannotated putative pseudogenes (anciently extinct). To this end, we used the following reciprocal BLAST approach. Each annotated constant and variable region of the four TCR loci, as well as the *MHC-I* and *MHC-II* genes, encodes one or more characteristic conserved protein domains, annotated by the NCBI Conserved Domains Database (CDD)[109] (Data S23). We thus first used the translated BLAST nucleotides (tBLASTn) tool[110] to search a translated genome for hits of a given conserved domain using its position specific scoring matrix (PSSM) as query and retaining the top ten hits, and subsequently used the reverse position-specific BLAST (rpsBLAST) tool[111] with all three open reading frame translations of these PSSM top ten hits as queries against the entire CDD, eventually retaining only those genomic regions which hit the conserved domain we initiated the search with. For the variable TCR regions we benchmarked this approach using the mouse and human genome annotations. Only in the human genome this approach found three unannotated Vδ domain hits, which translates to potential false positive rates of 0% and 2.3%, for the mouse and human genomes, respectively (Fig. S2A, B for the mouse and human genomes, respectively; Data S16 for hits in all genomes). The false negative rates across all TCR loci for the mouse and human genomes (annotated variable TCR regions that our approach did not find) were 6.75% and 9.1%, respectively, where the vast majority of them are annotated pseudogenes (Fig. S2A, B for the mouse and human genomes, respectively; Data S16 for hits in all genomes).

The situation for the *MHC* genes is more complicated. First, whereas each of the variable TCR regions has only a single unique conserved domain, annotated *MHC* genes are more diverse and hence harbor many more conserved domains: 20 for *MHC-I* (classical and non-classical) and 15 for *MHC-II* (from which we filtered out the DM and DO *MHC-II* genes because they function intracellularly) (Data S19). Some of these conserved domains are also harbored by *MHC*-related gene families we wished to exclude, such as the *MHC-I*-like *MR1*, *MICC*, and *CD1* gene families.

In order to minimize the number of false positives that can arise due to this issue we first clustered the proteome and transcriptome sequences of the 67 genomes to groups of orthologs (orthogroups) using OrthoFinder[112] (Data S24). Any conserved domain hit which lies within an annotated region of the genome that encodes a gene that is not a member of any of the *MHC* orthogroups was thus filtered out. In addition, any conserved domain hit which lies within an unannotated region of the genome was filtered out if that region lies in proximity to genomic regions on which non-*MHC* orthogroups are encoded (Data S25 lists the intervals that were eliminated). In other words, our conserved *MHC* domain hits can be classified as: (1) annotated: hits that are encoded on annotated regions of the genome where the annotated genes are members of *MHC* orthogroups; (2) syntenic unannotated: hits that are encoded on an unannotated region of the genome, which is syntenic to a genomic region on which *MHC* orthogroups are

encoded; and (3) non-syntenic unannotated: hits that are encoded on an unannotated region of the genome which is non-syntenic to any *MHC* orthogroup. In the well annotated human and mouse genomes, 97.6% and 89.8% of the conserved *MHC-I* domain hits are annotated, respectively, and the remaining respective 2.4% and 10.2% are syntenic unannotated (Fig. S4A; Data S19), which can thus be regarded as upper bounds to our false positive rate. With respect to false negatives, with the exception of human *HLA-S* (Enembl accession: ENSG00000225851), all annotated *MHC-I* genes in the human and mouse genomes were detected by our approach (Data S19). For *MHC-II*, in the human genome 83.3% of the conserved domain hits are annotated, 11.1% are syntenic unannotated, and the remaining 5.6% are non-syntenic unannotated. In the mouse genome, 94.4% of the hits are annotated and the remaining 5.6% are non-syntenic unannotated (Fig. S4D; Data S19). Thus, again these unannotated proportions can be regarded as upper bounds to our false positive rates. With respect to false negatives, all annotated humans and mouse *MHC-II* were detected by our approach (Data S19). Finally, across all 67 genomes, 72.6% of the conserved *MHC-I* domain hits are annotated, 20.1% are syntenic unannotated, and the remaining 7.3% are non-syntenic unannotated, whereas for *MHC-II* these proportions are 78.8%, 17.6%, and 6.6%, respectively (Fig. S4A, S4D, for *MHC-I* and *MHC-II*, respectively; Data S19).

A second issue with detecting *MHC* genes based on conserved domain hits is that *MHC* conserved domain hits is that *MHC* conserved domains are often encoded across multiple exons, in contrast to conserved domains in TCR variable regions. Therefore, counting each conserved *MHC* domain hit as an *MHC* gene would artificially inflate the size of the gene family. However, the distribution of distances between pairs of genomically adjacent conserved *MHC* domain hits, encoding the same gene, is distinguishably different from that of pairs of genomically adjacent conserved *MHC* domain hits encoding different genes (Fig. S4B, S4C showing the distributions for the annotated human and mouse *MHC-I* genes; Fig. S4E, S4F showing the distributions for the annotated human and mouse *MHC-II* genes). Thus, for each genome we used the annotated *MHC* genes to search for a genomic distance cutoff that maximizes the F-measure, which is the harmonic mean of precision and recall ($\frac{precision \cdot recall}{precision + recall}$), of correctly classifying the distances between pairs of genomically adjacent conserved *MHC* domain hits as either intragenic (i.e., the pair of adjacent conserved *MHC* domain hits encodes the same gene) and intergenic (i.e., the pair of adjacent conserved *MHC* domain hits encodes different genes) (Fig. S4B, S4C showing the cutoffs for human and mouse *MHC-I*; Fig. S4E, S4F showing the cutoffs for human and mouse *MHC-II*). With such a cutoff per each search genome, we grouped conserved *MHC* domain hits to genes, whereby per each scaffold with conserved domain hits, we started from the most upstream hit, and assigned each hit to the gene to which its upstream hit was assigned to if the genomic distance separating them was below the selected genomic distance cutoff. Otherwise, that hit was assigned to a 'new' *MHC* gene.

### Phylogenetic least squares

In the phyletic pattern of the α and β, and γ and δ variable TCR regions (Fig. 2A), we noticed that the sum of γ and δ variable TCR regions is positively associated with that of the α and β variable TCR regions. Expressing γ and δ TCR diversities as the sum of the γ and δ variable TCR regions scaled to the sum of the α and β variable TCR regions (Fig. S2D) would ignore the dependency of this ratio between different species in the phylogeny due to their shared evolutionary histories, and hence might misrepresent the distribution of this ratio across the phylogeny. To account for this issue, we fit a phylogenetic least squares model to the sum of the γ and δ variable TCR regions as a function of the sum of the α and β variable TCR regions, using the "gls" function implemented in the R[97] nlme package[113], using the ML method, where for the correlation argument we used the

"corBrownian" function implemented in the R[97,114] ape package[114]. The species tree used in this phylogenetic least squares, and presented in Figs. 2A and 4A, was obtained from http://vertlife.org/data/mammals[115]. The R[2] value for the fit was obtained using the "nagelkerke" function implemented in the R[97] rcompanion package[116]. Since the association between the number of *MHC-I* genes versus the number of *MHC-II* genes was not strong ($R^2 = 0.25$; Fig. S4G), we expressed the *MHC-I* diversity as the size of its gene family scaled to that of *MHC-II* (Fig. 4A).

### Quantifying relaxation in the intensity of purifying selective pressure operating on the *Cd8b* gene in hystricomorphs

Given the lack of *Cd8b* expression in any of the NMR CD8 T-cell subsets, in any scRNA-seq profiled tissue except for in the thymus, we wondered if it is related to a change in the molecular function of the gene, as a result of a different selective regime it evolved under. To test this hypothesis, we selected the one-to-one *Cd8b* orthologous protein isoforms of NMR and DMR, as closely related representatives of the hystricomorph clade, to be contrasted with those of mouse, rat, prairie vole, deer mouse, chinese hamster, and blind mole-rat, as representatives of the murine clade, and the human sequence as an outgroup (Fig. S1M; Data S15). One-to-one protein isoform orthology was obtained during the process of clustering the proteome and transcriptome sequences of the 67 mammalian genomes we used (see Phyletic patterns section for further details), where one-to-one orthologs were defined as best reciprocal BLAST hits. We then aligned these sequences using MAFFT version 7.490[117,118] (Data S8), and subsequently used the corresponding multiple codon-sequence alignment as input to the RELAX tool of the HyPhy suite of codon models[118], specifying the hystricomorph clade as "test" and the murine clade as "reference". The output of RELAX is the estimated *K* parameter, which quantifies the magnitude of relaxation in purifying selection operating on the foreground clade relative to the background clade. In order to compare the estimated *K* values of *Cd8b* with that of other closely related T-cell receptor genes, we repeated this procedure for *Cd8a*, *Cd4*, and the four CD3 subunit genes: *Cd3d*, *Cd3e*, *Cd3g*, and *Cd247* (Fig. S1M; Data S15 for the selected representative sequence IDs and Data S9−S14 for their respective multiple coding-sequence alignments). The phylogenetic tree used in this analysis (Fig. S1P), was obtained by pruning the phylogenetic tree of the 67 mammalian species (see Phylogenetic least squares for details).

### TCR-scRNA-seq data and analysis
**Hybridization-capture and TCR single-cell long-range sequencing.**
In order to estimate T-cell clonotypic diversity, full-length TCR sequencing (at minimum covering the variable, joining, and diversity in the case of the β and δ loci, TCR regions) is required at the single-cell level. Moreover, in order to associate full-length sequenced TCRs with the T-cell subsets they originated from, this single-cell TCR sequencing approach needs to be applied to the same cells whose whole transcriptomes are scRNA-seq profiled (see refs. 60,119 for recent reviews on TCR sequencing approaches). The commercial 10× V(D)J kit satisfies these requirements, yet is limited to human and mouse samples, only targeting their α and β TCRs, and additionally requires use of the 10× 5′ scRNA-seq kit[120]. These requirements make this commercial kit irrelevant for our problem because of its inability to sequence NMR samples and γ and δ TCRs, and since our scRNA-seq spleen data (Fig. 1D−F and S1D−S1F) have already been generated with the standard 10× 3′ kits (Methods). Tu et al. have devised an approach that is able to utilize 3′-barcoded scRNA-seq samples for TCR sequencing yet is reliant on primers targeting variable TCR regions[121], for which our NMR genomic information might not be sufficiently reliable. We thus borrowed from that approach and devised a hybridization-capture approach by adding biotinylated lockdown probes targeting all constant regions of the four TCR loci (Fig. S3A), whose NMR and mouse

genomic sequences are well annotated, to 75% excess 10× Chromium Next Gen Single-Cell 3′ V3 3′-barcoded scRNA-seq cDNA libraries of our sorted splenic T-cell data, which were not used for constructing libraries for short-read sequencing. Hence, we obtained unfragmented cDNA enriched for TCRs by using the xGen Hybridization and Wash kit with the customized Discovery Pools from Integrated DNA Technologies, Inc (IDT, Coralville, Iowa). Lockdown probes (Fig. S3A) were designed by IDT, based on the mouse and NMR TCR constant region sequences (Data S26). For each constant region, probes were generated against the coding sequence using 2× tiling. Probes were QC'ed using the IDT internal BLAST tool against the mouse and NMR genomes (used in the phyletic patterns analyses; Data S26). Custom blocking probes were designed and synthesized by IDT using their proprietary xGen Blocking oligo strategy. For each sample, 500 ng of amplified cDNA was mixed with the blocking probes and xGen human Cot DNA, and the mixtures were dried down in a SpeedVac system (ThermoFisher). The hybridization and washing steps were performed according to the protocol from IDT. The hybridization and post-capture amplification steps were performed in a C1000 Touch Thermal Cycler (Bio-Rad Laboratories, Hercules, CA). Fourteen cycles were used for post-capture PCR amplification, and the amplified cDNA was evaluated using a Bioanalyzer 2100 (Agilent Technologies, Santa Clara, CA) with a high sensitivity chip.

Amplified cDNA at >200 ng per sample was used to generate Pacific Biosciences (PB) SMRTcell libraries. Full-length PB sequencing was performed on the PB Sequel II platform at Histogenetics (Ossining, NY). Target library-depth was calculated based on the estimated distribution of TCR lengths (500–2500 bps), the average target number of cells per sample (6000–10,000), the expression level in the scRNA-seq data (95% of cells have less than 30 TCR transcripts), and the estimated HiFi circular consensus sequences (CCS) per chip (2–2.5 million). We pooled four libraries per one SMRTcell and expected to acquire 25–30 TCR reads per cell.

**Processing of raw read data.** For processing of the hybridization-capture reads we followed several of the steps in the cDNA_Cupcake workflow for IsoSeq single-cell analysis[122]. Namely, for each sample we used the PB SMRT LINK V11 software[123,124] to generate circular consensus sequences (CCS), requiring a predicted accuracy ≥ Q20. Following that, we used the lima software[124] for removing the 5′ and 3′ cDNA primers from the CCSs of each sample, using the command: lima --isoseq --dump-clips <sample_demultiplexed_CCS.bam> <primers.fa>, where primers.fa is a fasta file with these sequences:

>5p

AAGCAGTGGTATCAACGCAGAGTACATGGG

>3p

AGATCGGAAGAGCGTCGTGTAG

We then detected UMIs and cell barcodes using the "tag" command implemented in the Iso-Seq3 software[125], with the "--design T-12U-16B" argument. Subsequently, we removed the polyA tails and artificial concatemers using the "refine" command in the Iso-Seq3 software. Finally, we clustered the reads by the unique founder molecules using the "dedup" command in the Iso-Seq3 software, using the "--max-tag-mismatches 1 --max-tag-shift 1" arguments. We then filtered out any of these deduplicated reads which correspond to cells that we filtered out from our scRNA-seq data. Retained reads were then mapped to their respective genomes (the same ones used for the scRNA-seq data) using the minimap2 aligner[126,127] with the "-t 30 -ax splice -uf --secondary=no" arguments. As our hybridization-capture approach sequenced both the constant and variable, diversity, and joining TCR regions (Fig. S3A), we required that each demultiplexed CCS read, which we mapped to its respective genome, intersected one of the constant TCR regions (Data S27). Any read that did not meet that criterion was filtered out (Fig. S3B–D for NMR and S3E–G for mouse).

**Constructing a TCR IgBLAST database.** In order to use the IgBLAST tool[127] for identifying the variable, joining, and diversity composition in our processed TCR reads we downloaded the mouse and NMR annotated nucleotide sequences (F+ORF+all P) of these regions from the Immunogenetics (IMGT) V-QUEST reference directory[128], June 2022 release. We then used tBLASTn[110,111] with each of the variable region sequences as query against its respective species genome in order to identify their genomic coordinates and test whether they encode their respective TCR conserved protein domain by using rpsBLAST[129] with the three reading frame translations of a hit as queries against the CDD. Although not all these IMGT variable region sequences were found to encode their respective conserved domain, we nevertheless retained them in the IgBLAST database. On the other hand, in the mouse genome our phyletic patterns approach detected 21 and 2 α and δ unannotated variable TCR regions, respectively, where 18 of the α variable TCR regions and both δ variable TCR regions were observed in our TCR read data. In the NMR genome, our phyletic patterns (see Phyletic patterns section for more details) approach detected a single unannotated variable TCR region in each of the α, β, and γ TCR loci, where the γ variable TCR region was also observed in our TCR read data (see Data S28–S30 and S31–S33 for the respective variable, diversity, and joining IgBLAST sequence databases, for the mouse and NMR, respectively). Finally, we also downloaded the mouse germline auxiliary file from IMGT (mouse_gl.aux; Data S34) and created a similar file for NMR (Data S35) in order for IgBLAST to include the CDR3 amino-acid sequences in its hit results.

**Assigning clonotypes to cells.** Prior to using the reads retained after step 2 as IgBLAST queries, we limited the reads to lengths of up to 700 bps in order to prevent IgBLAST from reporting junctions between variable, diversity, and joining regions that are unrealistically downstream only due to cases where the sequencing extended much downstream to that region. We additionally required the constant region to be included in a retained read. We subsequently used the IgBLAST nucleotides (IgBLASTn) command providing the variable, diversity, and joining databases, and germline auxiliary files created in step 2, and the "-domain_system imgt -ig_seqtype TCR -show_translation -num_alignments_V 5 -num_alignments_D 5 -num_alignments_J 5" arguments. Reads with an IgBLAST E-value support >0.001 for their variable or joining TCR regions were filtered out because this implies that they are not TCR transcripts and subsequently, reads labeled as unproductive TCR due to a lack of an open reading frame were additionally filtered out. For the retained reads, we removed the allele information from the hit accessions and subsequently collapsed hits with identical accessions. For a small fraction of hits, IgBLAST maps the read to variable and joining accessions which are discordant with respect to their TCR loci. Most of these cases are typically variable regions that match equally well both to an α and δ variable regions, as they are encoded on the same locus, and IgBLAST does not take into account the TCR locus identities of the variable and joining regions it assigns a hit to. We resolved these discordances using the TCR locus identity of the constant region, which we obtained in step 1. Any read with such a discordance, which was not a combination of the α and δ TCR loci, was filtered out. We then extracted the constant region of each read by removing the read sequence portion downstream to the read start site of the variable region, as determined by IgBLAST. We then followed refs. 130,131, who also used full-length TCR sequencing, applied to immunoglobulin (Ig) and TCR repertoire of the rhesus macaque, and used the CD-HIT software[131], version 4.8.1, with the "-c 0.97 -G 0 -aL 0.95 -AL 100 -aS 0.99 -AS 30" arguments to cluster these constant regions, separately for each of the TCR loci. This step did not change the constant region assignments obtained from step 1. However, we also extended this approach to the variable, diversity, and joining regions, on a TCR-locus cell-specific basis, in order to aid with the clonotype assignment of each read. That is, in each sample, for

each cell, we ran CD-HIT with the same arguments as above, for the variable, diversity, and joining parts of the read sequences, separately for each of the four TCR loci. In cases where a set of cell-specific reads, originating from a specific rearranged TCR locus, spans a range of lengths due to the variability in the TCR fragment extraction and sequencing steps, IgBLAST might not report identical mapping information for them. For example, a sufficiently long read might be mapped to Vβ6, Dβ2, and Jβ2-3, whereas IgBLAST might assign uncertainty in the joining region of a shorter read, which is expressed by a comma-separated list of joining regions, such as: Jβ2-3, Jβ2-4. In such cases, CD-HIT clusters such reads to the same cluster and hence the IgBLAST uncertainty of the joining, diversity, and/or variable region identities can be resolved using all reads in the CD-HIT cluster. In order to use cells with both TCR chains, in addition to filtering out cells with no sequenced constant region, low IgBLAST support for a TCR, and an unproductive TCR, we additionally filtered any cell with either only a single (orphan) TCR chain, no mapping of the diversity region in either the β or δ TCR chain (for αβ and γδ T cells, respectively) (NMR: Fig. S3B–D; Data S17; mouse: Fig. S3E–G; Data S18). We hypothesize that high fractions of orphan-chained cells (Fig. S3D–G) likely stem from a combination of low expression levels of the α and δ TCR chains (relative to that of the γ and β TCR chains)[132,133], lockdown probe efficiency, and insufficient library and sequencing depths.

**Obtaining clonotype network graphs.** For the purpose of displaying T-cell clonotypic diversity we used the Python[103] Scirpy package[134,135], which is part of the Scverse single-cell omics computational ecosystem[135], as follows: distances between pairs of clonotypes were computed based on their CDR3 amino-acid sequence alignment with a cutoff score of 15, where the subsequent clonotype clusters were obtained using both VJ and VDJ TCR arms, and the subsequent clonotype cluster network required a minimum of three cells per clonotype cluster. Clonotype networks were plotted using the R[97] ggplot2 package[60,136].

**Quantifying T-cell clonotypic diversity.** For the purpose of quantifying clonotypic diversity for each T-cell subset in each sample we used three distinct metrics which capture different aspects of diversity: (1) Hill number, which relates to the number of unique clonotypes without accounting for their abundances and hence quantifies richness; (2) Shannon entropy, which weighs clonotypes by their abundances and hence quantifies evenness; and (3) Gini-Simpson index, which gives more weight to rare clonotypes (reviewed in ref. 60) (Fig. 3H for NMR and 3Q for mouse). In order to contrast between clonotypic diversities, either between old and adult samples of the same T-cell subset, or between two different T-cell subsets either of the same age in each species (Fig. 3E for NMR and 3J for mouse), or between the two species (Fig. 3K), we used the "mcpHill" function implemented in the R[97] simboot package[137], using all default parameters (hence 5000 bootstrap replications).

**Reconstructing the Vγ phylogenetic tree.** In order to shed light on the dominant clonotypes in the NMR cytotoxic γδT-cell subset, we reconstructed the phylogenetic tree of Vγ. To this end, we first retrieved the genomic sequences of the Vγ TCR regions from all 67 mammalian genomes included in our phyletic pattern analysis (see Phyletic patterns section for details). Following that, in order to obtain a reliable multiple Vγ sequence alignment, we first translated these sequences and filtered out any sequence for which we did not obtain an open reading frame, and subsequently ran these amino-acid sequences through the GUIDENCE2 tool[117,138,139] using MAFFT version 7.490[117,140] as the selected aligner (Data S36 is the resulting multiple sequence alignment). Subsequently, for reconstructing the Vγ phylogeny using the GUIDENCE2 corrected multiple Vγ sequence alignment we used RAxML version 8.2.9[140] with the "-p 1 -m PROTGAMMAWAG -T

5 -x 1 -# autoFC" arguments. For the purpose of presentation, we only kept the human, mouse, and NMR Vγ phylogenetic tree tips (Fig. S3H). Although the bootstrap support values for the splits in the Vγ phylogenetic tree are not large, that tree is in strong agreement a tree presented in a recent review on the evolution of the TCRγ locus in mammals[141].

## Statistics and reproducibility

Animal studies (sequencing and histology) were designed to have a minimum of three samples (animals) per group (species, and where relevant age and sex). Aside from the splenic T-cell sequencing study, where we did not have enough old-age females and hence used only males, all other studies conducted in this work included both sexes. For the NMR samples, animals were selected from distinct colonies in order to avoid colony-specific biases. Any statistical method used in this work is referred to from its relevant part within the Methods section. In all cases where multiple hypotheses statistical tests were performed p-values were adjusted using the False Discovery Rate method[142].

## Reporting summary

Further information on research design is available in the Nature Portfolio Reporting Summary linked to this article.

## Data availability

The Raw fastq files, filtered (empty barcodes and sparse genes) gene-by-barcode UMI count comma separated files, and aligned read BAM files, generated in this study have been deposited in the Gene Expression Omnibus (GEO), under accession code GSE214390. All other data produced and used in this work are provided as Supplementary Information. Source data are provided with this paper.

## Code availability

Scripts to build data resources and reproduce figures are available at https://zenodo.org/record/8384311.

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

## Acknowledgements

We thank Anh Diep for her advice on the histology work; Corey Smith, Silvino Sousa, Ruth Febo, Sheeba Mathew, Sakshi Bhargava, and Yuliya Kutskova from AbbVie for generating the anti-NMR CD3e monoclonal antibody. We thank Alex Chekholko for his generous help with setting up computational resources for this work. We also thank Vladimir Jojic and Jun Xu for their help in allocating resources for the work. Finally, we thank David Zeevi, Kevin Wright, Maria Pokrovskii, Fiona Harding, Jonathon O'Brian, Katie Podshivalova, David Botstein, and the members of senior staff at Calico for their critique of the manuscript. Calico Life Sciences kindly funded this study.

## Author contributions

Conceptualization: T.L., N.D.R. and R.B.; Data Curation: T.L. and N.D.R.; Data Analysis: N.D.R.; Methodology: T.L., N.D.R., N.L.F., B.M.M., R.P., M.D., and M.A.R.; Investigation: T.L. and N.D.R.; Project Administration: R.B.; Resources: M.S., W.C., and R.B.; Software: N.D.R.; Visualization: N.D.R.; Writing: T.L., N.D.R., and R.B.

## Competing interests

The authors declare no competing interests.
