## [Peer Review File · Nature Communications]

Evolution of T cells in the cancer-resistant naked mole-ratREVIEWER COMMENTS

Reviewer #1 (Remarks to the Author):

Review of Lin et al

Lin et al provide an interesting exploration of the immune system of the Naked Mole Rat (NMR), a species with an anomalously long life-span and strong cancer resistance, and curious immunological features including apparent loss of the NK cell compartment. The study focusses on the T cell compartment, with chief findings including a limited level of diversity in the CD8 T cell subset, concurrent evidence for smaller initial thymic size and modest thymic involution over age, a substantial circulating $\gamma\delta$ T cell compartment. They are interpreted as suggesting relaxed selective pressure from intracellular pathogens, and the authors suggest the $\gamma\delta$ T cell compartment might have evolved to mediated increased cancer immunosurveillance, driving greater longevity.

Strengths

On the positive side, the study includes substantial new data regarding the NMR T cell compartment, including comparisons to the mouse.

Weaknesses/Caveats

The study has several weaknesses, of which three are highlighted below.

(i) Correlative nature, lack on conclusiveness, and and lack of direct data on anti-cancer activity Chief amongst the weaknesses is that the study is inherently observational, and rests on the correlation of long lifespan in NMRs with its particular immune features. Whether there is a causal link between these immune features and the long lifespan is entirely unclear. Not only are no direct data included on cancer reactivity by any of the immune subsets involved, but other explanations are largely ignored.

In contrast, genetic processes underlying cancer protection such as the DNA damage response, or other non-immune factors, could be alternative explanations, and another critical factor, notably the strength of cancer risk factors in the NMR's subterranean habitat, is another important point, again ignored (presumably UV exposure would be substantially reduced). Consequently, distinct NMR immune features, including a $\gamma\delta$ compartment more skewed to the circulatory/splenic niche than solid tissues, may relate more to defence against pathogens in its distinct subterranean ecological niche. As a result, the study is not only fundamentally inconclusive (see major point 3) regarding key hypotheses, but is weak on discussing these alternative explanations for its long life-span.

(ii) The curious lifestyle of the NMR would be worth commenting on, in particular its eusocial nature, with potential monogamous breeding pairs (Szafranski et al, *Front. Ecol. Evol.*, 2022; *The Mating Pattern of Captive Naked Mole-Rats is Best Described by a Monogamy Model.*) involving a single 'queen' and 'king', and related coworkers with limited fertility, potentially reflecting a mammalian example of kin selection. Of relevance to the study is the underlying evolutionary drive towards a long life-span in the NMR, which could be discussed.

(iii) The male/female bias in some experiments should be commented on. Also, it should be stipulated if the animals are breeding or non-breeding animals.

Major criticisms

1) In the Introduction (second paragraph, Page 3), the description of $\gamma\delta$ TCR clonotypes as 'less diverse and pre-programmed during thymic development' and linking with recognition of 'self and foreign stress ligands' is severely flawed. Studies in humans have highlighted an innate-like subset expressing a semi-invariant V γ 9V δ 2 TCR repertoire (Davey et al, *Nat Commun* 2018, PMID: 29720665), but in parallel an adaptive-like subset (Davey et al, *Nat Commun* 2017, PMID: 28248310) that is highly diverse clonotypically, is entirely composed clonotype private to each

individual, and lacks effector function upon initial thymic generation. These features and studies should be referred to at this point in the introduction.

2) The study's results suggesting modest thymic involution over time, first mentioned in the introduction (Page 5, lines 1-2 and last paragraph before the Results), but again in the Results section (final paragraph before Discussion), appear to be at odds with those of a previous study (Emmrich et al, Reference 33) that concluded no involution took place. How this discrepancy can be resolved/explained should be addressed in the Discussion, which is currently lacking.

3) Regarding decreased α and β TCR V gene diversity relative to other mammals (eg final Results section; Discussion).

a) The decreased numbers of α and β TCR V regions observed in NMR relative to other mammals, which is indicative of most hystricomorph species, is suggested to reflect decreased evolutionary challenge from intracellular pathogens. However, the same TCR genes will be used for CD4 responses to mount responses to extracellular pathogens, so this interpretation is arguably flawed – could it not instead be interpreted as a decreased pathogenic challenge more generally ?

b) The finding that other hystricomorpha species have similar immune gene expansion/contraction profiles to NMR but do not exhibit disproportionately long life-spans (in contrast to the guinea pig which does not conform to the profile of immune gene expansion/contraction seen in NMR) adds to the impression that it is challenging to infer too much from these kind of inter-species differences without additional data.

c) If one is convinced that the NMR has been faced with a decreased challenge from intracellular pathogens, then it arguably begs the question of what the $\gamma\delta$ T cells are present for ? Is this for a residual role in intracellular pathogen defense in the circulatory/splenic niche, or a role in cancer immunosurveillance, in which case why would they be primarily circulatory/splenic in localisation ?

4) Clonotypic analysis

The clonotypic analysis as set out in the figures is limited and I have several points to consider and address.

a) The manuscript equates (eg on page 11, first sentence off sixth Results section) V region number with clonotypic diversity, but the extent of N/P region addition is a major variable, independent of the V region number, that will determine repertoire diversity and that can differ between TCR loci (eg TCR- γ vs TCR- δ) and different $\gamma\delta$ subsets.

b) When clonotype data are presented, nucleotide sequences should be shown, and assignments to V, D (where applicable), J and also N/P nucleotides assigned where possible. This will help explain public CDR3 sequences, which are fairly short and would be expected to be largely germline-derived with minimal N/P additions. The current text in relation to this (page 12 line 5) is highly confusing ('where in both...sample') and should be clarified.

c) In relation to V region usage of expanded clonotypes, it is of interest that the sequencing revealed a prevalent sequence which aligns with mouse V γ 4. This could be a component of an invariant population analogous to the V γ 4V δ 5 mouse population (Kashani et al, 2016 Nature Comms), a population which produces IL-17. It is interesting that no NMR $\gamma\delta$ population appears to make IL-17 or express Ror γ t, and also that no equivalent population to the V γ 6V δ 1 mouse population that produces IL-17 was observed in the spleen. As this mouse population is rare in the circulation and more prevalent in tissues such as lung and uterus, this again emphasises that the lack of analysis of NMR $\gamma\delta$ T cells in tissues is a shame.

d) The lack of Vdelta chain sequencing is highly unfortunate, particularly as TCR γ chain sequences generally tend to be more public (due to shorter CDR3 regions using only V-J gene segments and with fewer N/P additions) and this should be rectified. Importantly, the TCR-delta chain could potentially be highly diverse (owing to V-D-J gene segment usage and N nucleotide addition between each gene segment, indicating a semi-invariant subset if V γ sequences are public) or essentially invariant, and this is an important issue to resolve.

e) The significance of a TCR- γ clonotype being present only in one of the two $\gamma\delta$ T cell subsets is not discussed.

f) Page 13 line 4 'Neither..'

This should reference evidence for adaptive-like biology and infection-driven clonal focussing in human $\gamma\delta$ T cells, in particular Ravens et al, Nat Immunol 2017, PMID: 28218745; Davey et al, Nat Commun 2017, PMID: 28248310; and Davey et al, Nat Commun 2018, PMID: 29720665.

g) On page 14 (penultimate sentence before final Results section, starting 'By contrast...'), the decreased CD8 diversity in NMR vs mice is employed to suggest a decreased selection pressure from intracellular pathogens; conversely NMR have higher CD4 diversity than mice, therefore is it not logical to suggest an increased selection pressure from extracellular pathogens compared to mice ?

h) Given recent findings highlighting butyrophilin family proteins as direct ligands for germline-encoded regions of particular V γ chains (eg Willcox et al, Immunity, 2019, PMID: 31628053; Karunakaran et al, Immunity, 2020, PMID: 32155411), it would be interesting to compare the number and nature of butyrophilin family member genes in relation to V γ genes in the NMR compared to eg mice and human, as it is possible these have co-evolved alongside each other, and BTN/BTNL gene products could be involved in regulation of NMR $\gamma\delta$ T cell functionality.

5. Evolutionary arguments

Some of the arguments in the Discussion regarding evolutionary aspects are confusing or require clarification.

a) On page 17 (sentence starting 'Although intracellular pathogens...', in relation to intracellular pathogens exhibiting strong selective pressure, the authors speculate that the 'subterranean ecological niche' could present 'an evolutionary dead end'. The meaning of this is unclear – presumably a dead end (ie less viable) for intracellular pathogens ? This should be rephrased to increase clarity.

b) In the next sentence, the authors make the linked point that this 'allowed the selective pressure under which NMRs have evolved to shift towards elimination of early malignancies and maintenance of tissue homeostasis, thereby promoting a long reproductive lifespan, to which their immune system has responded by utilizing $\gamma\delta$ T cells.'

This argument seems to suggest that in most mammals, evolution couldn't work on tackling early malignancies and promoting long lifespan because it is 'too busy' tackling pathogens. But evolution is a multitasker. This seems either a bad explanation or a bad idea or both. A more acceptable rephrasing of the authors' argument could be that adaptations that help the immune system to tackle pathogens might simultaneously compromise the immune system's ability to tackle cancer and that the NMR represents a more mechanistically pure form of immune configuration that is more focussed on malignancy/lifespan. Arguably a more fundamental but important question is why evolutionarily it makes sense for NMRs to have a longer life-span ? Do they meet/mate infrequently ? Grandparental importance ? This is not discussed.

c) Regarding the potential increased role of $\gamma\delta$ T cells in cancer resistance in NMRs, it seems surprising that such a role would link with a more haematological localisation for the $\gamma\delta$ compartment. Could the authors comment on this ? Given that, at least in humans, 95% of cancers are non-haematological in origin, this preferential circulatory localisation does not exactly emphasise a role in cancer suppression. From this perspective, it is unfortunate the authors did not examine the TCR repertoire and $\gamma\delta$ numbers in epithelial tissue, particularly as $\gamma\delta$ T cells in other species do generally adopt a tissue localisation (eg gut, skin, etc).

Minor comments

(i) In the Introduction, line 6: the wording is misleading regarding the innate component, which one does not usually consider recognising MHC molecules. This could be rewritten to avoid this confusion.

(ii) On Page 7, line 4, what is the basis of suggesting the non-cytotoxic $\gamma\delta$ T cell subset may be analogous to the mouse IL-17 producing cells? This was unclear to this reviewer.

(iii) Page 13, 4th line from the bottom: there is a type with 'expectedmogilenko' – mogilenko is first author of publication number 43.

(iv) a period is required on Page 18 line 6 of the middle paragraph after 'activation'

(v) the text should read 'diversities OF their...' on Page 19, second line of the bottom paragraph

(v) 'steppingstone' should be two words on the 3rd line of the last paragraph.

(vi) regarding thymus involution on Page 19, the meaning of one sentence is very unclear: 'Based on that, the most striking differences...become intuitive'. What does this mean? This should be reworded to increase clarity.

Reviewer #2 (Remarks to the Author):

Review of article submitted to Nature Communications

Overall Critique.

The observations reported come from genetic and transcriptome analyses and are most interesting and the methods employed appear to be state-of-the-art. In general, the conclusions reflect the genetic data. Extrapolating this to pathogen levels without data is a weakness. The study may have relevance to understanding NK and T cell evolution, the control of intracellular pathogens and perhaps, malignant cells. The presentation could be improved by:

(a) Reducing redundancy in the abstract and introduction.

(b) Better appreciating that other mammals that do not live in subterranean burrows, also have an abundance of g/d T cells. This should have been acknowledged in the introduction.

(c) Clarifying that the study is solely based on genetics either comparing genome or using transcriptomes analysis, but without any evidence that NMR have fewer intracellular pathogens.

(d) Assuming that readers do not need a review of basic immunology, so they should immediately focus on RMD and the purpose of the study (see specific comments # 2 & 4).

(e) Confine their speculation and hypotheses to the discussion and minimize speculation about issues for which they provide no data.

Specific comments

1. Condenses the general speculative aspects of the abstract and instead provides more specific details as to the outcome of the study.

2. Page 3 and the first two paragraphs on page 4 are unnecessary unless the audience has no immunological background. Also, some details are wrong, since g/d T cells are abundant in circulation of pigs and cattle and not confined to peripheral /mucosal tissues. Authors should make clear they are comparing RMD to conventional lab mice and rats, not all other species.

3. Start the Introduction with the last paragraph on page 4.

4. Page 5, second paragraph. Extending their comparative study beyond lab mice and RMD in future would be wise and should include other species with high levels of g/d T cells such as pigs and cattle.

5. The section on T cell phenotypes ([5-8] is better judged by a reviewer who works in this area.

6. Page 8. Are there other criteria that can be used to distinguish a lymph node from a thymic tissue? B cells follicles in thymus were reported 50 years ago and in swine, thymocytes continue to rearrange Ig genes. Authors would consider this since they base their data on sc RNA (transcriptomes).

7. "SUMMARY et al" This is pure speculation without evidence that the lifestyle of NMR exposes them to fewer intracellular pathogens. Authors should provide comparative data to support their view. Perhaps there are published data that can be cited.

Reviewer #3 (Remarks to the Author):

In the paper "Evolution of T cells in the cancer-resistant naked mole-rat" authors show that Naked Mole Rat (NMR) display a splenic (circulating) population of $\gamma\delta$ T that can be divided into cytotoxic and non-cytotoxic subsets. $\gamma\delta$ T cells can be identified in NMR Bone Marrow (BM) and thymus/cervical lymphnode as well. NMR thymic $\gamma\delta$ T cells do not express ITGAE (as opposed to mouse) possibly akin to their circulating behaviour as opposed to the tissue residency observed in mouse. In addition, authors show a disproportionately large diversity of $\gamma\delta$ V genes as opposed to $\alpha\beta$ V in NMR together with a reduced diversity of MHC-I gene repertoires, suggesting a preferential exposure to non-MHCI ligands due to a relaxed selective pressure against intracellular pathogens. Finally, authors highlight that $\gamma\delta$ T cells show lower clonotype diversity as compared to $\alpha\beta$ T cell in NMR and higher level of clonotype sharing across subjects and that cytotoxic $\gamma\delta$ T cells show lower diversity (clonal expansion) when compared to non-cytotoxic $\gamma\delta$ T cells

To our knowledge, this is the first report to dissect at such level of resolution the adaptive immune system of NMR (contrasted with the more characterized mouse immune system). Given the characteristic features of NMR, namely the longer lifespan and cancer resistance, as reported by the authors, we believe the work can be of significance in the field of comparative genomics and, with additional work further dissecting the carcinogenetic process in NMR, of marginal interest in the field of cancer biology and cancer immunology.

The claim that in NMR exists a circulating $\gamma\delta$ T cell population, though supported only by scRNAseq data, looks solid. Additional evidences as provided by flow cytometry technology would further corroborate such claim although we realize that high quality antibodies may not be available for organisms under consideration. The claim that "NMR have evolved under a relaxed intracellular pathogenic selective pressure" is adequately supported and stands as a valid theory.

The annotation of a cellular subset in the splenic T cell dataset as ILC3 is based on the absence of expression of Cd3 isoforms and on the expression of Rorc and is further confirmed later on by the absence of reconstructed Tcr sequences. Nonetheless, the heatmap displayed in Fig1D suggests the existence of a set of genes characteristic for this population. It would be useful to explore whether those genes support the suggested annotation, also considering that Rorc expression does not appear to be sufficient for lineage specification as it appears expressed at similar level also in cytotoxic and non-cytotoxic $\gamma\delta$ T cells. Furthermore, these cells were identified only in the splenic compartment but not in BM or thymus, thus giving rise to the question as to what is the origin and role of these cells in NMR.

In figure S1D, relative proportions of identified cell types is reported for animals at different stages of life. The main text referring to this panel reads "both cytotoxic and non-cytotoxic $\gamma\delta$ T-cell subsets are overrepresented among the old age samples". While this statement may be considered true for the cytotoxic subset, for the non-cytotoxic subset data appear to be skewed by one single old age animal, while the others show similar or even lower proportions of these cells as compared to young animals. In general, this statement may benefit from a stronger statistical evaluation. Given the origin of data as scRNAseq, a tool that may prove helpful in this is MILO.

In Figure1, different tissues are analyzed by scRNAseq (Spleen, Bone Marrow and Thymus). Contextually, other covariates are evaluated (Age in the Spleen dataset, Sex in Bone Marrow and Thymus datasets) but no comment is made as to why in different samples different covariates were considered, in particular it would be important to assess age effects in cellular composition of the thymus considering subsequent analysis performed to evaluate the development of this organ during life span.

In Figure1G/H are reported 2D embeddings of BM derived scRNAseq datasets for NMR and mouse. In the main text referring these panels, they are compared and deemed similar for cellular composition and differentiation trajectories but no specific analysis is performed to evaluate

trajectories and UMAP embeddings (though able to partially capture developmental processes) are not designed specifically for this task. I would suggest either rephrasing the text, or adding specific analysis (using DiffMap, Monocle, or similar tools). The same goes for panels J/K where schematic developmental trajectories appear to be handwritten, without any further support.

The statement according to which: "while mouse thymic $\gamma\delta$ T cells show high and ubiquitous expression levels of *Itgae* [...], NMR thymic $\gamma\delta$ T cells show a far lower and limited *Itgae* expression pattern" does not appear to be sufficiently supported. In figure S1L, *Itgae* average normalized expression is comparable between mentioned cellular subset and the same goes for the fraction of cells effectively expressing the gene (in both cases apparently below 10% of the subset). More robust statistical analysis is required to state that expression levels differ between cell populations for the gene in analysis, let alone to leverage this observation as a mechanistic explanation for an observed phenomenon.

In Figure1Q it is unclear what the numbers reported over the bars stand for, especially considering that values do not match with provided axis. In addition, it is claimed that *Cd8b* gene has evolved "under relaxed purifying selective pressure relative to murine" but little is said about *Cd4* which shows a similar trend and no table with statistical test results is provided.

In general, in figure1, panels C/F/I/L representing heatmaps of gene expression levels across individual cells in different datasets are difficult to read and representations that take into consideration a smaller number of genes (the ones most widely discussed in the main text for example) could make easier following the discourse.

In its current form, figure S2C is not easily readable. It should also be noted that, although NMR show positive #Vg+#Vd residuals, they fall within the confidence interval of the regression line, taking away strength to the claim that the observed relative increase in the number of genes for gdV regions is the result of selective pressure.

Globally, Figure3 is too cluttered and difficult to follow. My suggestion would be to consider moving panels H/I/S/Q/R to a new figure more focused on statistical inference and reformat them to increase readability. In addition, since the function used for statistical testing (*mcpHill*) relies on bootstrapping, it would be important to also report the number of bootstrap iterations.

Furthermore, heatmaps in panels D/G/M/P do not seem the most efficient way to convey desired message and could be substituted with plots as produced by the *clonotype_network* function from the plotting submodule of the *scirpy* package.

Authors dedicate a lot of words speculating on the phenotype of cells associated with the observed public clonotype in cytotoxic $\gamma\delta$ T cells based on their previous phylogenetic reconstruction but nothing is said with regards to the observed molecular phenotype of these cells as assessed in their scRNAseq dataset.

In general, methods employed are in line with current standards and in particular, great effort has been poured into curating and integrating multiple annotations for a large number of species.

Assumptions made are reasonable and where possible, adequate controls have been proposed, yielding results that may prove useful in refining annotation for lesser characterized organisms like NMR. Commendable is also the experimental and bioinformatic work displayed for reconstructing and analysing TCR sequences.

Methods are generally well documented. Nonetheless, we believe that sharing original scripts, possibly organized in a reproducible pipeline with workflow management systems like *SnakeMake* or *NextFlow*, would greatly improve reproducibility and readability.

We wish to thank each of the reviewers for their critical and thoughtful comments, which have helped improved our manuscript.

R#2:

Review of Lin et al

Lin et al provide an interesting exploration of the immune system of the Naked Mole Rat (NMR), a species with an anomalously long life-span and strong cancer resistance, and curious immunological features including apparent loss of the NK cell compartment. The study focusses on the T cell compartment, with chief findings including a limited level of diversity in the CD8 T cell subset, concurrent evidence for smaller initial thymic size and modest thymic involution over age, a substantial circulating $\gamma\delta$ T cell compartment. They are interpreted as suggesting relaxed selective pressure from intracellular pathogens, and the authors suggest the $\gamma\delta$ T cell compartment might have evolved to mediated increased cancer immunosurveillance, driving greater longevity.

Strengths

On the positive side, the study includes substantial new data regarding the NMR T cell compartment, including comparisons to the mouse.

>

We thank the reviewer for appreciating our contribution.

Weaknesses/Caveats

The study has several weaknesses, of which three are highlighted below.

(i) Correlative nature, lack on conclusiveness, and and lack of direct data on anti-cancer activity

>

Despite decades of NMR research, the NMR immune system has not received any attention up until recently. In our previous publication, which was the first to characterize the NMR immune-cell repertoire, we discovered it lacks NK cells and provided a solid explanation for how this occurred during evolution. The current manuscript is the second part of the characterization of the NMR immune-cell repertoire and has similar merits where the focus herein is on T cells, the evolution of the genes that regulate their function, and their clonotypic diversity, contrasted with that of the well characterized mouse T cells. Anti-cancer T-cell function was not the intent and focus of this work, otherwise we would have presented anti-cancer functional data, hence we apologize if we gave that impression. However, due to the established roles of T cells in the context of cancer, and especially of $\gamma\delta$ T cells, it is impossible to characterize NMR T cells

without referring to the relevant literature. In our Discussion, we do speculate that the circulating cytotoxic $\gamma\delta$ T-cell population in NMR might have a pivotal role in cancer resistance by rapidly eliminating transformed cells and we suggest potential experiments to test this idea in follow-up studies. To help avoid creating the impression that the focus is on anti-cancer mechanisms we made two changes to our revised Discussion text where in addition to referring to cancer resistance we added homeostasis, thus expanding on the possible roles of $\gamma\delta$ T cells:

1. In the first paragraph of the Discussion we now write:

“The fact that we detect a dominant and public NMR T-cell clonotype only in its cytotoxic $\gamma\delta$ T cells makes it tempting to speculate that this subset recognizes frequently encountered ligands that signal a threat which if not rapidly reacted against wreaks havoc, such as newly early malignancies and acute disruptions of homeostasis.”

2. In the second paragraph of the Discussion we now write:

“The NMR is not the only mammalian species with a large proportion of $\gamma\delta$ T cells and large diversity of γ and δ variable TCR regions. Ruminants and equines also have a large proportion of circulating $\gamma\delta$ T cells^{76–78} and large genomic diversities of γ and δ variable TCR regions. While these studies afford some insights into $\gamma\delta$ T-cell function, we do not know if and how they may relate to cancer resistance, maintenance of homeostasis, and lifespan. This is because our understanding of those processes in these farm animals is limited and may have been affected by the domestication⁷⁹, albeit it would be interesting to profile the immune-cell repertoires and T-cell clonotypic diversities in these species.”

Chief amongst the weaknesses is that the study is inherently observational, and rests on the correlation of long lifespan in NMRs with its cancer reactivity by any of the immune subsets involved, but other explanations are largely ignored.

>

This comment is addressed in the answer to the first comment.

In contrast, genetic processes underlying cancer protection such as the DNA damage response, or other non-immune factors, could be alternative explanations, and another critical factor, notably the strength of cancer risk factors in the NMR’s subterranean habitat, is another important point, again ignored (presumably UV exposure would be substantially reduced).

Consequently, distinct NMR immune features, including a $\gamma\delta$ compartment more skewed to the circulatory/splenic niche than solid tissues, may relate more to defence against pathogens in its distinct subterranean ecological niche. As a result, the study is not only fundamentally inconclusive (see major point 3) regarding key hypotheses, but is weak on discussing these alternative explanations for its long life-span.

>

We have expanded a part of our revised Introduction text to acknowledge the possible contribution of cell autonomous factors to the NMR's cancer resistance. We now write: "A cell-autonomous mechanism explaining the NMR resistance to solid tumors³² has been challenged³³, although there may likely be other, yet to be characterized ones, such as evolutionary genomic expansion of tumor suppressor gene families³⁴, as in the case of the elephant's *TP53*³⁵⁻³⁷."

As for defense against pathogens in the subterranean niche, the evolutionary loss of NK cells and NK-cell receptors genes, along with the small genomic diversity of the *MHC-I* gene family, strongly suggest a relaxation of selective pressure imposed by intracellular pathogens. If NMRs have been evolving under a strong selective pressure by subterranean-specific extracellular pathogens we would expect to see some degree of expansion in their *MHC-II* gene family, however that is not the case. While protection from UV exposure might indeed lower the risk of cancer, this risk is limited to specific types of cancers, and more generally, it is not clear if the subterranean niche directly and substantially reduces risk of all cancers.

(ii) The curious lifestyle of the NMR would be worth commenting on, in particular its eusocial nature, with potential monogamous breeding pairs (Szafranski et al, Front. Ecol. Evol., 2022; The Mating Pattern of Captive Naked Mole-Rats is Best Described by a Monogamy Model.) involving a single 'queen' and 'king', and related coworkers with limited fertility, potentially reflecting a mammalian example of kin selection. Of relevance to the study is the underlying evolutionary drive towards a long life-span in the NMR, which could be discussed.

>

In our Discussion we write about the eusocial lifestyle of NMRs and its possible connection to cancer resistance and the immune system. That said, it is not clear to us how to connect the mating/breeding system of NMRs to the focus of this manuscript.

(iii) The male/female bias in some experiments should be commented on. Also, it should be stipulated if the animals are breeding or non-breeding animals.

>

Whenever both sexes were included, it was to make sure that our observations are not sex specific, which is the case. Sex effects in gene expression, if any, were found to be marginal and not of relevance to our focus in this work. In addition, the “Animals” paragraph in our Methods states that all are non-breeding virgins.

Major criticisms

1) In the Introduction (second paragraph, Page 3), the description of $\gamma\delta$ TCR clonotypes as ‘less diverse and pre-programmed during thymic development’ and linking with recognition of ‘self and foreign stress ligands’ is severely flawed. Studies in humans have highlighted an innate-like subset expressing a semi-invariant V γ 9V δ 2 TCR repertoire (Davey et al, Nat Commun 2018, PMID: 29720665), but in parallel an adaptive-like subset (Davey et al, Nat Commun 2017, PMID: 28248310) that is highly diverse clonotypically, is entirely composed clonotype private to each individual, and lacks effector function upon initial thymic generation. These features and studies should be referred to at this point in the introduction.

>

We have revised our Introduction text to refer to these studies and we now write: “Studies in humans have highlighted the presence of both innate-like $\gamma\delta$ T-cell subsets with semi-invariant TCR clonotypes as well as adaptive-like $\gamma\delta$ T-cell subsets with larger TCR clonotypic diversity^{20,21}.”

2) The study’s results suggesting modest thymic involution over time, first mentioned in the introduction (Page 5, lines 1-2 and last paragraph before the Results), but again in the Results section (final paragraph before Discussion), appear to be at odds with those of a previous study (Emmrich et al, Reference 33) that concluded no involution took place. How this discrepancy can be resolved/explained should be addressed in the Discussion, which is currently lacking.

>

Our results regarding the age trajectory of NMR thymi are indeed at odds with those published by Emmrich et al. However, our data are more comprehensive in terms of the ages we included and our histopathological assessment of involution. Mainly due to space considerations we did not devote lengthy text for discussing the disagreement between our results and those published by Emmrich et al., but rather left it to the readers and time for determining what is the more likely true biology.

3) Regarding decreased α and β TCR V gene diversity relative to other mammals (eg final Results section; Discussion).

a) The decreased numbers of α and β TCR V regions observed in NMR relative to other mammals, which is indicative of most hystricomorph species, is suggested to reflect decreased evolutionary challenge from intracellular pathogens. However, the same TCR genes will be used for CD4 responses to mount responses to extracellular pathogens, so this interpretation is arguably flawed – could it not instead be interpreted as a decreased pathogenic challenge more generally ?

>

The reason for developing a single-cell approach for TCR-seq in a non mouse/human immune system, which covers both $\alpha\beta$ as well as $\gamma\delta$ TCRs, was exactly because the genomic phyletic patterns we observed for these TCR loci are insufficient for accurately determining the clonotypic diversity they give rise to. It is our TCR-seq data that allowed us to observe the low CD8/CD4 ratio of $\alpha\beta$ TCR clonotypic diversity (relative to that in the mouse), which corroborates other parts of our work, e.g., the low CD8/CD4 cell ratio. Our Results sections: “*Naked mole-rats harbor dominant $\gamma\delta$ T-cell clonotypes and the $\alpha\beta$ clonotypic diversity of their CD8 T cells is smaller than that of their CD4 T cells*” and “*Naked mole-rat CD8 and CD4 $\alpha\beta$ T-cell clonotypic diversities, MHC-I and MHC-II gene family sizes, and thymic involution*” explicitly address this point, and in our Discussion we also refer to it. As stated above and in many locations in our manuscript, we believe that the selective pressure imposed by intracellular pathogens has significantly relaxed and hence the immune arm that evolved against them, namely NK and CD8 T cells, and the *MHC-I* gene family, has lost much of its diversity and/or has not evolved to the extent observed in the mouse. There is no reason for us to believe and suggest that selective pressure imposed by extracellular pathogens has significantly relaxed and the data we produced do not support that.

b) The finding that other hystricomorpha species have similar immune gene expansion/contraction profiles to NMR but do not exhibit disproportionately long life-spans (in contrast to the guinea pig which does not conform to the profile of immune gene expansion/contraction seen in NMR) adds to the impression that it is challenging to infer too much from these kind of inter-species differences without additional data.

>

We do not think that NMR is unique in the hystricomorph group with respect to its long lifespan, and possibly also cancer resistance, of which lifespan is not independent.

Damaraland mole rats live 20 years and the mean lifespan of chinchillas is reportedly 17.2, although according to this Merck manual: <https://www.merckvetmanual.com/exotic->

and-laboratory-animals/rodents/chinchillas, it is actually 20 years, and in addition, postmortem examinations of several hundreds of chinchillas showed very low rates of neoplasia. Among the hystricomorph group it is actually the guinea pig that is the outlier, both in its shorter lifespan and its genomic phyletic patterns of immune gene families, which we mention in several locations of our work.

c) If one is convinced that the NMR has been faced with a decreased challenge from intracellular pathogens, then it arguably begs the question of what the $\gamma\delta$ T cells are present for? Is this for a residual role in intracellular pathogen defense in the circulatory/splenic niche, or a role in cancer immunosurveillance, in which case why would they be primarily circulatory/splenic in localisation?

>

The idea of decreased selective pressure imposed by intracellular pathogens is strongly supported by the loss of NK cells, NK-cell receptor genes, low *MHC-I* genomic diversity, and low CD8 T-cell clonotypic diversity and relative cell count. Given this observation, it is not clear to us why the circulating NMR $\gamma\delta$ T cells would have evolved against intracellular pathogens or retained some of that function which was previously carried out by NK and CD8 T cells. The fact that we observe the NMR $\gamma\delta$ T cells in the splenic niche and that they do not express the epithelial homing CD103 integrin (*Itgae*) does not exclude the possibility that they might still exist in the intraepithelial compartment. The functions that these NMR $\gamma\delta$ T cells have evolved to fulfill is still an open question.

4) Clonotypic analysis

The clonotypic analysis as set out in the figures is limited and I have several points to consider and address.

a) The manuscript equates (eg on page 11, first sentence of sixth Results section) V region number with clonotypic diversity, but the extent of N/P region addition is a major variable, independent of the V region number, that will determine repertoire diversity and that can differ between TCR loci (eg TCR- γ vs TCR- δ) and different $\gamma\delta$ subsets.

b) When clonotype data are presented, nucleotide sequences should be shown, and assignments to V, D (where applicable), J and also N/P nucleotides assigned where possible. This will help explain public CDR3 sequences, which are fairly short and would be expected to be largely germline-derived with minimal N/P additions. The current text in relation to this (page 12 line 5) is highly confusing ('where in both...sample') and should be clarified.

>

We define clonotypes as: C-region, V-region, D-region, J-region, and the CDR3 amino-acid sequence, to which the N/P additions contribute. We have now included in our supplementary tables 10 and 11 columns which specify the N/P sequences.

c) In relation to V region usage of expanded clonotypes, it is of interest that the sequencing revealed a prevalent sequence which aligns with mouse V γ 4. This could be a component of an invariant population analogous to the V γ 4V δ 5 mouse population (Kashani et al, 2016 Nature Comms), a population which produces IL-17. It is interesting that no NMR $\gamma\delta$ population appears to make IL-17 or express Rorgt, and also that no equivalent population to the V γ 6V δ 1 mouse population that produces IL-17 was observed in the spleen. As this mouse population is rare in the circulation and more prevalent in tissues such as lung and uterus, this again emphasises that the lack of analysis of NMR $\gamma\delta$ T cells in tissues is a shame.

>

In a single study we have single-cell sequenced three immune organs from two species, one of them in two age groups, as well as developed a single-cell TCR-seq approach for non-model organisms. Surveying $\gamma\delta$ T cells in peripheral tissues would probably add a lot, but fitting this in one single study, in terms of time, budget, and text space is a huge challenge. In addition, which peripheral tissues would have been sufficient? We believe that a single-cell atlas is required for this and should be conducted in a separate study.

D) The lack of Vdelta chain sequencing is highly unfortunate, particularly as TCR γ chain sequences generally tend to be more public (due to shorter CDR3 regions using only V-J gene segments and with fewer N/P additions) and this should be rectified. Importantly, the TCR-delta chain could potentially be highly diverse (owing to V-D-J gene segment usage and N nucleotide addition between each gene segment, indicating a semi-invariant subset if V γ sequences are public) or essentially invariant, and this is an important issue to resolve.

>

We have reanalyzed our data, this time using cells in which both TCR chains were captured, hence filtering cells with orphan chains as well as $\gamma\delta$ and $\alpha\beta$ T cells in which the D region was not identified (in the δ and β chains, respectively). While this reduced the number of analyzed cells, the results we reported still hold. Namely, we still observe the C γ 4, V γ 4-2, V γ 5-3, CDR3 YWDSNYAKKL clonotype, where it is mainly paired with V δ 1-4, D δ 3, J δ 2, CDR3 ALWELRTGGITAQLV. Other than that, our previous observations regarding the lower clonotypic diversity in NMR CD8 $\alpha\beta$ T cells as compared to NMR CD4 $\alpha\beta$ T cells as well as mouse CD8 $\alpha\beta$ T cells, still hold for these bichain TCR cells. These changes are updated throughout our revised text.

e) The significance of a TCR- γ clonotype being present only in one of the two $\gamma\delta$ T cell subsets is not discussed.

>

In our revised Discussion text we now write: “The fact that we detect a dominant and public NMR T-cell clonotype only in its cytotoxic $\gamma\delta$ T cells makes it tempting to speculate that this subset recognizes frequently encountered ligands that signal a threat which if not rapidly reacted against wreaks havoc, such as newly early malignancies and acute disruptions of homeostasis.”

f) Page 13 line 4 ‘Neither...’

This should reference evidence for adaptive-like biology and infection-driven clonal focussing in human $\gamma\delta$ T cells, in particular Ravens et al, Nat Immunol 2017, PMID: 28218745; Davey et al, Nat Commun 2017, PMID: 28248310; and Davey et al, Nat Commun 2018, PMID: 29720665.

>

We believe the reviewer was referring to the sentence: “Neither any of the $\gamma\delta$ subsets nor any of the $\alpha\beta$ subsets of the NMR T cells show a significant (multiple-hypotheses-adjusted $p < 0.05$) age-related change in any of the clonotypic diversity metrics (Fig. 3H, I).”. Due to our reanalysis of the clonotypic diversities, the evenness and rarity of the cytotoxic $\gamma\delta$ T-cell subset does increase with age, so that sentence has slightly changed in our revised text. Notwithstanding, we looked into the references noted by the reviewer:

- Ravens et al, Nat Immunol 2017, PMID: 28218745 reports about dynamics of $\gamma\delta$ T-cell clonotypes before and after hematopoietic stem-cell transplantation in a prospective human cohort. It does mention that in healthy adults $\gamma\delta$ T-cell clonotypes are stable over time, but that does not seem relevant enough to our results, previous and updated.
- Davey et al, Nat Commun 2017, PMID 28248310 reports about the clonotypic diversity of human $V\delta 2+$ and $V\delta 2-$ TCRs with many conclusions of which we were not able to understand what may relate to our results, previous and updated.
- Davey et al, Nat Commun 2018, PMID: 29720665 further characterizes human $V\delta 2+$ clonotypes reporting a $V\gamma 9+V\delta 2+$ population with equivalent clonotypes in cord and adult blood. Again, we fail to understand how that may relate to our results, previous and updated.

Therefore, unfortunately we are not able to use these references in our revised text.

g) On page 14 (penultimate sentence before final Results section, starting 'By contrast...'), the decreased CD8 diversity in NMR vs mice is employed to suggest a decreased selection pressure from intracellular pathogens; conversely NMR have higher CD4 diversity than mice, therefore is it not logical to suggest an increased selection pressure from extracellular pathogens compared to mice ?

>

The clonotypic diversity of NMR CD8 T cells is mainly smaller than that of the mouse naive CD8 T cells, in both age groups, and modestly so vs. the mouse memory CD8 T cells among the old-age samples. The situation among the CD4 T cells is different, where only among old-age samples the diversity of the NMR CD4 T cells is larger than that of the mouse CD4 T cells, more so in the mouse memory CD4 T cells. In other words, the difference in CD4 clonotypic diversity is mainly an old-age observation and for this reason we avoided overinterpreting it, especially since selective pressure is predicted to be stronger at younger ages.

h) Given recent findings highlighting butyrophilin family proteins as direct ligands for germline-encoded regions of particular V γ chains (eg Willcox et al, Immunity, 2019, PMID: 31628053; Karunakaran et al, Immunity, 2020, PMID: 32155411), it would be interesting to compare the number and nature of butyrophilin family member genes in relation to V γ genes in the NMR compared to eg mice and human, as it is possible these have co-evolved alongside each other, and BTN/BTNL gene products could be involved in regulation of NMR $\gamma\delta$ T cell functionality.

>

This is a very good point. Analyzing the phyletic patterns of the range of possible $\gamma\delta$ TCR ligands, which in addition to the *BTN/BTNL* gene family includes others such as the *CD1* and *MR1 MHC-I-like* gene families. Unfortunately, these gene families do not have unique domains as is the case of the TCR variable regions and the *MHC* gene families.

Therefore, their phyletic patterns cannot be reliably constructed.

5. Evolutionary arguments

Some of the arguments in the Discussion regarding evolutionary aspects are confusing or require clarification.

a) On page 17 (sentence starting 'Although intracellular pathogens...', in relation to intracellular pathogens exhibiting strong selective pressure, the authors speculate that the 'subterranean ecological niche' could present 'an evolutionary dead end'. The meaning of this is unclear –

presumably a dead end (ie less viable) for intracellular pathogens ? This should be rephrased to increase clarity.

>

We have clarified this part in our revised Discussion text and we now write: “In contrast, the small genomic diversity of NMR α and β variable TCR regions, the smaller proportion of the CD8 relative to the CD4 $\alpha\beta$ T-cell subset in the NMR spleen along with a corresponding bias in the clonotypic diversities of these NMR $\alpha\beta$ T-cell subsets, and the smaller size of the NMR *MHC-I* relative to its *MHC-II* gene family, corroborate the loss of NMR NK cells⁴². These observations are all consistent with the hypothesis that this occurred as a result of relaxed selective pressure imposed by intracellular pathogens, albeit evidence of high susceptibility of NMRs to such insults is still limited⁷². Although intracellular pathogens are thought to be one of the most dominant selective forces in mammalian evolution⁷³, the subterranean ecological niche, which NMRs occupy, is likely an evolutionary dead end for intracellular pathogens due to its limiting effect on infectivity. By contrast, bats, which harbor more virus species than any other mammal, occupy a rich ecological niche⁷⁴.”

b) In the next sentence, the authors make the linked point that this ‘allowed the selective pressure under which NMRs have evolved to shift towards elimination of early malignancies and maintenance of tissue homeostasis, thereby promoting a long reproductive lifespan, to which their immune system has responded by utilizing $\gamma\delta$ T cells.’

This argument seems to suggest that in most mammals, evolution couldn't work on tackling early malignancies and promoting long lifespan because it is 'too busy' tackling pathogens. But evolution is a multitasker. This seems either a bad explanation or a bad idea or both. A more acceptable rephrasing of the authors' argument could be that adaptations that help the immune system to tackle pathogens might simultaneously compromise the immune system's ability to tackle cancer and that the NMR represents a more mechanistically pure form of immune configuration that is more focussed on malignancy/lifespan. Arguably a more fundamental but important question is why evolutionarily it makes sense for NMRs to have a longer life-span ? Do they meet/mate infrequently ? Grandparental importance ? This is not discussed.

>

Extrinsic risk is thought to be the principal driver in the evolution of longevity (Evolution of Ageing, Kirkwood TBL, Mechanisms of Ageing and Development, 2002; Comparative aging and life histories in mammals, Austad SN, Exp. Gerontol, 1997; Why do we age? Kirkwood TBL and Austad SN, Nature, 2000), hence NMR lifespan and healthspan most likely evolved due to the low extrinsic risks in their subterranean ecological niche, such as

reductions in predation, starvation, climatic extremes, and most relevant to our point intracellular pathogens. In species which are exposed to strong intracellular pathogenic selective pressure (in addition to other high extrinsic risks), such as mice and rats, the selective pressure imposed by early malignancies is probably too weak for the immune system to have evolved mechanisms to suppress it, rather than 'too busy'. More generally, although evolution is a 'multitasker', species evolve under resource-limiting constraints and hence the presence of tradeoffs act as a limit to adaptations. Our point in the Discussion tries to integrate that in a compact way. The idea that adaptations that help the immune system to tackle pathogens might simultaneously compromise the immune system's ability to tackle cancer, is likely an indirect consequence of evolutionary tradeoffs: the selective pressure to eliminate sporadic tumors in late life dwarfs in comparison to the selective pressure for defending against highly infectious pathogens throughout all phases of life.

c) Regarding the potential increased role of $\gamma\delta$ T cells in cancer resistance in NMRs, it seems surprising that such a role would link with a more haematological localisation for the $\gamma\delta$ compartment. Could the authors comment on this ? Given that, at least in humans, 95% of cancers are non-haematological in origin, this preferential circulatory localisation does not exactly emphasise a role in cancer suppression. From this perspective, it is unfortunate the authors did not examine the TCR repertoire and $\gamma\delta$ numbers in epithelial tissue, particularly as $\gamma\delta$ T cells in other species do generally adopt a tissue localisation (eg gut, skin, etc).

>

Circulating immune cells monitor the entire body through their ability to extravasate to locations which are sending stress signals. This is likely why NK cells, which have evolved to detect and immediately destroy cells infected with intracellular pathogens, circulate in high fractions. That said, we do not know if in addition to their presence in the circulation NMRs have high fractions of $\gamma\delta$ T cells in intraepithelial tissues, which will have to be the goal of future studies.

Minor comments

(i) In the Introduction, line 6: the wording is misleading regarding the innate component, which one does not usually consider recognising MHC molecules. This could be rewritten to avoid this confusion.

>

MHC-I are the ligands of NK-cell receptors and NK cells are a critical component of the innate immune system. We do not understand how our wording is misleading.

(ii) On Page 7, line 4, what is the basis of suggesting the non-cytotoxic $\gamma\delta$ T cell subset may be analogous to the mouse IL-17 producing cells ? This was unclear to this reviewer.

>

Since a large proportion of the mouse $\gamma\delta$ T cells commit to either IFN- γ or IL-17 producing cells (upon stimulation) and the cytotoxic $\gamma\delta$ T-cell subset of the NMR has transcriptional resemblance to mouse IFN- γ producing $\gamma\delta$ T cells, we saw it fit to address whether the cytotoxic and non-cytotoxic $\gamma\delta$ T-cell subsets of the NMR might be homologous to the IFN- γ or IL-17 producing $\gamma\delta$ T-cell subsets of the mouse. We clarified our revised Results text in that location and now write: "Hence, these data do not allow determining with high certainty whether the cytotoxic and non-cytotoxic $\gamma\delta$ T-cell subsets in the NMR spleen are homologous to the tissue-resident IFN- γ and IL-17 producing $\gamma\delta$ T-cell subsets of the mouse."

(iii) Page 13, 4th line from the bottom: there is a type with 'expectedmogilenko' – mogilenko is first author of publication number 43.

>

We apologize for this bibliography-tool error. It is fixed in our revised text.

(iv) a period is required on Page 18 line 6 of the middle paragraph after 'activation'

>

We apologize for this typo. It is fixed in our revised text.

(v) the text should read 'diversities OF their...' on Page 19, second line of the bottom paragraph

>

We apologize for this typo. It is fixed in our revised text.

(v) 'steppingstone' should be two words on the 3rd line of the last paragraph.

>

We apologize for this typo. It is fixed in our revised text.

(vi) regarding thymus involution on Page 19, the meaning of one sentence is very unclear: 'Based on that, the most striking differences...become intuitive'. What does this mean? This should be reworded to increase clarity.

>

We corrected that sentence and in our revised Discussion text we now write: "Based on the much larger potential diversity of mouse thymocyte clonotypes, the most striking differences we observe between the age trajectories of mouse and NMR thymi, namely their disparate early-life weights and subsequent rates of decline, become intuitive."

R#3

Review of article submitted to Nature Communications

Overall Critique.

The observations reported come from genetic and transcriptome analyses and are most interesting and the methods employed appear to be state-of-the-art. In general, the conclusions reflect the genetic data. Extrapolating this to pathogen levels without data is a weakness. The study may have relevance to understanding NK and T cell evolution, the control of intracellular pathogens and perhaps, malignant cells. The presentation could be improved by:

(a) Reducing redundancy in the abstract and introduction.

>

Our Introduction included a paragraph which most likely accounted for that redundancy (starting with "*Cellular adaptive immunity*"). We apologize for this redundancy and have removed that paragraph in our revised Introduction and have also reduced some redundancy in our revised Abstract.

(b) Better appreciating that other mammals that do not live in subterranean burrows, also have an abundance of g/d T cells. This should have been acknowledged in the introduction.

>

In our Introduction we focus on introducing the various T-cell subsets mostly based on their known human/mouse functions and do not mention that NMR is a subterranean mammal, thinking it would be too esoteric at that point. We therefore left it to the Discussion.

(c) Clarifying that the study is solely based on genetics either comparing genome or using transcriptomes analysis, but without any evidence that NMR have fewer intracellular pathogens.

>

We have acknowledged that in our revised Discussion text, where we now write: "In contrast, the small genomic diversity of NMR α and β variable TCR regions, the smaller proportion of the CD8 relative to the CD4 $\alpha\beta$ T-cell subset in the NMR spleen along with a corresponding bias in the clonotypic diversities of these NMR $\alpha\beta$ T-cell subsets, and the smaller size of the NMR *MHC-I* relative to its *MHC-II* gene family, corroborate the loss of NMR NK cells⁴². These observations are all consistent with the hypothesis that this occurred as a result of relaxed selective pressure imposed by intracellular pathogens, albeit evidence of high susceptibility of NMRs to such insults is still limited⁷². Although intracellular pathogens are thought to be one of the most dominant selective forces in

mammalian evolution⁷³, the subterranean ecological niche, which NMRs occupy, is likely an evolutionary dead end for intracellular pathogens due to its limiting effect on infectivity. By contrast, bats, which harbor more virus species than any other mammal, occupy a rich ecological niche⁷⁴.”

(d) Assuming that readers do not need a review of basic immunology, so they should immediately focus on RMR and the purpose of the study (see specific comments # 2 & 4).

>

Our revised Introduction and Abstract have been revised and shortened (see response to comment a). That said, for the diverse readership of Nature Communications we believe that being inclusive by providing the immunology terminology that is used throughout the Results and Discussion is preferable.

(e) Confine their speculation and hypotheses to the discussion and minimize speculation about issues for which they provide no data.

>

In all the paragraphs of the Results section we have removed the summary sentences which might have created the notion of speculation.

Specific comments

1. Condenses the general speculative aspects of the abstract and instead provides more specific details as to the outcome of the study.

>

This has been addressed in the responses above.

2. Page 3 and the first two paragraphs on page 4 are unnecessary unless the audience has no immunological background. Also, some details are wrong, since g/d T cells are abundant in circulation of pigs and cattle and not confined to peripheral /mucosal tissues. Authors should make clear they are comparing RMD to conventional lab mice and rats, not all other species.

>

This has been addressed in the responses above.

3. Start the Introduction with the last paragraph on page 4.

>

This has been addressed in the responses above.

4. Page 5, second paragraph. Extending their comparative study beyond lab mice and RMD in future would be wise and should include other species with high levels of g/d T cells such as pigs and cattle.

>

We have acknowledged this in our revised Discussion text where we now write: “While these studies afford some insights into $\gamma\delta$ T-cell function, we do not know if and how they may relate to cancer resistance, maintenance of homeostasis, and lifespan. This is because our understanding of those processes in these farm animals is limited and may have been affected by the domestication⁷⁹, albeit it would be interesting to profile the immune-cell repertoires and T-cell clonotypic diversities in these species.”

5. The section on T cell phenotypes ([5-8]) is better judged by a reviewer who works in this area.

6. Page 8. Are there other criteria that can be used to distinguish a lymph node from a thymic tissue? B cells follicles in thymus were reported 50 years ago and in swine, thymocytes continue to rearrange Ig genes. Authors would consider this since they base their data on sc RNA (transcriptomes).

>

The histological features of cortex/medulla and the unique feature of Hassall’s corpuscles are the gold standard for distinguishing the thymus from other lymphoid organs. We should also note that Emmrich et al. have published that the NMR cervical thymic lobe is a T-cell-specific lymph node in their 2019 bioRxiv preprint yet in their actual 2021 Aging Cell publication referred to is as an ectopic cervical thymus without referring to their 2019 bioRxiv preprint.

7. “SUMMARY et al” This is pure speculation without evidence that the lifestyle of NMR exposes them to fewer intracellular pathogens. Authors should provide comparative data to support their view. Perhaps there are published data that can be cited.

>

This has been addressed in the responses to comment c above.

Reviewer #3 (Remarks to the Author):

In the paper “Evolution of T cells in the cancer-resistant naked mole-rat” authors show that Naked Mole Rat (NMR) display a splenic (circulating) population of $\gamma\delta$ T that can be divided into cytotoxic and non-cytotoxic subsets. $\gamma\delta$ T cells can be identified in NMR Bone Marrow (BM) and thymus/cervical lymphnode as well. NMR thymic $\gamma\delta$ T cells do not express ITGAE (as opposed to mouse) possibly akin to their circulating behaviour as opposed to the tissue residency observed in mouse. In addition, authors show a disproportionately large diversity of $\gamma\delta$ V genes as opposed to $\alpha\beta$ V in NMR together with a reduced diversity of MHC-I gene repertoires, suggesting a preferential exposure to non-MHCI ligands due to a relaxed selective pressure against intracellular pathogens. Finally, authors highlight that $\gamma\delta$ T cells show lower clonotype diversity as compared to $\alpha\beta$ T cell in NMR and higher level of clonotype sharing across subjects and that cytotoxic $\gamma\delta$ T cells show lower diversity (clonal expansion) when compared to non-cytotoxic $\gamma\delta$ T cells

To our knowledge, this is the first report to dissect at such level of resolution the adaptive immune system of NMR (contrasted with the more characterized mouse immune system). Given the characteristic features of NMR, namely the longer lifespan and cancer resistance, as reported by the authors, we believe the work can be of significance in the field of comparative genomics and, with additional work further dissecting the carcinogenetic process in NMR, of marginal interest in the field of cancer biology and cancer immunology.

The claim that in NMR exists a circulating $\gamma\delta$ T cell population, though supported only by scRNAseq data, looks solid. Additional evidences as provided by flow cytometry technology would further corroborate such claim although we realize that high quality antibodies may not be available for organisms under consideration. The claim that “NMR have evolved under a relaxed intracellular pathogenic selective pressure” is adequately supported and stands as a valid theory.

The annotation of a cellular subset in the splenic T cell dataset as ILC3 is based on the absence of expression of Cd3 isoforms and on the expression of Rorc and is further confirmed later on by the absence of reconstructed Tcr sequences. Nonetheless, the heatmap displayed in Fig1D suggests the existence of a set of genes characteristic for this population. It would be useful to explore whether those genes support the suggested annotation, also considering that Rorc expression does not appear to be sufficient for lineage specification as it appears expressed at similar level also in cytotoxic and non-cytotoxic $\gamma\delta$ T cells. Furthermore, these cells were identified only in the splenic compartment but not in BM or thymus, thus giving rise to the question as to what is the origin and role of these cells in NMR.

>

We thank the reviewer for pointing this out. According to Mazzurana et al. (Cell Research, 2021), He et al. (Hepatology, 2022), and Song et al. (Frontier Immunology, 2023) who annotated ILCs, expression of *Cd127 (I17r)* defines ILCs. In our data, *I17r* is expressed in what we annotated as ILC3 cells as well as in the non-cytotoxic $\gamma\delta$ T-cell subset, which

can be distinguished by the expression of the CD3 encoding genes (*Cd3d*, *Cd3e*, *Cd3g*, and *Cd247*). What we annotated as the ILC3 subset additionally expresses *Trdc*, which was also reported by Mazzurana et al. (Cell Research, 2021), who labeled that subset as immature ILC1. Since what we annotated as ILC3 also expresses ILC2 markers, including *Gata3*, *Maf*, and *Ptgdr2*, we have revised our text and figures and relabeled it as ILC since we cannot confidently distinguish which ILC subset it might be. We have also added the *Rorc* gene to the heatmaps in Fig 1 C, F, I, and L and to Fig S1 C, F, I, and L. The splenic T-cell dataset is the only dataset that was both profiled using the more advanced 10x V3 chemistry and included aged NMRs (~24 years old). Either of these two variables or both might explain why this cell type is not observed in the bone marrow dataset. As for the thymus, we do not expect to detect these cells in that tissue since they are CD3 negative.

In figure S1D, relative proportions of identified cell types is reported for animals at different stages of life. The main text referring to this panel reads “both cytotoxic and non-cytotoxic gdT-cell subsets are overrepresented among the old age samples”. While this statement may be considered true for the cytotoxic subset, for the non-cytotoxic subset data appear to be skewed by one single old age animal, while the others show similar or even lower proportions of these cells as compared to young animals. In general, this statement may benefit from a stronger statistical evaluation. Given the origin of data as scRNAseq, a tool that may prove helpful in this is MILO.

>

We followed this comment through and applied a formal statistical approach to test for age differences in the proportions of cells in each cluster. However, MILO is not the right tool for this task because it was developed for data where transitions between cell types/states follow a pseudotime continuous trajectory, which is not the case in our splenic T-cell data. Accordingly, applying MILO to the NMR splenic T-cell dataset does not indicate significant age effects, even in the ILC subset, where the age effect is very apparent in Fig S1D. The figure below is produced by MILO and shows the distribution of its estimated age-effect sizes, where a strong age bias is not apparent in the ILC subset:

We therefore chose to fit a multinomial logit random effects model for this task, using the `mblogit` function implemented in the `mclgit` R package, specifying the cell subset label as the response, age as the fixed effect, and sample as the random effect (i.e., $\text{cell-type} \sim \text{age} + 1|\text{sample}$), where we created an artificial baseline cell-type category which is the mean across all ages and samples. Hence, using this baseline we are estimating the age effect on cell-type count relative to the mean cell-type count per each age and sample. We have added this part to our revised Methods text, and the p -values obtained using this model to our revised Results text, where the old-age bias in the cytotoxic $\gamma\delta$ T-cell subset comes out significant, as well as the adult-age bias in the CD8 T-cell subset (both multiple-hypotheses-adjusted $p \ll 0.05$).

In Figure1, different tissues are analyzed by scRNAseq (Spleen, Bone Marrow and Thymus). Contextually, other covariates are evaluated (Age in the Spleen dataset, Sex in Bone Marrow and Thymus datasets) but no comment is made as to why in different samples different covariates were considered, in particular it would be important to assess age effects in cellular composition of the thymus considering subsequent analysis performed to evaluate the development of this organ during life span.

>

In all datasets, leave the splenic T cells, we included both sexes in order to avoid obtaining sex-specific results. In the case of the splenic T-cell dataset, we did not have enough ~24 year-old females to include hence had to settle for only using males. Although it would be interesting to have thymus scRNA-seq data from old-age animals,

analyzing age effects on cellular composition in this tissue (and we imagine the reviewer is referring to the thymic epithelial cells), would be very specific to thymic involution and hence slightly outside the T-cell focus of our study, not mention adding more text to what is already close to the word-count limit. Notwithstanding, the old-age splenic T-cell dataset does provide some insight into the effects of thymic involution, as these T cells represent the output of the old-age thymus.

In Figure 1G/H are reported 2D embeddings of BM derived scRNAseq datasets for NMR and mouse. In the main text referring these panels, they are compared and deemed similar for cellular composition and differentiation trajectories but no specific analysis is performed to evaluate trajectories and UMAP embeddings (though able to partially capture developmental processes) are not designed specifically for this task. I would suggest either rephrasing the text, or adding specific analysis (using DiffMap, Monocle, or similar tools). The same goes for panels J/K where schematic developmental trajectories appear to be handwritten, without any further support.

>

Unfortunately, such tools do not always provide an additional value to a UMAP embedding. Using the slingshot R package, applying its `getLineages` function by specifying the hematopoietic stem-cell subset as the starting point (center of the UMAP figures below, which are what we used in Fig. 1G, H) and the most differentiated subsets (e.g., neutrophils, RBCs, monocytes) as end points, we get this result for the bone marrow datasets (NMR left, mouse right):

We also applied slingshot's `getCurves` function, which uses an unsupervised approach, and got this result for the bone marrow datasets (NMR left, mouse right):

We additionally applied the diffMap + PAGA approach, implemented in the scanpy Python library, and got this result for the bone marrow datasets (NMR left, mouse right; top row are the diffMap-guided PAGA UMAP embeddings and bottom row are the diffMap dpt graphs):

Similarly, for the thymus datasets, in which the differentiation trajectory is much more obvious and simpler, slingshot getLineages produces this result (NMR left, mouse right, using the UMAP embeddings in Fig. 1J, K):

And slingshot getCurves produces this result (NMR left, mouse right):

Accordingly, diffMap + PAGA produces this result (NMR left, mouse right, top row diffMap-guided PAGA UMAP embeddings. and bottom row diffMap dpt graphs):

In conclusion, given that neither slingshot options are able to capture the simple trajectory in the thymus, and that the diffMap+PAGA embeddings come out very similar to the UMAP embeddings we have in Fig. 1J, K, where the diffMap dpt graphs do not seem to correctly capture the expected trajectories, we found that these approaches do not add any valuable information to the UMAP embeddings we have in Fig. 1, which already do a reasonable job for the simple purpose of presenting cell-type repertoires rather than delving deeply into the differential differentiation trajectories between NMR and mouse.

The statement according to which: “while mouse thymic $\gamma\delta$ T cells show high and ubiquitous expression levels of *Itgae* [...], NMR thymic $\gamma\delta$ T cells show a far lower and limited *Itgae* expression pattern” does not appear to be sufficiently supported. In figure S1L, *Itgae* average normalized expression is comparable between mentioned cellular subset and the same goes for the fraction of cells effectively expressing the gene (in both cases apparently below 10% of the subset). More robust statistical analysis is required to state that expression levels differ between cell populations for the gene in analysis, let alone to leverage this observation as a mechanistic explanation for an observed phenomenon.

>

We thank the reviewer for bringing up this point. We added the violin plot below as Fig. 1M to provide a more intuitive representation of this claim:

We think this strongly supports the claim we make in the text about the mouse $\gamma\delta$ T-cell subset showing high and ubiquitous levels of *Itgae*. Nevertheless, we revised our Results text and now write: “Among the mouse thymic T-cell subsets, the $\gamma\delta$ T cells show the highest and most ubiquitous expression levels of *Itgae*, one of the two genes encoding the CD103 integrin that homes mouse and human $\gamma\delta$ T cells to their epithelial target tissues upon their egress of the thymus⁵³ (Figs. 1L, M and S1L; Table S1). By contrast, NMR thymic $\gamma\delta$ T cells show a low and limited *Itgae* expression pattern (Figs. 1L, M and S1L; Table S1), supporting our findings of a large proportion of NMR splenic $\gamma\delta$ T cells.”

Any statistical test one would apply will give a p-value of ~0 for this difference in *Itgae* expression levels, and not that we are opposed to statistical modeling, but in this specific case we think the figure conveys this result strongly enough.

In Figure 1Q it is unclear what the numbers reported over the bars stand for, especially considering that values do not match with provided axis. In addition, it is claimed that Cd8b gene has evolved “under relaxed purifying selective pressure relative to murine” but little is said about Cd4 which shows a similar trend and no table with statistical test results is provided.

>

We think this comment is referring to Fig S1Q. We apologize for not including a description for these values in the figure legend. These are the multiple-hypothesis-adjusted p values of the test for hystricomorph vs. muroid relaxation of purifying selection, which therefore reversely correspond to the Y-axis (although these are not the Y-axis values themselves). We have revised Fig. S1Q’s legend and now write: “Quantification of the intensity of purifying selection operating on (P) hystricomorph relative to muroid (Q) T-

cell co-receptor genes, revealing the significant relaxation of purifying selection in the hystricomorph *Cd8b*, where the values on top of the bars are multiple-hypothesis-adjusted p values of the test for hystricomorph vs. muroid relaxation of purifying selection.”

We also do refer to *Cd4*'s trend in our Results text, although it is much weaker than *Cd8b*'s.

In general, in figure1, panels C/F/I/L representing heatmaps of gene expression levels across individual cells in different datasets are difficult to read and representations that take into consideration a smaller number of genes (the ones most widely discussed in the main text for example) could make easier following the discourse.

>

The challenge with heatmaps is to limit the number of genes that are included such that the heatmap is easy to follow and refer to, on the one hand, yet maintains the clustering signal, on the other hand. For the number of cells and their gene markers in these datasets, and to be as consistent as possible throughout the four heatmaps in Fig. 1, with respect to genes, we found that our selection of genes struck at/near that optimum. We think that the dot heatmaps in Fig. S1 provide an additional easier/cleaner visualization to follow our discourse.

In its current form, figure S2C is not easily readable. It should also be noted that, although NMR show positive $\#Vg+\#Vd$ residulas, they fall within the confidence interval of the regression line, taking away strength to the claim that the observed relative increase in the number of genes for gdV regions is the result of selective pressure.

>

We apologize that the Fig. S2C came out grainy and difficult to read. We have replaced it with an improved version that is easier to read. In our text we only mentioned that the NMR's $\#V\gamma+\#V\delta$ is ~18-fold larger than what is expected from its $\#V\alpha+\#V\beta$. This places it at 1.46 SD units above the mean (fitted line). While it does fall within the 95% CI, it precisely lies on the 93rd percentile, which is still indicative of directional selection.

Globally, Figure3 is too cluttered and difficult to follow. My suggestion would be to consider moving panels H/I/S/Q/R to a new figure more focused on statistical inference and reformat them to increase readability. In addition, since the function used for statistical testing (mcpHill) relies on bootstrapping, it would be important to also report the number of bootstrap iterations. Furthermore, heatmaps in panels D/G/M/P do not seem the most efficient way to convey

desired message and could be substituted with plots as produced by the clonotype_network function from the plotting submodule of the scirpy package.

>

We thank the reviewer for bring up this point and have implemented it, and in our revised Fig. 3 we have replaced the bar plot, histogram, and heatmaps for each of the clonotype groups in each species (B-D and E-G for the NMR $\gamma\delta$ and $\alpha\beta$ clonotypes, respectively, and K-M and N-P for the mouse $\gamma\delta$ and $\alpha\beta$ clonotypes, respectively), with a single figure for each clonotype group - a clonotype network graph following scirpy's practices:

We believe that this achieved the desired de-cluttering of this figure. We also revised our Methods text to describe our usage of scirpy and added the number of bootstrap replications we used with the mcPill function (the default 5,000).

Authors dedicate a lot of words speculating on the phenotype of cells associated with the observed public clonotype in cytotoxic $\gamma\delta$ T cells based on their previous phylogenetic reconstruction but nothing is said with regards to the observed molecular phenotype of these cells as assessed in their scRNAseq dataset.

>

We have revised our Results text to add more important genes marked by the NMR cytotoxic $\gamma\delta$ T-cell subset, and now write: (3) a subset marked by high expression levels of the γ and δ constant TCR regions, the *Gzma*, *Nkg7*, and *Xcl1* cytotoxicity-related genes, the *Il2rb* IL-2 induced proliferation gene, the *Klra1* and *Klrd1* cytotoxicity inhibitory genes, and of *Cd8a* (yet absent in expression of *Cd8b*), labeled as cytotoxic $\gamma\delta$ T cells;” and also revised our Discussion text to refer to that, where we now write: “Our investigation of NMR T cells has uncovered several novel intriguing findings. NMRs have a splenic population of $\gamma\delta$ T cells at roughly the same proportion as that of mouse splenic NK cells, comprising two subsets. One of these $\gamma\delta$ T-cell subsets expresses an inhibited cytotoxic molecular profile homologous to that of mouse splenic NK cells, suggesting the two cell types are functionally similar yet are likely to differ in their activation mechanisms.”

In general, methods employed are in line with current standards and in particular, great effort has been poured into curating and integrating multiple annotations for a large number of species. Assumptions made are reasonable and where possible, adequate controls have been proposed, yielding results that may prove useful in refining annotation for lesser characterized organisms like NMR. Commendable is also the experimental and bioinformatic work displayed for reconstructing and analysing TCR sequences.

Methods are generally well documented. Nonetheless, we believe that sharing original scripts, possibly organized in a reproducible pipeline with workflow management systems like SnakeMake or NextFlow, would greatly improve reproducibility and readability.

>

We have followed the guidelines of Nature Communications and have uploaded scripts to zenodo (record 8384311), which is now referred to from our revised text under the “Code availability” section.

REVIEWER COMMENTS

Reviewer #1 (Remarks to the Author):

The authors have generally tried to address most concerns, however a few important points remain where this has not been done adequately.

1) Introduction

a) In relation to my 'Major Criticism 1' despite explicitly pointing out that the following sentence is 'severely flawed' (second paragraph of Introduction) this has been retained in unaltered form.

'Unlike $\alpha\beta$ T cells, $\gamma\delta$ T cells are not restricted to MHC-I-presented peptide antigens but rather, their less diverse $\gamma\delta$ TCR clonotypes are preprogrammed during thymic development to recognize a broad and ubiquitous set of both self and foreign stress-ligands, activating a rapid response^{16–18}.'

Just to be explicit, by 'severely flawed' I mean unacceptably so, thus requiring alteration. All the authors have done is insert references to studies that highlight innate-like and adaptive human subsets. However, retained as it is in its current unaltered form, it perpetuates misconceptions about the human $\gamma\delta$ T cell repertoire, as the sentence does not specify species, and it is objectively incorrect regarding human $\gamma\delta$ T cells. The comment about 'less diverse $\gamma\delta$ TCR clonotypes' that are 'preprogrammed during thymic development' applies well to the human innate-like V γ 9V δ 2 T cell subset focussed on a restricted number of (butyrophilin-family) ligands, whereas adaptive-like human $\gamma\delta$ T cells have an extremely diverse repertoire and are generated as TCR-diverse naïve cells, that are not pre-programmed in terms of effector function, and likely recognise diverse ligands. Also, from a ligand recognition perspective, the idea the current sentence provides, that decreased diversity in the $\gamma\delta$ T cell repertoire relative to $\alpha\beta$ T cells links with a much more diverse range of ligands, makes no sense. While (inconsistent with their earlier flawed sentence) the authors state later that adaptive-like human $\gamma\delta$ T cells are more diverse, this doesn't alter the fact that this first sentence is inaccurate, such that if a Masters student came out with it they would be marked down. Nature Communications deserves a little better.

Short of rewriting the current text, I suggest two solutions: option a) restrict the scope of this sentence to innate-like $\gamma\delta$ T cell populations that are pre-programmed for some kind of effector function (into which human innate-like V γ 9V δ 2 T cells would actually fit), and then later highlight TCR-diverse, adaptive-like $\gamma\delta$ T cells generated in naïve form but that can acquire effector function subsequently in response to eg infectious challenge. Option b) include both aspects of biology in this first sentence, ie both innate-like/restricted TCR repertoire/pre-programmed effector functionality vs adaptive/highly diverse repertoire/naïve functionality initially. I think Option a) might be less disruptive for the authors' current narrative.

b) In alignment with Reviewer 3, I agree the 'immunological primer' present in the opening paragraph of the Introduction should be largely cut and the main narrative should begin earlier. This will also help condense the paper marginally. This opening paragraph is also not written terribly well – two sentences jar from an immunological angle.

Sentence 1:

'Lymphocytes provide both innate and adaptive immunity through the ability of their receptors to sense malignancies presented by their major histocompatibility complex (MHC) ligands.'

This is very much a stretch vis a vis innate lymphocyte recognition and is not a great sentence. Even if one argues that NK receptor recognition of MHC molecules permits recognition of tumours via missing self signatures on tumours exhibiting MHC loss, it is the integration of multiple signals from diverse receptors, some not recognising MHC molecules, that permits sensing of malignancies, rather than direct sensing of malignancies by MHC-focussed receptors, as implied.

Sentence 2: 'CD8 T cells have evolved to eradicate cells infected by intracellular pathogens and early cancer transformed cells through the ability of their $\alpha\beta$ T-cell receptors (TCRs) to recognize

intracellularly-produced foreign peptides presented on MHC-I of the affected cells¹⁻⁶.
As written this slightly lazily seems to connect CD8 T cell TCR recognition of foreign peptides presented on MHC-I of affected cells with eradication of 'early cancer transformed cells' – which is confusing and wrong.

Overall I would cut this section.

2) In relation to Major Criticism 2, ie the inconsistencies of the results of this study on thymic involution with previous studies in this area, the authors shy away from discussing this directly. I have two issues with this.

a) firstly, in the Introduction the narrative on this is highly confusing for a reader.

In paragraph 3 of the Introduction, the authors state in relation to previous NMR-related immunology research:

'in contrast to most vertebrates in which thymic involution is apparent by puberty⁴⁴, NMR thymi do not display signs of involution⁴⁰ even at ages twenty-fold greater than their age of sexual maturity²⁷'

And then in the final paragraph where they summarise their results they highlight that thymic involution is observed similar to mice.

'Consistent with that, we also observe that early-life NMR thymi are considerably smaller than those of mice, yet undergo similar involution progression, which is already apparent in young adults.'

This is highly confusing for readers. I appreciate the results are contradictory however given that, does it not make sense to caveat the previous work with eg 'IT HAS BEEN REPORTED that NMR thymi do not display signs of involution...'. Also, there is no acknowledgement of the contradiction in their final summary point in the Intro and this could be addressed 'also observe, CONTRARY TO PREVIOUS STUDIES, that early-life...'. Otherwise the understanding of the reader is sacrificed in a confusing narrative.

b) secondly, regarding the request to discuss the contrary findings, the authors' riposte ('Mainly due to space considerations we did not devote lengthy text for discussing the disagreement between our results and those published by Emmrich et al., but rather left it to the readers and time for determining what is the more likely true biology.') is in my view weak. The idea that readers are going to have time to check out the technical vagaries of different approaches to assessment of thymic involution in the naked mole rat is somewhat wishful thinking. More importantly, as scientists, it is in my view not an unreasonable expectation of us to integrate our results with comparable datasets in the field. The authors are clearly best placed in the field to make this comparison, and also clearly have reasons to think their approach is superior, which sound reasonable (assessment of thymi over a wider life-span, and more comprehensive histological assessment, which sound reasonable). There could be a single/two-sentence addition to the Discussion, and this would be more than compensated by curtailing their immunological primer section at the start of the Introduction.

3) Evolutionary Arguments

This relates to query 5 in my original review, and specifically point b, regarding the text 'allowed the selective pressure under which NMRs have evolved to shift towards elimination of early malignancies and maintenance of tissue homeostasis, thereby promoting a long reproductive lifespan, to which their immune system has responded by utilizing $\gamma\delta$ T cells.'

The authors essentially agree with my suggestion that a better phrasing of the argument is around the idea that adaptations to pathogen-focussed immunity might compromise evolution of more potent anti-tumour immunity. However, the current wording in the manuscript still seems to be phrased around 'selective pressure shifting' towards cancer immunosurveillance, whereas the

compromise argument is about a more focussed/adaptation of the response (immunological adaptation) to a consistent selective pressure around cancer development. This is a subtle but important distinction, and for the reader might make the difference between appearing to state that 'evolution has now decided to tackle cancer' to a more valid point (as outlined nicely in the author's riposte to my point) about trade-offs and compromise in the nature of the response to pathogen infection vs cancer.

Relevant text to adjust here would be in the first paragraph of the Discussion.

4) I think the manuscript would benefit from addition of a short paragraph in the Discussion to highlight caveats with the current study, to include chiefly two points outlined below. I think this would be beneficial in acknowledging explicitly the speculative nature of some of the conclusions, and in drawing clear lines about next phases of the work, to be addressed in subsequent publications. Arguably the two most pressing caveats to highlight are:

a) The fact there is no direct assessment of decreased intracellular pathogen exposure in the NMR; in addition assessment of increased susceptibility to such infections (with the expectation of a lack of immunological defense) would be of interest.

b) The fact there is no evidence included for direct anti-cancer effector function of $\gamma\delta$ T cells from the NMR – something future studies could address.

Where some elements of these points are included in the current Discussion, I think they would be best coalesced into a separate 'caveat' paragraph.

Reviewer #3 (Remarks to the Author):

All observations raised by us have been addressed adequately and the paper in its current form results more readable and sound and could be considered for publication.

One last remaining minor issue concerns the trajectories depicted in figure 1J/K. We still believe that it should be clearly stated that the developmental trajectories are rather knowledge based rather than data driven to avoid possible confusion.

Reviewer #4 (Remarks to the Author):

As asked by the handling editor, I only assessed whether the authors have sufficiently addressed the comments from Rev 2. In my opinion, the authors have sufficiently addressed the technical concerns raised by Rev 2. However, Rev 2 also raised many conceptual concerns that the current study will not be able to address - such as the speculation that gdT cells in NMR contribute to longevity and cancer protection - without further and extensive experiments. Based on addressing technical concerns alone, then the authors have addressed Rev 2 comments. Based on the conceptual concerns Rev 2 raised (which often sounded more like ambiguous open-ended questions), it is impossible to address in the same study. However, my opinion is that this study does contribute to the T cell, and esp gdT cell, biology of the NMR that may be highly relevant to the broader immunology in the future. The authors may consider toning down their speculation on the potential role of gdT in NMR and instead stick to the known science and discussion based on their data instead of speculating too much.

Reviewer #1 (Remarks to the Author)

The authors have generally tried to address most concerns, however a few important points remain where this has not been done adequately.

1) Introduction

a) In relation to my 'Major Criticism 1' despite explicitly pointing out that the following sentence is 'severely flawed' (second paragraph of Introduction) this has been retained in unaltered form.

'Unlike $\alpha\beta$ T cells, $\gamma\delta$ T cells are not restricted to MHC-I-presented peptide antigens but rather, their less diverse $\gamma\delta$ TCR clonotypes are preprogrammed during thymic development to recognize a broad and ubiquitous set of both self and foreign stress-ligands, activating a rapid response^{16–18}.'

Just to be explicit, by 'severely flawed' I mean unacceptably so, thus requiring alteration. All the authors have done is insert references to studies that highlight innate-like and adaptive human subsets. However, retained as it is in its current unaltered form, it perpetuates misconceptions about the human $\gamma\delta$ T cell repertoire, as the sentence does not specify species, and it is objectively incorrect regarding human $\gamma\delta$ T cells. The comment about 'less diverse $\gamma\delta$ TCR clonotypes' that are 'preprogrammed during thymic development' applies well to the human innate-like V γ 9V δ 2 T cell subset focussed on a restricted number of (butyrophilin-family) ligands, whereas adaptive-like human $\gamma\delta$ T cells have an extremely diverse repertoire and are generated as TCR-diverse naïve cells, that are not pre-programmed in terms of effector function, and likely recognise diverse ligands. Also, from a ligand recognition perspective, the idea the current sentence provides, that decreased diversity in the $\gamma\delta$ T cell repertoire relative to $\alpha\beta$ T cells links with a much more diverse range of ligands, makes no sense. While (inconsistent with their earlier flawed sentence) the authors state later that adaptive-like human $\gamma\delta$ T cells are more diverse, this doesn't alter the fact that this first sentence is inaccurate, such that if a Masters student came out with it they would be marked down. Nature Communications deserves a little better.

Short of rewriting the current text, I suggest two solutions: option a) restrict the scope of this sentence to innate-like $\gamma\delta$ T cell populations that are pre-programmed for some kind of effector function (into which human innate-like V γ 9V δ 2 T cells would actually fit), and then later highlight TCR-diverse, adaptive-like $\gamma\delta$ T cells generated in naïve form but that can acquire effector function subsequently in response to eg infectious challenge. Option b) include both aspects of biology in this first sentence, ie both innate-like/restricted TCR repertoire/pre-programmed effector functionality vs adaptive/highly diverse repertoire/naïve functionality initially. I think Option a) might be less disruptive for the authors' current narrative.

>

We thank the reviewer for pointing this out. In our previous Introduction text the sentence highlighted in yellow below:

“Unlike $\alpha\beta$ T cells, $\gamma\delta$ T cells are not restricted to MHC-I-presented peptide antigens but rather, their less diverse $\gamma\delta$ TCR clonotypes are pre-programmed during thymic development to recognize a broad and ubiquitous set of both self and foreign stress-ligands, activating a rapid response”

was at odds with the with sentence one after it:

“Studies in humans have highlighted the presence of both innate-like $\gamma\delta$ T-cell subsets with semi-invariant TCR clonotypes as well as adaptive-like $\gamma\delta$ T-cell subsets with larger TCR clonotypic diversity^{20,21}“

We have now corrected that issue and completely revised that paragraph, and now write:

“During genomic rearrangement some of the T cells are committed to the $\gamma\delta$ lineage^{14,15}, which is not restricted to MHC-I-presented peptides, but rather the $\gamma\delta$ TCRs recognize both self and foreign stress-ligands¹⁶⁻¹⁸. $\gamma\delta$ T cells are the first to emerge in human and mouse embryonic thymi and subsequently mainly populate peripheral tissues, such as the skin and gut, rather than remain in the circulation^{17,19}. Studies in humans have highlighted the presence of $\gamma\delta$ T-cell subsets with semi-invariant and hence public (shared across individuals) clonotypes, which serve innate-like functions, as well as $\gamma\delta$ T-cell subsets with larger, and hence more private, clonotypes, which serve more adaptive-like roles²⁰⁻²³. Hence, $\gamma\delta$ T cells are thought to expand the temporal and spatial immune responsiveness of $\alpha\beta$ T cells, bridging the gap between innate and adaptive immunity²⁴⁻²⁶. In recent times, $\gamma\delta$ T cells have been shown to be involved in cancer, performing both tumorotoxic functions as well as proinflammatory and immunosuppressive roles that favor tumor growth^{27,28}.”

We have also added two additional citations to those requested by the reviewer.

b) In alignment with Reviewer 3, I agree the ‘immunological primer’ present in the opening paragraph of the Introduction should be largely cut and the main narrative should begin earlier. This will also help condense the paper marginally.

>

The Introduction has been significantly trimmed.

This opening paragraph is also not written terribly well – two sentences jar from an immunological angle.

Sentence 1:

‘Lymphocytes provide both innate and adaptive immunity through the ability of their receptors to sense malignancies presented by their major histocompatibility complex (MHC) ligands.’

This is very much a stretch vis a vis innate lymphocyte recognition and is not a great sentence. Even if one argues that NK receptor recognition of MHC molecules permits recognition of

tumours via missing self signatures on tumours exhibiting MHC loss, it is the integration of multiple signals from diverse receptors, some not recognising MHC molecules, that permits sensing of malignancies, rather than direct sensing of malignancies by MHC-focussed receptors, as implied.

>

We believe the problem was with the term “malignancies”, which we used in a general sense both for infections and for cancer. We have modified this sentence and now write:

“Lymphocytes provide both innate and adaptive immunity through the ability of their receptors to sense infections and other stressful conditions.”

Sentence 2: ‘CD8 T cells have evolved to eradicate cells infected by intracellular pathogens and early cancer transformed cells through the ability of their $\alpha\beta$ T-cell receptors (TCRs) to recognize intracellularly-produced foreign peptides presented on MHC-I of the affected cells^{1–6}.’

As written this slightly lazily seems to connect CD8 T cell TCR recognition of foreign peptides presented on MHC-I of affected cells with eradication of ‘early cancer transformed cells’ – which is confusing and wrong.

>

We have modified this sentence to make it clearer and now write:

“CD8 and CD4 T cells have largely evolved to eradicate intracellular and extracellular infections, respectively, through $\alpha\beta$ T-cell receptor (TCR) -recognition of peptides derived from these respective pathogens, presented on major histocompatibility complex (MHC) I and II, respectively^{1–7}.”

Overall I would cut this section.

>

We respectfully disagree. While that section might be obvious to the reviewer it is an important primer to less immunologically-informed readers of Nature Communications and hence we believe it is crucial to provide this background as our paper is heavily focused on the evolution of *MHC* and TCR gene families.

2) In relation to Major Criticism 2, ie the inconsistencies of the results of this study on thymic involution with previous studies in this area, the authors shy away from discussing this directly. I have two issues with this.

a) firstly, in the Introduction the narrative on this is highly confusing for a reader.

In paragraph 3 of the Introduction, the authors state in relation to previous NMR-related immunology research:

'in contrast to most vertebrates in which thymic involution is apparent by puberty⁴⁴, NMR thymi do not display signs of involution⁴⁰ even at ages twenty-fold greater than their age of sexual maturity²⁷'

And then in the final paragraph where they summarise their results they highlight that thymic involution is observed similar to mice.

'Consistent with that, we also observe that early-life NMR thymi are considerably smaller than those of mice, yet undergo similar involution progression, which is already apparent in young adults.'

This is highly confusing for readers. I appreciate the results are contradictory however given that, does it not make sense to caveat the previous work with eg 'IT HAS BEEN REPORTED that NMR thymi do not display signs of involution...'. Also, there is no acknowledgement of the contradiction in their final summary point in the Intro and this could be addressed 'also observe, CONTRARY TO PREVIOUS STUDIES, that early-life...'. Otherwise the understanding of the reader is sacrificed in a confusing narrative.

>

We have modified these parts of the Introduction and now write:

1. "(3) contrary to most vertebrates in which thymic cellularity is dramatically reduced by puberty⁴², this is not apparent in NMR thymi⁴³, even at ages twenty-fold greater than their age of sexual maturity²⁹;"
2. "Consistent with that, we also observe that early-life NMR thymi are considerably smaller than those of mice yet based on a comprehensive histological assessment, and contrary to Emmrich et al.'s report⁴³, undergo similar involution progression that is already apparent in young adults."

Hence, we now explicitly point out that the previous study used the less accurate cell counting approach while we used a comprehensive histological assessment.

b) secondly, regarding the request to discuss the contrary findings, the authors' riposte ('Mainly due to space considerations we did not devote lengthy text for discussing the disagreement between our results and those published by Emmrich et al., but rather left it to the readers and time for determining what is the more likely true biology.') is in my view weak. The idea that readers are going to have time to check out the technical vagaries of different approaches to assessment of thymic involution in the naked mole rat is somewhat wishful thinking. More importantly, as scientists, it is in my view not an unreasonable expectation of us to integrate our results with comparable datasets in the field. The authors are clearly best placed in the field to make this comparison, and also clearly have reasons to think their approach is superior, which sound reasonable (assessment of thymi over a wider life-span, and more comprehensive histological assessment, which sound reasonable). There could be a single/two-sentence

addition to the Discussion, and this would be more than compensated by curtailing their immunological primer section at the start of the Introduction.

>

We have modified the part in the Results text reporting these opposing findings and now write: “By counting thymocytes, Emmrich et al. reported that the NMR thymus increases in cellularity between 30 and 150 months with no concomitant reduction in thymic mass⁴³. Given our observations, obtained using the gold-standard histology approach over a wider age range, it is difficult to see how NMR thymic involution only starts after 150 months of age.”

Hence, we are explicitly pointing out, again, the possible reasons why our study reached different conclusions. We believe that repeating this point in the Discussion is excessive and thus suffice with this text.

3) Evolutionary Arguments

This relates to query 5 in my original review, and specifically point b, regarding the text ‘allowed the selective pressure under which NMRs have evolved to shift towards elimination of early malignancies and maintenance of tissue homeostasis, thereby promoting a long reproductive lifespan, to which their immune system has responded by utilizing $\gamma\delta$ T cells.’

The authors essentially agree with my suggestion that a better phrasing of the argument is around the idea that adaptations to pathogen-focussed immunity might compromise evolution of more potent anti-tumour immunity. However, the current wording in the manuscript still seems to be phrased around ‘selective pressure shifting’ towards cancer immunosurveillance, whereas the compromise argument is about a more focussed/adaptation of the response (immunological adaptation) to a consistent selective pressure around cancer development. This is a subtle but important distinction, and for the reader might make the difference between appearing to state that ‘evolution has now decided to tackle cancer’ to a more valid point (as outlined nicely in the author’s riposte to my point) about trade-offs and compromise in the nature of the response to pathogen infection vs cancer.

Relevant text to adjust here would be in the first paragraph of the Discussion.

>

We have modified our text and hopefully corrected this misunderstanding. We did not claim that adaptations to pathogen-focused immunity might compromise evolution of more potent anti-tumor immunity. Our point was that most mammalian genomes that we used show $\alpha\beta$ TCR and *MHC-I* genomic patterns consistent with strong selective pressure imposed by intracellular pathogens. In contrast, the hystricomorph genomes, with the exception of the guinea pig, show a pattern consistent with relaxation of that selective pressure. This is also consistent with the genomic pattern of NK-cell receptor genes and the loss of NMR NK cells. Thus, in a species such as mouse, whose survival is much more strongly threatened by spreads of intracellular-pathogen infections at any point in life, than of tumors late in life, and whose lifespan is relatively

short due to these factors as well as probably due to predation and above-ground climatic conditions, we would expect the immune system to be heavily invested in defense against intracellular pathogens and it might even be a stretch to claim that this comes at the expense of immunosurveillance. For the NMRs, we are proposing that their environment, in which isolated colonies occupy a sealed subterranean niche, considerably limits spreads of infections, and therefore might have been the environmental change that brought about the relaxation in intracellular pathogenic selective pressure. This, along with other factors, perhaps such as reduction in predation, may have allowed selection to shift towards longer lifespan hence operating on traits such as cancer resistance and tissue homeostasis, facilitated by utilizing the $\gamma\delta$ T cells. Again, claiming that this is the flip side of the trade off mentioned above would also be a stretch.

We therefore now write in our Discussion:

“These observations support the hypothesis that this occurred as a result of relaxed selective pressure imposed by intracellular pathogens³⁹. Although intracellular pathogens are thought to be one of the most dominant selective forces in mammalian evolution⁷², the subterranean ecological niche that NMRs occupy in isolated colonies, is likely an evolutionary dead end for mammalian intracellular pathogens due to its limiting effect on spread of infections. This contrasts with bats, which occupy a much more exposed and richer ecological niche and harbor more virus species than any other mammal⁷³. Thus, the selective pressure operating on NMRs might have shifted from defense against intracellular pathogens towards elimination of early malignancies, maintenance of tissue homeostasis, and perhaps defense against various extracellular pathogens, to which their immune system has responded by utilizing $\gamma\delta$ T cells”

Note that we include defense against various extracellular pathogens in addition to elimination of early malignancies and maintenance of tissue homeostasis, in the possible functions that NMR $\gamma\delta$ T cells may have evolved to carry out, in line with the request to tone down the focus on cancer resistance and accommodate other possible scenarios.

4) I think the manuscript would benefit from addition of a short paragraph in the Discussion to highlight caveats with the current study, to include chiefly two points outlined below. I think this would be beneficial in acknowledging explicitly the speculative nature of some of the conclusions, and in drawing clear lines about next phases of the work, to be addressed in subsequent publications. Arguably the two most pressing caveats to highlight are:

- a) The fact there is no direct assessment of decreased intracellular pathogen exposure in the NMR; in addition assessment of increased susceptibility to such infections (with the expectation of a lack of immunological defense) would be of interest.
- b) The fact there is no evidence included for direct anti-cancer effector function of $\gamma\delta$ T cells from the NMR – something future studies could address.

Where some elements of these points are included in the current Discussion, I think they would be best coalesced into a separate 'caveat' paragraph.

1. speculative nature

- a. There is no direct assessment of decreased intracellular pathogen exposure in the NMR and increased susceptibility to such infections has not been assessed
- b. There is no direct evidence of direct anti-cancer effector function of $\gamma\delta$ T cells from the NMR

>

We have reorganized our Discussion text by cutting out four separate speculative parts and combining them into a single paragraph which focuses on the caveats of our work.

The yellow highlighted texts below have been removed:

1. First paragraph: “These observations are all consistent with the hypothesis that this occurred as a result of relaxed selective pressure imposed by intracellular pathogens, albeit evidence of high susceptibility of NMRs to such insults is still limited.”
2. End of first paragraph: “By scRNA-seq profiling of T cells, in combination with our hybridization-capture TCR sequencing approach, it should be possible to test whether these dominant cytotoxic $\gamma\delta$ T-cell clonotypes become significantly more dominant in NMR tumor microenvironments (TMEs), indicating clonotypic expansion. Coupled with spatial transcriptomics on the same TME samples, it can be further tested whether the expanded clonotypes infiltrate the tumor, indicating they engage in cancer-cell killing. Given that NMR fibroblasts have been demonstrated to be susceptible to oncogenic transformation by the SV40 large T antigen and oncogenic HRAS (HRASG12V)³³, such suggested experiments are feasible. However, they critically depend on the ability of the transformed NMR cells to form tumors in the NMR host they are implanted in. This fails in immunocompetent mice because their CD8 T cells recognize the foreign T antigen and hence eliminate the transformed cells⁷⁵. Thus, other oncogenic transformation models, which do not suffer from this drawback, can serve as possible alternatives.”
3. End of second paragraph: “In humans, however, larger proportions of $\gamma\delta$ T cells in a diverse set of tumor types were found to be significantly associated with a favorable cancer prognosis, leading one to speculate that the large proportion of splenic $\gamma\delta$ T cells in NMRs might contribute to their cancer resistance.”
4. End of third paragraph: “It remains to be established whether these human and mouse CD8 α T-cell subsets are homologs of the NMR Cd8a-expressing T-cell subsets and the extent to which either have tumor infiltrating and cytotoxic capabilities”.

Our modified Discussion text is now:

“Our investigation of NMR T cells has uncovered several novel intriguing findings. NMRs have a splenic population of $\gamma\delta$ T cells at roughly the same proportion as that of mouse splenic NK cells, comprising two subsets. One of these $\gamma\delta$ T-cell subsets expresses an inhibited cytotoxic molecular profile homologous to that of mouse splenic NK cells, suggesting the two cell types are functionally similar yet are likely to differ in their activation mechanisms. Mouse $\gamma\delta$ T cells

mostly reside in epithelial tissues^{17,71} and the thymus is the only tissue where we observe them to express high levels of *Itgae*, whose encoded CD103 integrin binds to epithelial cadherin⁵³. If these *Itgae*-high mouse $\gamma\delta$ T cells are indeed homed to epithelial tissues as they exit the thymus⁵³, the absence of equivalent *Itgae* expression patterns in NMR $\gamma\delta$ T cells in the thymus may explain their large proportion in the circulation and hence spleen. Compared to other mammalian genomes, the NMR genome, like that of other hystricomorphs, with the exception of the guinea pig, has a considerably larger diversity of γ and δ variable TCR regions relative to that expected from its small diversity of α and β variable TCR regions. This may have evolved to generate a large clonotypic diversity of circulating $\gamma\delta$ T cells capable of recognizing a diverse spectrum of non-MHC-I ligands. In contrast, the small genomic diversity of NMR α and β variable TCR regions, the smaller proportion of the CD8 relative to the CD4 $\alpha\beta$ T-cell subset in the NMR spleen along with a corresponding bias in the clonotypic diversities of these NMR $\alpha\beta$ T-cell subsets, and the smaller size of the NMR *MHC-I* relative to its *MHC-II* gene family, corroborate the loss of NMR NK cells³⁹. These observations support the hypothesis that this occurred as a result of relaxed selective pressure imposed by intracellular pathogens³⁹. Although intracellular pathogens are thought to be one of the most dominant selective forces in mammalian evolution⁷², the subterranean ecological niche that NMRs occupy in isolated colonies, is likely an evolutionary dead end for mammalian intracellular pathogens due to its limiting effect on spread of infections. This contrasts with bats, which occupy a much more exposed and richer ecological niche and harbor more virus species than any other mammal⁷³. Thus, the selective pressure operating on NMRs might have shifted from defense against intracellular pathogens towards elimination of early malignancies, maintenance of tissue homeostasis, and perhaps defense against various extracellular pathogens, to which their immune system has responded by utilizing $\gamma\delta$ T cells. It is tempting to speculate that the dominant and public clonotype among the cytotoxic NMR $\gamma\delta$ T cells recognizes frequently encountered ligands that signal a threat of strong selective magnitude, which therefore must be rapidly eliminated.

The absence of *Cd8b* expression in NMR *Cd8a*-expressing T cells outside the thymus and the relaxed purifying selective pressure operating on the NMR *Cd8b* gene, suggest that the cytotoxic activity of NMR *Cd8a*-expressing T cells is regulated differently than of human and mouse CD8 $\alpha\beta$ T cells⁷⁴⁻⁷⁷. In these latter species the CD8 α -CD8 β heterodimer interacts with the monomorphic part of MHC-I thereby facilitating tight $\alpha\beta$ TCR-MHC-I binding and subsequent CD8-T-cell activation. Conversely, human and mouse CD8 $\alpha\alpha$ homodimers decrease TCR sensitivity to MHC-I⁷⁸, but can also facilitate cytotoxic-T-cell activation by interacting with non-classical MHC-I ligands⁷⁹. Human and mouse CD8 $\alpha\alpha$ T cells populate the gut as $\gamma\delta$ T cells⁸⁰ and the skin as $\alpha\beta$ T cells, and even comprise tumor-infiltrating innate-like T cells with high cytotoxic

potential (ILTCK) as $\alpha\beta$ T cells⁸¹. It remains to be established whether these human and mouse CD8 $\alpha\alpha$ T-cell subsets are homologs of the NMR *Cd8a*-expressing T-cell subsets.

Compared to the mouse genome the NMR genome has a larger diversity of γ and δ variable TCR regions. However, the absolute diversity of α and β variable TCR regions in the mouse genome is substantially larger than in the NMR genome, likely reflective of the stronger intracellular pathogenic selective pressure that mice have evolved under. This translates to a much larger potential diversity of mouse thymocyte clonotypes, which need to pass the strict selection imposed by thymic epithelial cells to ensure they are released into the circulation as functional and self-tolerant T cells, a process especially critical in early postnatal life when the various immune system compartments need to be populated^{42,82}. Based on the much larger potential diversity of mouse thymocyte clonotypes, the strikingly disparate early-life weights and subsequent rates of decline between the age trajectories of mouse and NMR thymi become intuitive. One of the theories explaining thymic involution suggests that once a T-cell clonotypic repertoire is established the metabolically costly process of thymopoiesis is turned off in order to divert energy to other physiological processes^{42,83}. This reasoning can explain the thymus age trajectories we observed in this study: the larger perinatal thymi of mice are required for generating their larger initial ‘pulse’ of T cells, with their overall larger clonotypic diversity and a subsequent steeper decline, yet by their equivalent middle age, thymi of both species show signs of involution.

Notwithstanding our novel findings our study has several limitations. Although the evidence for relaxed selective pressure imposed by intracellular pathogens under which NMRs have evolved is strong, data regarding NMR susceptibility to such insults is limited^{84,85} and challenging to generate because it is unclear which intracellular pathogens might be able to infect NMR cells. In addition, despite the surprisingly large fraction of splenic $\gamma\delta$ T cells in NMRs, and especially the subset with the inhibited cytotoxic transcriptional profile that resembles mouse splenic NK cells, we do not offer direct evidence regarding their effector functions. Hence whether and how these splenic $\gamma\delta$ T cells contribute to any of the unique NMR features, namely cancer resistance, remains to be investigated. A possible follow-up study to that end would involve single-cell and TCR profiling combined with spatial transcriptomics in NMR tumor microenvironments to test for the presence of expanded NMR $\gamma\delta$ T-cell clonotypes which engage in cancer-cell killing. This obviously depends on the ability to transform NMR cells and have them form tumors in an NMR host yet might be feasible as the former has already been demonstrated³⁵. It should be noted that

a large proportion of circulating $\gamma\delta$ T cells and large genomic diversities of γ and δ variable TCR regions is not unique to NMRs as these are also found in ruminants and equines⁸⁶⁻⁸⁸. Whether $\gamma\delta$ T cells in these species relate to cancer resistance, maintenance of homeostasis, and lifespan is also unknown and compounded by the possibility that these traits may have been affected by the domestication of these farm animals^{89,90}. In humans, however, larger proportions of $\gamma\delta$ T cells in a diverse set of tumor types were found to be significantly associated with a favorable cancer prognosis⁹⁰. Finally, phyletic patterns, considerably utilized in this work, are sensitive to the integrity levels of the genome assemblies. Although we limited our data to genomes assembled at the scaffold and chromosome levels, and benchmarked our approach using the well assembled and annotated human and mouse genomes, the gene-family sizes we obtained are likely only an approximation of their true magnitudes.”

The remainder of the Discussion text is unaffected by this point.

Reviewer #3 (Remarks to the Author)

All observations raised by us have been addressed adequately and the paper in its current form results more readable and sound and could be considered for publication.

One last remaining minor issue concerns the trajectories depicted in figure 1J/K. We still believe that it should be clearly stated that the developmental trajectories are rather knowledge based rather than data driven to avoid possible confusion.

>

We have made the following modifications to our text to make this point clear:

1. In the Methods section, in the last paragraph under the paragraph titled: "ScRNA-seq data and analysis" we added this text:

"Maturation trajectories of NMR and mouse thymocytes (Fig. 1J and K, respectively) were drawn based on biological knowledge since neither the R⁹⁷ Slingshot package¹⁰² nor the Python¹⁰³ Scanpy¹⁰⁴ diffMap^{101,105} and PAGA¹⁰⁵ methods produced trajectories that fit this well-established biological knowledge as it is laid out in the UMAP embedding space"

2. In the caption of Fig 1 we added the sentence:

"Maturation trajectory arrows in K and L are knowledge based"

Reviewer #4 (Remarks to the Author):

As asked by the handling editor, I only assessed whether the authors have sufficiently addressed the comments from Rev 2. In my opinion, the authors have sufficiently addressed the technical concerns raised by Rev 2. However, Rev 2 also raised many conceptual concerns that the current study will not be able to address - such as the speculation that gdT cells in NMR contribute to longevity and cancer protection - without further and extensive experiments. Based on addressing technical concerns alone, then the authors have addressed Rev 2 comments. Based on the conceptual concerns Rev 2 raised (which often sounded more like ambiguous open-ended questions), it is impossible to address in the same study. However, my opinion is that this study does contribute to the T cell, and esp gdT cell, biology of the NMR that may be highly relevant to the broader immunology in the future. The authors may consider toning down their speculation on the potential role of gdT in NMR and instead stick to the known science and discussion based on their data instead of speculating too much.

>

We believe that the new Discussion paragraph, which writes about the caveats of our work (last point of reviewer #1), satisfies this comment.

REVIEWERS' COMMENTS

Reviewer #1 (Remarks to the Author):

The authors have done a good job at addressing my remaining comments, and I have only extremely minor comments remaining, which the Editor can deal with. I congratulate the authors on nice final manuscript.

To address my comments and the changes in order:

1) Introduction

Point a) has been nicely addressed with text changes and new references included.

Regarding point b), this has mainly been addressed but one aspect, the opening sentence of the manuscript is still suboptimal in simply removing 'malignancies' and replacing with 'stressful conditions', and does not take into account the point that it is the integration of signals rather than individual receptors, that detect a 'stressful condition/malignancy'.

I would suggest the following rewording:

"Lymphocytes provide both innate and adaptive immunity, sensing infections and other stressful conditions via integration of signals from cell surface receptors."

2) I am happy with the authors' response regarding integration (or differences) of the results with a previous study. The text changes in the Introduction and in the Results make things clearer re dramatic cellular reduction (or lack thereof) in the thymus and thymic involution, and also why the current results likely differ from those of Emrich et al.

3) Regarding the evolutionary related queries, the authors have clearly considered my points and have made appropriate changes to the text, that clarify the relevant issues.

4) As requested, the authors have collated some disparate sentences in the manuscript into a coherent paragraph focussed on the caveats of the study. This is a nice addition to the manuscript and will help the reader considerably.

A final request would be rewording of one (now modified) sentence in the abstract, which reads as follows:

"Using single-cell transcriptomics we find that NMRs have a large circulating population of the unconventional $\gamma\delta$ T cells, which in mice and humans mostly reside in peripheral tissues and induce cancer cytotoxicity."

(i) Use of the word 'unconventional'

Aside from the grammatically incorrect 'the' in front of 'unconventional', the use of 'unconventional' itself adds ambiguity here. I assume the intended meaning is merely that as opposed to $\alpha\beta$ T cells (typically regarded as 'conventional'), these are $\gamma\delta$ T cells we are talking about and hence 'unconventional' by definition. If this is the case, then 'unconventional' can just be removed because it does not add anything above stating they are $\gamma\delta$ T cells. Alternatively it could mean that the type of $\gamma\delta$ T cell present is unconventional relative to other $\gamma\delta$ T cells – which might be the meaning understood from within the $\gamma\delta$ T cell community when reading the paper.

(ii) The current wording, partly due to lack of a comma, implies that in mice and humans $\gamma\delta$ T cells may mostly induce cancer cytotoxicity – which actually likely goes against a relevant implication of the study. I would suggest the following rewording of this sentence:

"Using single-cell transcriptomics we find that NMRs have a large circulating population of $\gamma\delta$ T cells, a compartment which in mice and humans mostly resides in peripheral tissues, and can mediate anti-tumour cytotoxicity."

Reviewer #3 (Remarks to the Author):

Authors have addressed all concerns raised by us in previous rounds of revision in a satisfactory way. For this reason and owing to the relevance of the work in the field of comparative immunology, we would support its publication.

REVIEWERS' COMMENTS

Reviewer #1 (Remarks to the Author):

The authors have done a good job at addressing my remaining comments, and I have only extremely minor comments remaining, which the Editor can deal with. I congratulate the authors on nice final manuscript.

- We thank the reviewer for his comments thorough the review process.

To address my comments and the changes in order:

1) Introduction

Point a) has been nicely addressed with text changes and new references included.

Regarding point b), this has mainly been addressed but one aspect, the opening sentence of the manuscript is still suboptimal in simply removing 'malignancies' and replacing with 'stressful conditions', and does not take into account the point that it is the integration of signals rather than individual receptors, that detect a 'stressful condition/malignancy'.

I would suggest the following rewording:

"Lymphocytes provide both innate and adaptive immunity, sensing infections and other stressful conditions via integration of signals from cell surface receptors."

- We thank the reviewer for their suggestion. We think that the opening sentence of our Introduction, which writes: "Lymphocytes provide both innate and adaptive immunity through the ability of their receptors to sense infections and other stressful conditions" is, on the one hand, inclusive and does not suggest that lymphocytes do not integrate the signals they perceive through their individual receptors, yet on the other hand satisfies comments raised by this reviewer and others, asking us to keep the immunological primer of our Introduction short, as well as meeting the journal's word-count limit.

2) I am happy with the authors' response regarding integration (or differences) of the results with a previous study. The text changes in the Introduction and in the Results make things clearer re dramatic cellular reduction (or lack thereof) in the thymus and thymic involution, and also why the current results likely differ from those of Emrich et al.

3) Regarding the evolutionary related queries, the authors have clearly considered my points and have made appropriate changes to the text, that clarify the relevant issues.

4) As requested, the authors have collated some disparate sentences in the manuscript into a coherent paragraph focussed on the caveats of the study. This is a nice addition to the manuscript and will help the reader considerably.

A final request would be rewording of one (now modified) sentence in the abstract, which reads as follows:

“Using single-cell transcriptomics we find that NMRs have a large circulating population of the unconventional $\gamma\delta$ T cells, which in mice and humans mostly reside in peripheral tissues and induce cancer cytotoxicity.”

(i) Use of the word ‘unconventional’

Aside from the grammatically incorrect ‘the’ in front of ‘unconventional’, the use of ‘unconventional’ itself adds ambiguity here. I assume the intended meaning is merely that as opposed to $\alpha\beta$ T cells (typically regarded as ‘conventional’), these are $\gamma\delta$ T cells we are talking about and hence ‘unconventional’ by definition. If this is the case, then ‘unconventional’ can just be removed because it does not add anything above stating they are $\gamma\delta$ T cells. Alternatively it could mean that the type of $\gamma\delta$ T cell present is unconventional relative to other $\gamma\delta$ T cells – which might be the meaning understood from within the $\gamma\delta$ T cell community when reading the paper.

(ii) The current wording, partly due to lack of a comma, implies that in mice and humans $\gamma\delta$ T cells may mostly induce cancer cytotoxicity – which actually likely goes against a relevant implication of the study. I would suggest the following rewording of this sentence:

“Using single-cell transcriptomics we find that NMRs have a large circulating population of $\gamma\delta$ T cells, a compartment which in mice and humans mostly resides in peripheral tissues, and can mediate anti-tumour cytotoxicity.”

- We have followed this comment, integrating it with editing suggestions provided by the journal, and have changed that Abstract sentence to: “Here we find, using single-cell transcriptomics, that NMRs have a large circulating population of $\gamma\delta$ T cells, which in mice and humans mostly reside in peripheral tissues and induce anti-cancer cytotoxicity.”

Reviewer #3 (Remarks to the Author):

Authors have addressed all concerns raised by us in previous rounds of revision in a satisfactory way. For this reason and owing to the relevance of the work in the field of comparative immunology, we would support its publication.

- We thank the reviewer for his comments thorough the review process.